# Enhancing Causal Reasoning in Large Language Models: A Causal Attribution Model for Precision Fine-Tuning

## Abstract

This paper introduces a causal attribution model to enhance the interpretability of large language models (LLMs) and improve their causal reasoning abilities via precise fine-tuning. Despite LLMs' proficiency in diverse tasks, their reasoning processes often remain black box and thus restrict targeted enhancement. We propose a novel causal attribution model that utilizes "do-operators" for constructing interventional scenarios, allowing us to quantify the contribution of different components in LLMs's causal reasoning process systematically. By assessing the proposed attribution scores through causal discovery tasks across various domains, we demonstrate that LLMs' effectiveness in causal discovery heavily relies on provided context and domain-specific knowledge but can also utilize numerical data with limited calculations in correlation, not causation. This motivates the proposed fine-tuned LLM for pairwise causal discovery, effectively and correctly leveraging both knowledge and numerical information.

## 1 Introduction

Large language models (LLMs) have been at the forefront of advancing artificial intelligence, marking significant breakthroughs in diverse fields (Vaswani et al., 2017; Kenton & Toutanova, 2019; Lewis et al., 2019; Brown et al., 2020; Neelakantan et al., 2022; Stiennon et al., 2020; OpenAI, 2023). Despite the proficiency of LLMs in a range of tasks (Floridi, 2019; Jiang et al., 2019; Tambe et al., 2020), their reasoning capabilities, particularly in causal reasoning (Pearl, 2009a; Schölkopf et al., 2021), are yet less investigated (see a recent survey in Liu et al.). Causal reasoning, central to human cognition, enables understanding and predicting the consequences of events and actions (Spirtes et al., 2000b; Glymour & Zhang, 2019). It is essential in higher-level cognitive tasks such as decision-making, problem-solving, and understanding complex narratives (Hagmayer & Sloman, 2013; Griffiths et al., 2019). Equipping LLMs with causal reasoning abilities thus is a significant leap from mere pattern recognition to a profound understanding of real-world phenomena (Lake et al., 2017; Tenenbaum et al., 2019; Marcus, 2020; Chen et al., 2021; Jiralerspong et al., 2024). On the other hand, LLM-based causal reasoning recently has become an attractive and demanding paradigm, by leveraging the large knowledge base and strong expressive abilities of LLMs for important causal inference in daily analyses, as supported by fast-growing literature (see e.g., Riedel et al., 2019; Keith et al., 2020; Weidinger et al., 2021). This raises two critical questions:

*Can LLMs really understand causal relationships? If not, how to effectively bridge the gap?*

Several recent studies have attempted to explore the first question. Kiciman et al. (2023) considered integrating the variable names into a text template and tasked LLMs with discerning the cause among the options. Gao et al. extended such an evaluation on the contexts of a board causal knowledge graph and found LLMs outperformed traditional machine learning methods. Jin et al. (2024) also examined the LLMs' capacity for causal discovery via correlation descriptions and tested if LLMs can determine the true causalities. Recently, Jin et al. (2023) proposed to assess causal reasoning in LLMs and revealed that LLMs struggled with the complex dataset. Zečević et al. further argued that LLMs seem to succeed in causal inference simply by reciting the knowledge embedded. Despite their insights, all these works mainly focus on *context information* without investigation on *numerical data* known to intrinsically reflect causalities (Pearl, 2009a).

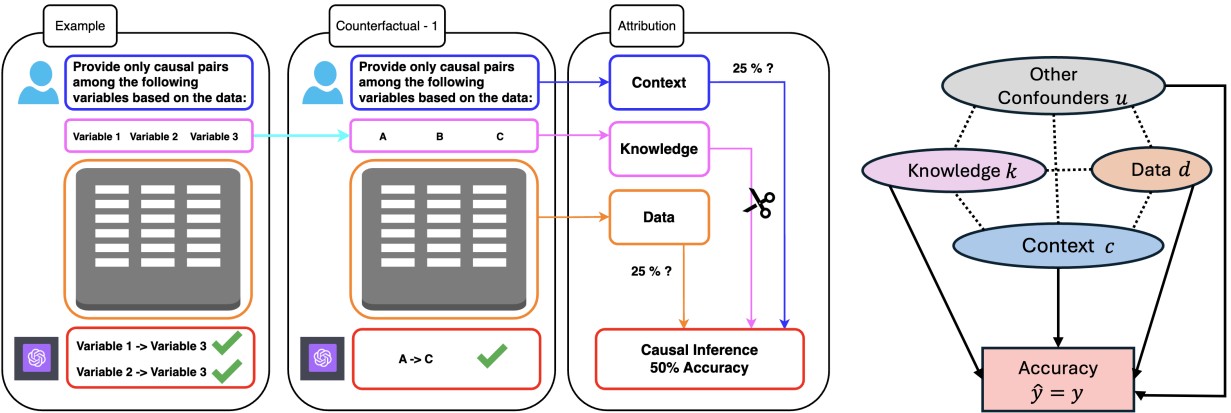

Figure 1: Left: The first panel presents one sample using the LLM to provide causal discovery results, where the blue box is the i-th context, the pink box is the i-th knowledge embedded in variable names, the orange box is the i-th numerical information, and the red box at the last is the i-th output. The second panel shows the generation of one interventional sample where the variable names were replaced by non-meaningful letters; and the third one describes our proposed causal attribution framework, where the knowledge is omitted corresponding to the interventional scenario in the second panel. Right: The assumed causal graph, where the main outcome of interest, i.e., the causal reasoning accuracy made by an LLM $\hat{y} = y$ is determined by the input context $c$, the embedded causal knowledge $k$, the numerical data $d$, and all other controlled confounders $u$; while we do not specify the relationship between $c$, $k$, $d$, and $u$, presented by the dash lines.

In this paper, to answer the first question, we propose a systematic and general evaluation framework to disentangle how LLMs understand causal relationships or similar reasoning processes. The main technique lies in *generating interventional examples*, which allows us to produce all combinations of different input components to observe changes in model output, and thus quantifies the influence of each individual component on model causal reasoning performance. In particular, we focus on the causal discovery tasks and are interested in the roles of the inherent causal knowledge embedded within the variable names, the explicit numerical data that denote causal links, and the context provided for the causal task. Refer to Figure 1 for an illustration. The conclusion of our attribution model further locates the directions for LLMs to improve their causal reasoning and motivates our fine-tuned LLM for causal discovery, which thus solves the second question. The **contributions** of our research are threefold.

● We develop *a causal attribution model* and propose the definitions of marginal and conditional attributions of knowledge and data, through the notion of "do-operators" (Pearl, 2009a). The proposed definitions differentiate and quantify the effects of different components in LLM's reasoning processes.

● We design a series of novel experiments to estimate proposed attribution scores through causal discovery tasks across various domains. Our evaluation reveals that LLMs' effectiveness in causal reasoning *heavily relies on provided context and domain-specific knowledge* but can also utilize numerical data with *limited calculations in correlation, not causation.*

● To effectively and correctly leverage both knowledge and numerical information, this work firstly designs a *precision fine-tuning* to enhance the causal discovery abilities in LLMs. Our fine-tuned model achieves the highest accuracy among LLMs in identifying the true causal relationships without losing generalizability on other tasks. The Python implementation of our method and synthetic data for fine-tuning is available in supplementary material.

## 2 Proposed Framework

### 2.1 Causal Attribution Model

We formalize the causal attribution model to understand how different components affect the performance of LLMs. For the $i$-th sample in our dataset, we decompose the input into several distinct elements. Here,

we focus on the causal reasoning task for LLMs to detail these components as follows: the input context $c_i$, which provides the scenario for the causal reasoning task; the embedded causal knowledge $k_i$ can take many formats and considering causal discovery tasks, we view variable names as knowledge which may carry implicit causal cues; and the numerical data $d_i$, representing explicit causal relationships which may take forms as summary statistics or other numerical information. All other controlled confounding variables that we do not include as interventions, such as training data, language model version, etc., are denoted as $u_i$. The goal is to compare the causal reasoning made by an LLM, represented as $\widehat{y}_i$, against the true causal responses $y_i$. We first introduce the attribution to knowledge given fixed data to quantify the impact of knowledge.

**Definition 2.1.** Conditional Attribution of Knowledge (CAK) Given Data:

$$CAK_i = \mathbb{P}\left(\widehat{y}_i = y_i \mid do\left(k_i = k_i, d_i = d_i\right), c_i, u_i\right) - \mathbb{P}\left(\widehat{y}_i = y_i \mid do\left(k_i = \emptyset, d_i = d_i\right), c_i, u_i\right),$$

where the do-calculus $do(A = a)$ is a mathematical operator (Pearl, 2009a) to simulate interventions that hold $A$ constant as $a$ while keeping the rest of the model unchanged, and the probability $\mathbb{P} := \mathbb{P}_S$ is defined on a structural equation model (SEM) $S$ that is compressed for clear presentation.

This equation defines the probability difference of an LLM making a correct causal inference when variable names containing potential causal knowledge are included versus when they are absent, given the same data. A $CAK_i$ value near zero suggests that embedded knowledge has a negligible effect on the model's causal reasoning accuracy. Similarly, we define the attribution to data given the knowledge below.

**Definition 2.2.** Conditional Attribution of Data (CAD) Given Knowledge:

$$CAD_i = \mathbb{P}\left(\widehat{y}_i = y_i \mid do\left(k_i = k_i, d_i = d_i\right), c_i, u_i\right) - \mathbb{P}\left(\widehat{y}_i = y_i \mid do\left(k_i = k_i, d_i = \emptyset\right), c_i, u_i\right).$$

Here, we assess the impact of numerical data on causal inference accuracy by comparing the LLM's performance with full numerical data access to its performance with numerical data systematically removed while retaining the embedded knowledge. The first term in Definitions 2.1 and 2.2 represents the original prediction accuracy with full information. The second term shows the prediction accuracy after omitting knowledge or data. We further examine the marginal attributions for data.

**Definition 2.3.** Marginal Attribution of Data (MAD):

$$MAD_i = \mathbb{P}\left(\widehat{y}_i = y_i \mid do\left(k_i = \emptyset, d_i = d_i\right), c_i, u_i\right) - \mathbb{P}\left(\widehat{y}_i = y_i \mid do\left(k_i = \emptyset, d_i = \emptyset\right), c_i, u_i\right).$$

Here, the marginal attribution of the data measures the impact of data alone without any additional knowledge. This distinction is crucial to evaluating the independent effectiveness of the data in the inference process. In a similar logic, we define the marginal attribution of knowledge as follows.

**Definition 2.4.** Marginal Attribution of Knowledge (MAK):

$$MAK_i = \mathbb{P}\left(\widehat{y}_i = y_i \mid do\left(k_i = k_i, d_i = \emptyset\right), c_i, u_i\right) - \mathbb{P}\left(\widehat{y}_i = y_i \mid do\left(k_i = \emptyset, d_i = \emptyset\right), c_i, u_i\right).$$

Conversely, this attribution evaluates the unique influence of knowledge embedded in variable names when numerical data are deliberately omitted. The second term, the baseline, in Definitions 2.3 and 2.4, is identical. When these components are manipulated independently, we can determine their separate contributions to the LLM's causal reasoning. Combining Definitions 2.1 to 2.4, we derive the following relationship:

$$MAD_i - MAK_i = CAD_i - CAK_i.$$

Here, the difference between MAK and MAD shows the discrepancy between the marginal attributions in terms of knowledge and data, i.e., how marginally including knowledge or data would increase the accuracy. This difference reflects the more important factor among knowledge and data, for LLMs' reasoning process. Similarly, the difference between CAK and CAD helps to identify the dominating factor while conditional on the other factor's information. The above equation indicates that no matter which difference (either MAK-MAD or CAK-CAD), the conclusion is **consistent** and thus shows the robustness of our model in attributing significance to different inputs.

Table 1: Description of nine benchmark datasets for causal discovery tasks.

| Dataset | Galton | Sachs | Alcohol | EcoSystem | MPG | DWD | Cement | Stock | Arrhythmia |
|---|---|---|---|---|---|---|---|---|---|
| Number of nodes | 4 | 12 | 6 | 4 | 5 | 6 | 9 | 5 | 4 |
| Number of causal relations | 3 | 20 | 5 | 3 | 6 | 6 | 8 | 3 | 3 |
| Number of samples | 898 | 7466 | 345 | 721 | 392 | 349 | 1030 | 1331 | 450 |
| Domain | Biology | Biology | Biology | Physics | Engineering | Geography | Engineering | Finance | Biology |

## 2.2 Causal Discovery Task and Terminology

Following the existing literature (see e.g., Kiciman et al., 2023; Jin et al., 2024), we focus on causal discovery (Spirtes et al., 2000b; Pearl, 2009b) as the main causal reasoning task. We first detail necessary terminologies. Consider a graph $\mathcal{G} = (\boldsymbol{X}, \boldsymbol{D_X})$ with a node set $\boldsymbol{X}$ and an edge set $\boldsymbol{D_X}$. A node $X_i$ is said to be a parent of $X_j$ if there is a directed edge from $X_i$ to $X_j$, i.e., $X_i$ is a direct cause of $X_j$. A directed graph $\mathcal{G}$ that does not contain directed cycles is called a directed acyclic graph (DAG). The DAG $\mathcal{G}$ can be estimated up to a Markov equivalence class (MEC) based on observational data (Pearl, 2009b; Peters & Bühlmann, 2014), with a number of causal discovery methods developed recently (see e.g., Spirtes et al., 2000a; Shimizu et al., 2006; Bühlmann et al., 2014; Ramsey et al., 2017; Zheng et al., 2018; Yu et al., 2019; Zhu et al., 2020). Throughout this paper, we task the LLM with causal discovery to identify the graph $\mathcal{G}$ based on the provided information. The advantages of using LLMs can be witnessed by their better performance of accuracy in Tables 3 and E.1-E.3 compared with the classical causal discovery methods in Table B.1.

## 3 Experiment Design and Estimation

We detail the experimental design tailored to estimate the proposed attribution scores. We first introduce the dataset for the causal discovery task in Section 3.1. More preliminaries and related works are provided in Appendix A. We detail the system prompt in Section 3.2. Subsequent experiments are then designed to measure LLMs' performance in the absence of knowledge (see Section 3.3), data (see Section 3.4), or both (see Section 3.5) for general causal discovery tasks with multiple variables. We start to examine LLMs's abilities in a pairwise causal discovery task in Section 3.6. Beyond the literature, we further design a reverse causal discovery task in Section 3.7 that inverts the causal directions by switching the numerical data.

## 3.1 Data Construction for Causal Discovery

Our experimental design utilizes nine distinct datasets (Mooij et al., 2016; Zheng et al., 2024), each containing verified causal relationships established through expert knowledge and empirical analysis. These datasets benchmark the LLMs' performance in causal discovery tasks and span various domains—biology, physics, geography, atmospheric science, finance, and engineering—ensuring diverse reasoning scenarios. Table 1 provides a detailed description of the datasets, including the number of nodes, causal relations, samples, and their respective domains. Additional details are in Appendix B. Each dataset includes variable names with causal meanings and numerical data supporting intrinsic causal relationships. In this paper, we consider generating data $\{y_i, k_i, d_i, \widehat{y}_i\}_{i=1}^n$ based on the above benchmark datasets. Specifically, we consider one benchmark data and randomizing the variable/column ordering as one sample, with 15 randomization in total, which yields $n = 135$ samples overall. We aim to evaluate the LLMs' causal discovery results against the ground truth of these causal pairs.

## 3.2 System Prompt Generation

The initial phase of our experimental design employs zero-shot prompting strategies to engage LLMs in causal discovery tasks (Kojima et al., 2022). Our goal is to generate a comprehensive prompt to enhance the accuracy of LLMs' causal discovery, establishing the baseline accuracy component, i.e., the first term $(\mathbb{P}\left(\widehat{y}_i = y_i \mid do\left(k_i = k_i, d_i = d_i\right), c_i, u_i\right))$ in Definitions 2.1 and 2.2. Preliminary trials reveal that prompts containing directives like "provide" often elicit non-committal responses from LLMs. This indicates a reluctance or inability to generate causal analyses without additional context. To address this, we modify our prompts to

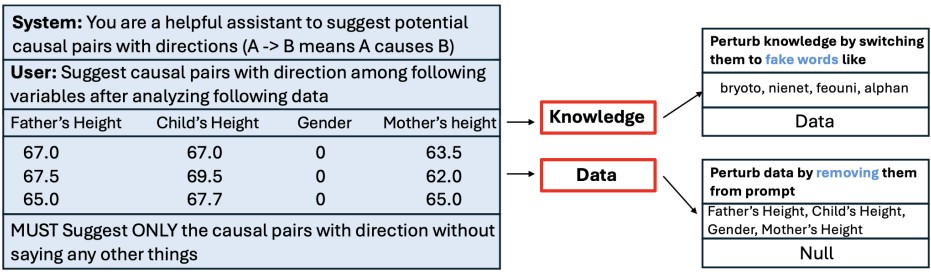

Figure 2: Experiment design for LLMs' answering causal questions with encouraging prompts.

use more suggestive language, such as "suggest", guiding LLMs to produce more analytically useful responses. Figure 2 illustrates the attribution ability of ChatGPT in answering causal questions with optimized prompts. We observed that LLMs might infer causal relationships based on the sequence of data columns, likely due to biases from the pre-training phase. Specifically, LLMs often assume the first variable causes the second, and the second causes the third. To counter this bias, we recommend randomizing column order in each dataset for every experiment and tracking accuracy across multiple replications.

### 3.3 Ability Attribution: Omit Knowledge

To examine the internal knowledge of LLMs, referred to $\mathbb{P}\left(\widehat{y}_i = y_i \mid do\left(k_i = \emptyset, d_i = d_i\right), c_i, u_i\right)$ in Definition 2.1, we conduct experiments to assess the LLM performance when explicit knowledge is systematically restricted. Recognizing that LLMs possess vast stores of implicit knowledge, we aim to understand the influence of this internal knowledge on causal discovery. To isolate the effect of knowledge from other variables, we retain the numerical data but obscure the variable names by substituting them with arbitrary terms like "bryoto", "nienet", and "feouni" (see Figure 2). These placeholders are chosen deliberately to avoid sequences that LLMs might interpret as inherently ordered or connected, such as alphabetical sequences or familiar names. This strategy prevents the models from leveraging pre-existing associative patterns.

### 3.4 Ability Attribution: Omit Data

We develop a subsequent experiment to measure the LLM performance in the absence of data, i.e., $\mathbb{P}\left(\widehat{y}_i = y_i \mid do\left(k_i = k_i, d_i = \emptyset\right), c_i, u_i\right)$ in Definition 2.2. To this end, we exclude data values from the prompt and present only the column names in random order, intentionally omitting numerical data, as shown in Figure 2. This method allows us to assess the LLMs' capacity for reasoning with structural but non-quantitative information, contrasting their operation with full data availability. This technique has been previously implemented to some extent (e.g., Kiciman et al., 2023), where LLMs were tasked with causal discovery using limited variable sets, each containing 3-4 variables. These studies demonstrated LLMs' responses when prompted with only variable names and metadata. Our experiment builds on this foundation by scaling up to seven datasets with varied complexity, including the Sachs dataset that encompasses numerous variables and interactions (Sachs et al., 2005).

### 3.5 Ability Attribution: Random Guess

To set the baseline for LLMs' performance without input data and knowledge, we design an experiment of random guess to omit both data and knowledge inputs. This approach estimates $\mathbb{P}\left(\widehat{y}_i = y_i \mid do\left(k_i = \emptyset, d_i = \emptyset\right), c_i, u_i\right)$ in Definitions 2.3-2.4, where LLMs must operate without informative cues, relying on random guesses to generate causal pairs. The rationale is twofold: first, to quantify the lowest bound of the LLM accuracy in causal discovery by omitting meaningful input, and second, to assess the models' default response patterns in the absence of guiding information.

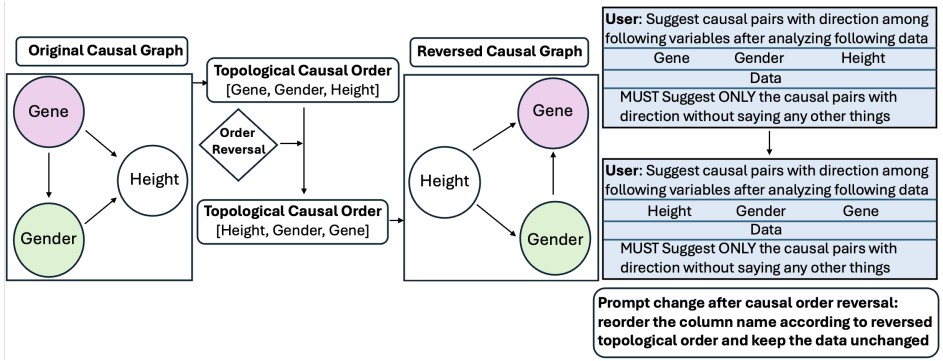

Figure 3: Illustration of the reverse causal discovery task.

### 3.6 Pairwise Causal Discovery Task

We detail the pairwise causal discovery task (Hoyer et al., 2008), where the causal relationships can be uniquely identified based on observational data with non-Gaussian noises (Shimizu et al., 2006; Peters & Bühlmann, 2014; Loh & Bühlmann, 2014; Zheng et al., 2020). Causal discovery beyond two variables follows the same logic. We design the experiments by the theorem below.

**Theorem 3.1** ( Shimizu et al. (2006))**.** *In the linear non-Gaussian noise setting, if the true structural causal model is $Y := f(X) + U$, $X \perp U$, then there does not exist a structural causal model in the reverse direction $X := g(Y) + \tilde{U}$, $Y \perp \tilde{U}$ that can generate data consistent with $P(x, y)$.*

Using the Galton Family dataset as an example, we simulate Father's Height data by Father's Height := $f$(Child's Height) + U, where $f$ is a linear function and $U$ is non-Gaussian noise, such as chi-squared noise. The evaluation is conducted under three scenarios: (1) employing LiNGAM directly on simulated data as a baseline, where LiNGAM uses the independent component analysis (ICA) (Comon, 1994; Hyvärinen et al., 2009; Shimizu et al., 2006) to estimate causal ordering and connection strengths based on non-Gaussianity; (2) formulating the prompt using data with variable names for LLMs; and (3) constructing the prompt with data and variable names, complemented by simplified instructions to execute LiNGAM for LLMs. The detailed procedure is shown in Figure D.1.

### 3.7 Reverse Causal Discovery

The reverse causal discovery task is critical for evaluating the LLMs' dependence on numerical data when inferring causality (see the detailed workflow in Figure 3). We achieve this by manipulating the structure of the datasets while keeping their data content intact. Specifically, we first establish the topological ordering of the variables within the original DAGs (Pearl, 2009b) representing the causal structures of the datasets. We then systematically reverse this order using a transposed adjacency matrix, which maintains a one-to-one mapping with the original causal graph structure, creating a new set of inverted relationships for the LLMs to analyze. Despite preserving the data values, this rearrangement challenges the models by disrupting the directionality they have learned from previous exposures. As illustrated in Figure 3, the original causal links (e.g., Gene leading to Height and Gender, with Gender influencing Height) are reversed, presenting the LLMs with the premise that Height influences Gender and Gene. We evaluate the reverse task prediction accuracy with respect to the original graph structure. If there is minimal impact of changing causal topologies on LLMs' discovery outcomes, this suggests that LLMs primarily depend on their inherent knowledge to deduce causal relationships among variables.

### 3.8 Estimations of Attribution Scores

We estimate the attribution scores, $CAK$, $CAD$, $MAD$, and $MAK$, via the designed experiments. Specifically, we utilize the true discovery rate (TDR) w.r.t. to the whole graph to access the accuracy of predication of causal discovery results:

$$\text{TDR} = \frac{\text{\# Correctly Predicted Causal Pairs}}{\text{\# True Causal Pairs}}.$$

Table 2: Attribution scores of LLMs for different datasets.

| Method / | Dataset | Sachs | Galton | Alcohol | EcoSystem | MPG | DWD | Cement | Stock | Arrhythmia |
|---|---|---|---|---|---|---|---|---|---|---|
| **GPT-4 turbo** | CAK | 0.49±0.21 | 0.58±0.36 | 0.75±0.26 | 0.33±0.35 | 0.51±0.22 | 0.17±0.14 | 0.87±0.16 | 0.07±0.30 | 0.09±0.26 |
| | CAD | 0.01±0.38 | 0±0.07 | 0.01±0.10 | 0.20±0.23 | -0.05±0.18 | -0.01±0.19 | 0±0.04 | -0.12±0.41 | -0.13±0.39 |
| | MAD | 0.04±0.14 | 0.02±0.40 | -0.03±0.60 | 0.03±0.46 | 0.16±0.19 | 0.29±0.09 | -0.62±0.38 | 0.05±0.39 | 0.39±0.23 |
| | MAK | 0.52±0.29 | 0.60±0.49 | 0.70±0.44 | 0.16±0.49 | 0.72±0.13 | 0.48±0.20 | 0.26±0.48 | 0.24±0.46 | 0.62±0.24 |
| **GPT-4** | CAK | 0.01±0.30 | 0.67±0.28 | 0.58 ±0.28 | 0.41±0.36 | 0.28±0.21 | 0.07±0.20 | 0.79±0.09 | 0.12±0.09 | 0.15±0.27 |
| | CAD | -0.17±0.28 | 0±0.05 | 0.08±0.13 | 0.02±0.27 | -0.14±0.18 | -0.10±0.20 | 0.02±0.26 | -0.12±0.22 | -0.10±0.19 |
| | MAD | 0.26±0.17 | 0.18±0.46 | 0.29±0.40 | 0.29±0.30 | 0.32±0.40 | 0.33±0.31 | 0.01±0.48 | 0.04±0.39 | 0.44±0.28 |
| | MAK | 0.45±0.23 | 0.85±0.36 | 0.79±0.36 | 0.67±0.38 | 0.74±0.34 | 0.51±0.28 | 0.78±0.53 | 0.28±0.43 | 0.70±0.27 |
| **GPT-3.5** | CAK | 0.20±0.30 | 0.42±0.32 | 0.13 ±0.40 | 0.24±0.35 | 0.19±0.24 | 0.09±0.24 | 0.88±0.19 | 0.08±0.12 | 0.18±0.37 |
| | CAD | 0.04±0.37 | 0.01±0.23 | -0.14±0.39 | 0.12±0.20 | -0.09±0.34 | -0.15±0.24 | 0±0.23 | -0.03±0.28 | 0.14±0.13 |
| | MAD | 0.05±0.36 | 0.17±0.45 | 0.33±0.34 | 0.01±0.37 | 0.02±0.21 | 0.06±0.26 | 0.02±0.08 | 0.06±0.18 | 0.01±0.29 |
| | MAK | 0.20±0.24 | 0.58±0.29 | 0.60±0.41 | 0.13±0.37 | 0.30±0.28 | 0.30±0.25 | 0.91±0.08 | 0.17±0.30 | 0.05±0.36 |
| **LLaMa2-13B** | CAK | 0.08±0.27 | 0.09±0.26 | 0.05±0.23 | 0.14±0.35 | 0.32±0.34 | 0.23±0.22 | 0.51±0.24 | 0.09±0.32 | 0.11±0.22 |
| | CAD | -0.04±0.26 | -0.48±0.18 | -0.11±0.26 | -0.07±0.36 | -0.04±0.38 | 0.18±0.21 | -0.15±0.34 | 0.12±0.24 | 0.08±0.21 |
| | MAD | 0.01±0.25 | 0.24±0.25 | 0.01±0.12 | 0.10±0.26 | 0.05±0.13 | 0.12±0.09 | -0.02±0.12 | 0.13±0.22 | 0.06±0.34 |
| | MAK | 0.13±0.27 | 0.82±0.18 | 0.17±0.22 | 0.31±0.37 | 0.41±0.32 | 0.17±0.17 | 0.64±0.20 | 0.10±0.34 | 0.09±0.27 |
| **Claude 2** | CAK | 0.32±0.18 | 0.56±0.23 | 0.52±0.22 | 0.44±0.27 | 0.37±0.25 | 0.44±0.23 | 0.85±0.22 | 0.20±0.34 | 0.26±0.24 |
| | CAD | -0.16±0.27 | -0.07±0.15 | -0.04±0.14 | 0.22±0.18 | 0.16±0.17 | 0.11±0.19 | -0.06±0.17 | 0.14±0.49 | -0.01±0.26 |
| | MAD | 0.08±0.16 | 0.12±0.14 | 0.03±0.20 | 0.02±0.45 | 0.16±0.35 | -0.12±0.22 | -0.09±0.33 | -0.04±0.23 | 0.08±0.47 |
| | MAK | 0.57±0.24 | 0.74±0.20 | 0.59±0.16 | 0.24±0.28 | 0.37±0.41 | 0.21±0.32 | 0.82±0.33 | 0.02±0.35 | 0.35±0.43 |

We consider calculating TDRs in each of the following scenarios to construct our estimations for attribution scores: (1). Raw Data (using the whole original data as in Section 3.2); (2). Omit Knowledge (as in Section 3.3); (3). Omit Data (as in Section 3.4); (4). Random Guess (as in Section 3.5); (5). Reverse (evaluating with the reversed causal graph after causal order reversal, as in Section 3.7); and (6). Reverse-Raw (evaluate the original causal graph after causal graph reversal, as in Section 3.7). For example, TDR (raw data) corresponds to the TDR of LLM predictions using the whole original data as in Section 3.2, which estimates the first term ($\mathbb{P}\left(\widehat{y}_i = y_i \mid do\left(k_i = k_i, d_i = d_i\right), c_i, u_i\right)$) in Definitions 2.1 and 2.2. Following Section 3, based on Definitions 2.1 to 2.4, we can derive the causal attribution as,

$$\text{CAK} = \text{TDR (raw data)} - \text{TDR (no knowledge)}, \quad \text{CAD} = \text{TDR (raw data)} - \text{TDR (no numerical data)},$$
$$\text{MAK} = \text{TDR (no numerical data)} - \text{TDR (guess)}, \quad \text{MAD} = \text{TDR (no knowledge)} - \text{TDR (guess)},$$

where TDR (no knowledge) is the TDR of LLM predictions when omitting knowledge which estimates $\mathbb{P}\left(\widehat{y}_i = y_i \mid do\left(k_i = \emptyset, d_i = d_i\right), c_i, u_i\right)$ in Definition 2.1, TDR (no numerical data) is the TDR of LLM predictions when omitting numerical data which estimates $\mathbb{P}\left(\widehat{y}_i = y_i \mid do\left(k_i = k_i, d_i = \emptyset\right), c_i, u_i\right)$ in Definition 2.2, and TDR (guess) is the TDR of LLM predictions without input data and knowledge which estimates $\mathbb{P}\left(\widehat{y}_i = y_i \mid do\left(k_i = \emptyset, d_i = \emptyset\right), c_i, u_i\right)$ in Definitions 2.3-2.4. These metrics enable us to quantify the LLMs' causal discovery accuracy. Structural Hamming Distance (SHD) is also calculated to compare the difference between the predicted graph and the true graph. We also use the false discovery rate (FDR) as FDR = (# Incorrectly Predicted Causal Pairs)/(# Predicted Causal Pairs), and F1 score that is calculated based on the binary decision of whether the LLM correctly predicts the direction of the directed edge under each scenario, as additional accuracy metrics.

## 4 Analyses of Causal Attribution Model

**Implementation and evaluation metrics.** To ensure a comprehensive evaluation, we employ a suite of general-purpose autoregressive LLMs based on GPT (Radford et al., 2019): GPT-3.5 (ChatGPT), GPT-4, and GPT-4 Turbo (OpenAI, 2023), accessed via the OpenAI API with a zero temperature for consistency. Additionally, we include Claude 2 (Models) and LLaMa2-13B (Touvron et al., 2023) to cover a broad spectrum of model architectures and training backgrounds. Implementation details are provided in Section 3, with specific examples illustrated in Appendix C. The mean and standard deviation of metrics estimated based on Section 3.8 under different LLMs across various datasets are summarized in Table 2 for the proposed attribution scores and Table 3 for the F1 score over 15 replications. Results for TDR, FDR, and SHD are presented in Tables E.1, E.2, and E.3, respectively, in Appendix E, with pairwise causal discovery detailed in Appendix D.

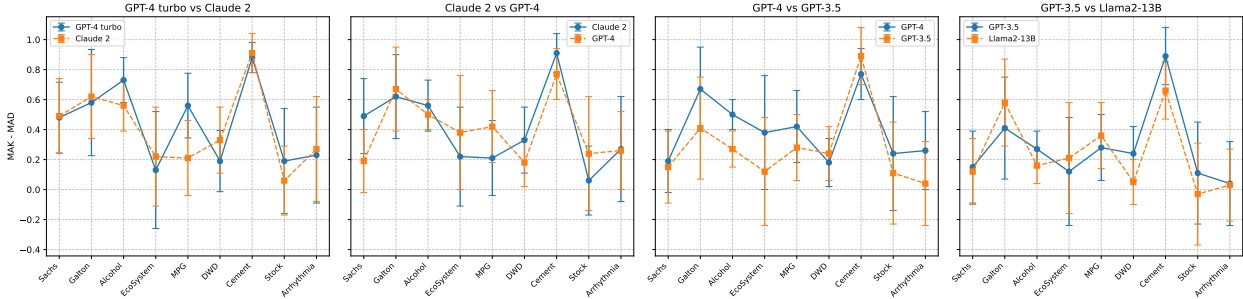

Figure 4: The differences between attribution scores (MAK-MAD) among LLMs to demonstrate a hierarchy in their knowledge depth.

Table 3: The results of F1 scores of LLMs for different datasets.

|  | Method/Dataset | Sachs | Galton | Alcohol | EcoSystem | MPG | DWD | Cement | Stock | Arrhythmia |
|---|---|---|---|---|---|---|---|---|---|---|
| **GPT-4 turbo** | Raw Data | 0.53±0.20 | 1±0 | 1±0 | 0.74±0.23 | 0.62±0.14 | 0.50±0.15 | 0.99±0.04 | 0.30±0.33 | 0.51±0.16 |
|  | Omit Data | 0.47±0.20 | 1±0 | 0.99±0.05 | 0.55±0.23 | 0.72±0.09 | 0.52±0.18 | 0.97±0.08 | 0.42±0.46 | 0.65±0.13 |
|  | Omit Knowledge | 0.09±0.07 | 0.25±0.24 | 0.22±0.15 | 0.33±0.28 | 0.16±0.16 | 0.35±0.20 | 0.12±0.15 | 0.17±0.18 | 0.45±0.28 |
|  | Reverse | 0.28±0.14 | 0.13±0.16 | 0.23±0.16 | 0.16±0.24 | 0.06±0.10 | 0.25±0.20 | 0.64±0.19 | 0.24±0.17 | 0.09±0.26 |
|  | Reverse-Raw | 0.48±0.29 | 1±0 | 0.93±0.13 | 0.83±0.19 | 0.53±0.13 | 0.49±0.20 | 0.97±0.12 | 0.25±0.16 | 0.54±0.20 |
|  | Random Guess | 0.05±0.19 | 0.40±0.49 | 0.28±0.41 | 0.40±0.49 | 0±0 | 0.06±0.12 | 0.73±0.44 | 0.20±0.40 | 0.13±0.34 |
| **GPT-4** | Raw Data | 0.34±0.24 | 1±0 | 0.70±0.25 | 0.70±0.14 | 0.57±0.13 | 0.36±0.09 | 1±0 | 0.30±0 | 0.54±0.11 |
|  | Omit Data | 0.53±0.17 | 1±0 | 0.69±0.20 | 0.66±0.11 | 0.73±0.08 | 0.49±0.12 | 1±0 | 0.40±0.23 | 0.66±0.08 |
|  | Omit Knowledge | 0.33±0.20 | 0.29±0.21 | 0.31±0.11 | 0.43±0.16 | 0.32±0.15 | 0.33±0.10 | 0.21±0.25 | 0.17±0.09 | 0.44±0.13 |
|  | Reverse | 0.26±0.15 | 0.13±0.21 | 0.32±0.09 | 0.16±0.16 | 0.30±0.13 | 0.26±0.12 | 0.72±0.41 | 0.29±0.03 | 0.34±0.11 |
|  | Reverse-Raw | 0.25±0.13 | 1±0 | 0.53±0.14 | 0.67±0.16 | 0.50±0.15 | 0.29±0.09 | 0.95±0.19 | 0.30±0.01 | 0.52±0.15 |
|  | Random Guess | 0.07±0.25 | 0.15±0.36 | 0.13±0.34 | 0.21±0.41 | 0±0 | 0±0 | 0.19±0.39 | 0.14±0.35 | 0±0 |
| **GPT-3.5** | Raw Data | 0.15±0.10 | 0.94±0.15 | 0.64±0.37 | 0.68±0.31 | 0.39±0.19 | 0.29±0.21 | 0.99±0.04 | 0.41±0.26 | 0.28±0.25 |
|  | Omit Data | 0.22±0.17 | 0.96±0.10 | 0.77±0.34 | 0.55±0.23 | 0.41±0.20 | 0.46±0.13 | 0.99±0.03 | 0.42±0.29 | 0.28±0.13 |
|  | Omit Knowledge | 0.11±0.10 | 0.38±0.14 | 0.28±0.15 | 0.36±0.35 | 0.19±0.14 | 0.20±0.13 | 0.09±0.06 | 0.34±0.28 | 0.26±0.22 |
|  | Reverse | 0.12±0.09 | 0.15±0.16 | 0.26±0.13 | 0.11±0.27 | 0.18±0.17 | 0.11±0.13 | 0.14±0.24 | 0.24±0.12 | 0.25±0.19 |
|  | Reverse-Raw | 0.22±0.13 | 0.91±0.16 | 0.60±0.38 | 0.93±0.19 | 0.39±0.23 | 0.31±0.22 | 0.94±0.17 | 0.35±0.20 | 0.35±0.20 |
|  | Random Guess | 0.09±0.09 | 0.32±0.36 | 0.17±0.10 | 0.42±0.48 | 0.18±0.17 | 0.16±0.25 | 0.08±0.07 | 0.25±0.23 | 0.31±0.27 |
| **LLaMa2-13B** | Raw Data | 0.29±0.23 | 0.49±0.33 | 0.21±0.12 | 0.42±0.24 | 0.60±0.26 | 0.45±0.26 | 0.64±0.36 | 0.35±0.18 | 0.39±0.14 |
|  | Omit Data | 0.33±0.17 | 1±0 | 0.33±0.19 | 0.50±0.40 | 0.64±0.22 | 0.28±0.30 | 0.77±0.26 | 0.26±0.25 | 0.33±0.22 |
|  | Omit Knowledge | 0.20±0.11 | 0.42±0.14 | 0.15±0.10 | 0.29±0.12 | 0.26±0.15 | 0.19±0.16 | 0.12±0.10 | 0.29±0.18 | 0.29±0.12 |
|  | Reverse | 0.22±0.17 | 0.18±0.20 | 0.25±0.11 | 0.32±0.08 | 0.25±0.25 | 0.42±0.17 | 0.21±0.22 | 0.23±0.18 | 0.26±0.14 |
|  | Reverse-Raw | 0.27±0.15 | 0.61±0.33 | 0.20±0.10 | 0.47±0.13 | 0.47±0.25 | 0.44±0.20 | 0.69±0.38 | 0.29±0.15 | 0.39±0.16 |
|  | Random Guess | 0.20±0.11 | 0.18±0.22 | 0.15±0.12 | 0.19±0.18 | 0.23±0.15 | 0.11±0.11 | 0.14±0.06 | 0.16±0.16 | 0.23±0.20 |
| **Claude 2** | Raw Data | 0.50±0.16 | 0.93±0.13 | 0.78±0.25 | 0.72±0.24 | 0.64±0.21 | 0.57±0.16 | 0.91±0.13 | 0.41±0.18 | 0.62±0.15 |
|  | Omit Data | 0.67±0.21 | 1±0 | 0.82±0.18 | 0.52±0.19 | 0.50±0.24 | 0.49±0.12 | 1±0 | 0.29±0.38 | 0.66±0.23 |
|  | Omit Knowledge | 0.19±0.11 | 0.35±0.22 | 0.27±0.16 | 0.35±0.22 | 0.28±0.15 | 0.15±0.12 | 0.09±0.14 | 0.22±0.18 | 0.36±0.21 |
|  | Reverse | 0.38±0.08 | 0.02±0.08 | 0.19±0.12 | 0.24±0.30 | 0.16±0.14 | 0.29±0.19 | 0.50±0.34 | 0.26±0.16 | 0.24±0.15 |
|  | Reverse-Raw | 0.55±0.10 | 0.88±0.17 | 0.83±0.18 | 0.81±0.20 | 0.63±0.25 | 0.52±0.12 | 0.93±0.12 | 0.45±0.24 | 0.61±0.15 |
|  | Random Guess | 0.10±0.12 | 0.22±0.22 | 0.25±0.08 | 0.13±0.19 | 0.12±0.17 | 0.27±0.22 | 0.18±0.33 | 0.27±0.30 | 0.32±0.33 |

**Results of causal attribution model on LLMs' capabilities in causal discovery.** Firstly, the high MAK scores in Table 2 highlight the crucial role of knowledge alone in enabling LLMs to derive causal relationships across various datasets and models. Furthermore, the high CAK scores in Table 2 reveal that even with numerical data, variable names significantly enhance LLMs' causal discovery accuracy by leveraging their internal knowledge. This suggests that an LLM's performance in complex causal analysis *heavily depends on the extent and sophistication of its pre-existing knowledge base.* Comparative analysis of attribution scores among different LLMs demonstrates *a hierarchy in knowledge depth.* Specifically, as shown in Figure 4, excluding the second pair, the other three pairs demonstrate that one LLM consistently outperforms the other across almost all datasets, with at most a slight disadvantage in one dataset. Conversely, the MAD and CAD scores are relatively low. We further conduct the Mann-Whitney U test with an asymptotic approximation on the Sachs dataset for 20 iterations under the random guess and numerical data-only conditions. The resulting p-value of 0.05, though close to the threshold, indicates a statistically significant difference, which lends to support that LLMs still display *limited but existent causal discovery abilities based on numerical data.* These findings are consistent with Tables E.1, E.2, and E.3 in Appendix. In addition, the results in Table 3, where 'omitted data' setups often perform as well or better than 'raw data' setups, suggest that the LLMs may not be directly performing complex statistical operations. Instead, these results may indicate that providing preprocessed statistical information, rather than relying on the LLMs' native capabilities, is a more effective method for boosting performance.

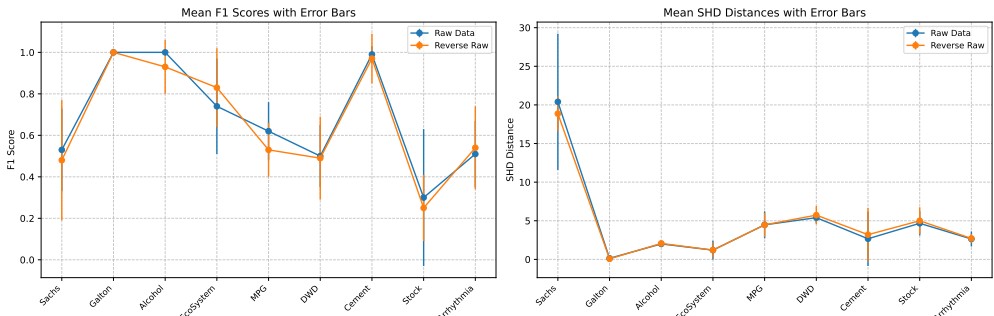

Figure 5: Performance of GPT-4 turbo under Raw Data versus Reverse-Raw.

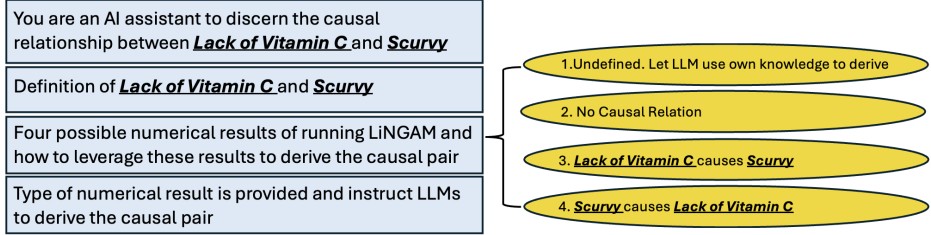

Figure 6: Illustration of the instruction of the generated dataset for fine-tuning LLMs. The blue boxes represent sequential prompts and steps, while the yellow boxes are used to highlight four possible causal relations among two variables.

**The results from the reverse causal discovery experiment** reveal that altering the causal topological orders of variables does not significantly impact LLMs' causal discovery outcomes, as shown in Figure 5. More specifically, the numerical information that indicates the true data-generating process has been largely ignored by LLMs and the variable names are the main sources they use for judgments. This finding further supports the priority of the knowledge component in guiding LLMs' causal discovery processes even if the results are contradictory to the truth. More analyses of causal discovery accuracy under different LLMs can be found in Appendix E.

## 5 Precise Fine-tune for Causal Discovery

As witnessed in our experiments based on the proposed attribution score, the current LLMs' effectiveness in causal discovery heavily relies on provided context and domain-specific knowledge but can also utilize certain numerical data though not in a right way. To further understand how LLMs infer causal relationships, we expand our prompting repertoire to include advanced techniques such as Chain of Thought (CoT) prompting (Wei et al., 2022) in a zero-shot format, to encourage LLMs "think step by step" and provide a sequential and transparent reasoning path. A notable observation is *the misuse of numerical data by many LLMs*. For example, when knowledge is omitted and CoT is used, the LLM conducts a correlation analysis first and then outputs the causal relation *purely based on the correlation* without performing a conditional independence test (see Figure F.1 and more details for the CoT of LLMs in causal discovery in Appendix F). However, **correlation is often not causation** (Pearl, 2009b;a). This motivates us to consider precisely fine-tuning the way LLMs' utilizing numerical data for causal discovery. More specifically, we would like to answer the following question: after the fine-tuning, can the LLM learn how to *correctly utilize the numerical information provided in the input to assist causal discovery inquiries besides the knowledge they already know?* We detail the precision fine-tuning procedure with a focus on pairwise causal discovery as an illustration.

**Task formulation.** Our dataset, denoted as $\mathcal{D} := \{(q_i, a_i)\}_{i=1}^{N}$, consists of $N$ triplets. Each triplet comprises $q_i$, an instruction containing two variables of interest with their definitions and the numerical results from causal discovery algorithms (e.g., Shimizu et al., 2006), and $a_i$, the correct directed causal pair. Our primary objective is to assess the accuracy of causal discovery by fine-tuning LLMs to incorporate numerical reasoning capabilities.

Table 4: Accuracy of LLMs in the pairwise causal discovery analyses.

| Model | Accuracy | Scenario 1 | Scenario 2 | Scenario 3 | Scenario 4 |
|---|---|---|---|---|---|
| GPT-4 turbo | 0.82 | 0.66 | 0.98 | 0.87 | 0.75 |
| GPT-4 | 0.76 | 0.60 | 1 | 0.98 | 0.44 |
| GPT- 3.5 | 0.58 | 0.03 | 1 | 0.96 | 0.33 |
| LLaMa2-13B | 0.74 | 0.08 | 1 | 1 | 0.86 |
| Claude 2 | 0.74 | 0.36 | 0.99 | 0.84 | 0.78 |
| Mistral-7B | 0.75 | 0.38 | 0.94 | 0.92 | 0.77 |
| Finetuned | 0.90 | 0.58 | 1 | 1 | 1 |

**Design principles.** To generate a fine-tuning dataset, we consider all possible outcomes by executing LiNGAM (Shimizu et al., 2006). Specifically, in the context of two variables ($X$ and $Y$), within the linear non-Gaussian noise setting, LiNGAM initially regresses $X$ against $Y$, followed by regressing $Y$ against $X$ and checking if the residual is correlated with the covariates in these two cases. This results in four scenarios (see Figure 6): (1) If the residual is correlated with the covariate in both cases, the causal direction between $X$ and $Y$ is *undefined*; (2) If the residual is not correlated with the covariate in both cases, then *X and Y do not have a causal relation*; (3) If regressing $Y$ on $X$ and the residual is uncorrelated with $X$ but the reverse does not hold, then *X causes Y*; (4) If regressing $Y$ on $X$ and the residual is correlated with $X$ but the reverse does not hold, then *Y causes X*.

**Data generation pipeline.** We generate four sets of data for each causal pair, corresponding to the outlined four scenarios. These sets share certain instruction components, such as variable names, definitions, and scenario descriptions. Additionally, for scenario (1), we instruct the LLM that *if the causal relationship is undefined, it should utilize its knowledge to infer the causal pairs*. This highlights the higher priority of numerical data in our proposed fine-tuned model. Each dataset contains unique numerical outcomes from LiNGAM analyses, matching the described scenarios (see Figure 6). The identified ground-truth pairs align with LiNGAM results. During fine-tuning, we randomize variable order when introducing names and definitions to prevent LLM overfitting and promote broader information consideration. We compile 300 causal pairs, generating 1200 training samples and an additional 106 pairs producing 424 testing samples. The test samples contain more challenging pairs extracted from Sachs data.

**Selected LLM for fine-tuning.** Adopting Mistral-7B-v0.2 (Jiang et al., 2023) as the LLM backbone, we run instruction fine-tuning with LoRA (Hu et al., 2022) with rank 8 and alpha 32 to perform parameter-efficient tuning and store adapter weights. We use *AdamW* as the optimizer. We conduct 1 epoch of fine-tuning using one *ml.p3dn.24xlarge* instance from SageMaker, taking 33 minutes.

**Main evaluation on the fine-tuned model.** We consider baselines using closed-source LLMs such as GPT-4 turbo, GPT-4, GPT-3.5, and Claude 2, as well as open-source LLMs such as LLaMa2-13B and Mistral-7B-v0.2 in a zero-shot manner with the provided instructions in the generated data. We evaluate our model not only in overall accuracy but also in the accuracy of the four scenarios we construct in Section 5. The results are summarized in Table 4. There are four key points. First, our fine-tuned model achieves the best overall accuracy. Notably, it performs well in scenarios 2 to 4, adhering strictly to numerical reasoning results even when encountering counter-intuitive hypotheses. Second, in scenario 1, Mistral-7B-v0.2 shows slightly lower performance compared to GPT-4 and GPT-4 turbo. This is expected as scenario 1 requires language models to utilize their own knowledge for causal discovery, and GPT-4 (turbo) excels in this aspect due to its superior performance on common knowledge benchmarks. Surprisingly, our findings reveal that after fine-tuning, Mistral-7B-v0.2 outperforms the zero-shot approach, primarily due to training on the output format. Third, LLaMa2-13B and GPT-3.5 struggle with scenario 1. They find it hard to grasp the instructions for both scenarios 1 and 2, often responding with *No Causal Relation* for scenario 1 without leveraging their own knowledge. Another interesting key finding is that GPT-3.5 and GPT-4 struggle with scenario 4, as they recognize the counter-intuitive causal relationship and prefer their own knowledge over numerical reasoning.

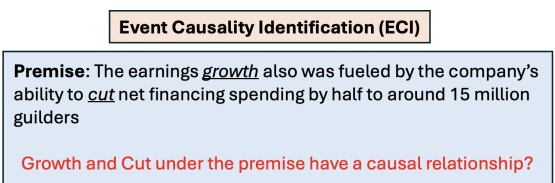

Figure 7: Illustration of knowledge-based pairwise causal discovery (PCD) tasks with one and two hypotheses.

**Event Causality Identification (ECI)**

**Premise:** The earnings *growth* also was fueled by the company's ability to *cut* net financing spending by half to around 15 million guilders

Growth and Cut under the premise have a causal relationship?

Figure 8: Illustration of causality event identification (CEI) task.

Table 5: Accuracy of the finetuned Mixtral v0.2 and the original Mixtral v0.2 on three causal-related tasks, including knowledge-based PCD with one hypothesis and two hypotheses, and CEI.

| Datasets | PCD (One Hypo) | PCD (Two Hypo) | ECI |
|---|---|---|---|
| Finetuned Model | 0.68 | 0.76 | 0.62 |
| Original Model | 0.68 | 0.76 | 0.62 |

**Performance in other tasks and generalizability.** We conduct additional experiments regarding the generalizability of our finetuned model in other causal tasks and show its robustness to catastrophic forgetting. To be specific, we consider evaluating the performance of the finetuned model in knowledge-based pairwise causal discovery and causality event identification tasks studied by Chen et al. (2024). The detailed data structure is provided in Appendix G with flowcharts of tasks in Figure 7 for PCD and Figure 8 for ECI, respectively. By applying the proposed finetuned model to these additional tasks, as shown in Table 5, we can conclude that the finetuned model does not lose generalization for contextual causal-related tasks and there is no catastrophic forgetting.

## 6 Related Works, Limitations, and Conclusion

**Causal reasoning in large language models** is an emerging area focused on understanding cause-effect relationships within text. While LLMs like OpenAI's GPT series excel in generating coherent and contextually relevant text, their ability to parse and apply causal reasoning has been less explored (Weidinger et al., 2021). Recent studies (Riedel et al., 2019; Keith et al., 2020) have started addressing this by integrating causal inference mechanisms into the models' architecture, such as enhancing text generation with causal structure and incorporating latent causal variables during pre-training. In this work, we focus on disentangling the causal reasoning abilities of advanced LLMs, which has garnered considerable interest due to recent breakthroughs (e.g., Vaswani et al., 2017; Kenton & Toutanova, 2019; Lewis et al., 2019; Brown et al., 2020; Neelakantan et al., 2022; Stiennon et al., 2020; OpenAI, 2023). Instead of pre-training LLMs with more causal-related texts, we propose a novel attribution approach to enhance the understanding of how LLMs use their knowledge and data for causal reasoning, offering a more effective fine-tuning method.

Over the past a few decades, **attribution models** have gained prominence for their role in fairly allocating contributions across features in predictive modeling. The Shapley value (Roth, 1988), emerging from cooperative game theory, offers a principled method to apportion payoffs by considering the marginal contribution of each feature within all possible combinations of feature subsets. Recently, Heskes et al. (2020) proposed a new framework to compute causal Shapley values using Pearl's do-calculus to distinguish

direct and indirect effects. Janzing et al. (2020) further clarified the distinction between observational and interventional probabilities in Shapley value calculations based on Pearl's causality principles. Adapted to machine learning, this approach aids in gauging feature importance in complex models, including LLMs (Lundberg & Lee, 2017). Techniques like saliency mapping and influence functions highlight relevant data and trace predictions back to training instances, revealing how LLMs handle complex inputs to produce coherent outputs (Simonyan et al., 2013; Koh & Liang, 2017). Furthermore, visual interpretation methods like Layer-wise Relevance Propagation (LRP) and Integrated Gradients have been crucial for evaluating individual input contributions to outputs, thereby enhancing the transparency and trustworthiness of LLMs (Bach et al., 2015; Sundararajan et al., 2017). Nevertheless, the intricacies of LLMs' internal representations, which are often intricate and intertwined, present substantial challenges in clear attribution. Moreover, the inherent computational demand still requires the use of approximation techniques for LLMs with numerous features (Strumbelj & Kononenko, 2010; Datta et al., 2016; Kumar et al., 2020).

**The need for causal attribution**—to identify and quantify the influence of input factors on outputs—becomes more critical, as the complexity of LLMs increases. This process illuminates the model's decision-making and enhances the transparency and justifiability of its predictions (Chattopadhyay et al., 2019). Studies indicate that attention mechanisms, although commonly employed for interpretability, may not reliably indicate the reasoning behind a model's decisions (Doshi-Velez & Kim, 2017; Jain & Wallace, 2019), highlighting the need for more sophisticated attribution methods that can capture the nuances of large-scale neural network decisions. Causality-centric frameworks within LLMs are crucial for ensuring model fidelity in practical scenarios by understanding the 'why' behind outputs, making them reliable and actionable (Wachter et al., 2017; Goyal et al., 2019). Methods such as counterfactual explanations and structural causal models are key in unraveling the interplay between input features and model predictions, vital for diagnosing failures and strengthening model robustness (Ribeiro et al., 2016; Goyal et al., 2019). Yet, the attribution model for LLMs in causal inference tasks is still less studied challenging due to their non-linear, high-dimensional, and opaque nature (Kim et al., 2017; Moraffah et al., 2020). Our research contributes to this field by proposing a novel causal attribution model that aligns with the experimental design, aiming to bridge gaps in current methodologies and create interpretable and ethical LLMs.

Our proposed causal attribution model can be easily extended to handle various attribution tasks to disentangle the black box and provide explainability for model performance. Our evaluation reveals that not only that knowledge is current LLMs mainly used for causal reasoning, but also, in the absence of such knowledge, LLMs can *still maintain a degree of causal discovery using the available numerical data*, albeit with limited calculations in correlation instead of causation. These observations, to the best of our knowledge, have not been studied and verified systematically yet. By utilizing these new insights, our precisely fine-tuned model achieves the highest accuracy in identifying the true causal relationship by mastering both inherent knowledge and instructed logic of using numerical data, thus filling the gap effectively. Furthermore, the proposed evaluation framework is general and applicable to a broader range of scenarios in causal inference and machine learning tasks such as causal effect estimation, correlation prediction, and mathematical reasoning.

We acknowledge several limitations and consider the following future works. First, our approach relies on current causal discovery benchmarks, potentially limiting the generalizability of our findings, and thus more extensive and diverse benchmarks are needed. Although our fine-tuned model effectively uses both numerical data and embedded knowledge for causal discovery, the integration process prioritizes numerical data. In practical applications, there may exist finite-sample issues or possible data missingness in the real applications where internal knowledge of LLMs is more reliable and helpful. For instance, the existing knowledge could generate valid constraints on causal discovery and help to identify a more precise causal graph. The proposed fine-tuning process can be extended to knowledge-enhanced causal discovery. To better balance these two inputs, one promising method involves utilizing uncertainty scores as a mechanism to dynamically adjust the influence of numerical data versus internal knowledge. Specifically, these scores can be derived from the model's predictions, reflecting the confidence level in each input type. By applying a weighted scheme based on these scores, we can fine-tune the model in a way that optimally leverages both sources of information, potentially leading to more accurate and robust causal inferences. Further research could explore various weighting algorithms to refine this approach, ensuring that the model can adaptively adjust its reliance on numerical data or internal knowledge based on the task complexity and the available data quality.

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

## A    More Related Works

**Causal reasoning in large language models** is an emerging area of study that seeks to endow these models with the ability to understand cause-effect relationships within text. While large language models like OpenAI's GPT series have shown remarkable performance in generating coherent and contextually relevant text, their ability to parse and apply causal reasoning has been less explored (Weidinger et al., 2021). Recent studies, however, have begun to address this by integrating causal inference mechanisms into the models' architecture. For instance, Keith et al. (2020) proposed a method for enhancing text generation with causal structure, showing that it can improve a model's reasoning capabilities. Riedel et al. (2019) introduced latent causal variables into the training of language models, which helps in disentangling the underlying causal factors from observed data. Such enhancements are believed to make language models not only better at language understanding and generation but also at more sophisticated tasks like summarization, question answering, and decision-making that require causal reasoning (Pearl & Mackenzie, 2019). Hence, understanding the ability of causal reasoning in large language models is extremely important. In this work, we focus on disentangling the causal inference abilities of advanced LLMs, which has piqued considerable interest in light of recent breakthroughs (e.g., Vaswani et al., 2017; Kenton & Toutanova, 2019; Lewis et al., 2019; Brown et al., 2020; Stiennon et al., 2020; Neelakantan et al., 2022; OpenAI, 2023).

**Attribution models** have gained prominence for their role in fairly allocating contributions across features in predictive modeling. The Shapley value (Roth, 1988), emerging from cooperative game theory, offers a principled method to apportion payoffs by considering the marginal contribution of each feature within all possible combinations of feature subsets. Adapted to machine learning, this approach aids in gauging feature importance in complex models, including LLMs (Lundberg & Lee, 2017). However, the inherent computational demand still requires the use of approximation techniques for LLMs with numerous features (Strumbelj & Kononenko, 2010; Datta et al., 2016; Kumar et al., 2020). Furthermore, visual interpretation methods like Layer-wise Relevance Propagation (LRP) and Integrated Gradients have been crucial for evaluating individual input contributions to outputs, thereby enhancing the transparency and trustworthiness of LLMs (Bach et al., 2015; Sundararajan et al., 2017). Nevertheless, the intricacies of LLMs' internal representations, which are often intricate and intertwined, present substantial challenges in clear attribution. Studies indicate that attention mechanisms, although commonly employed for interpretability, may not reliably indicate the reasoning behind a model's decisions (Doshi-Velez & Kim, 2017; Jain & Wallace, 2019), highlighting the need for more sophisticated attribution methods that can capture the nuances of large-scale neural network decisions.

**Causal inference in LLMs.** As the complexity of LLMs increases, the critical need for causal attribution—to identify and quantify the influence of input factors on outputs—becomes more pronounced. This process not only sheds light on the model's decision-making but also bolsters transparency and justifiability of model predictions (Chattopadhyay et al., 2019). Techniques like saliency mapping and influence functions are instrumental in highlighting relevant data and tracing predictions back to training instances, thereby revealing how LLMs handle complex inputs to produce coherent outputs (Simonyan et al., 2013; Koh & Liang, 2017). The advent of causality-centric frameworks within LLMs is crucial for assuring model fidelity in practical scenarios by understanding the 'why' behind outputs, making outputs reliable and actionable (Wachter et al., 2017; Goyal et al., 2019). Furthermore, methods such as counterfactual explanations and structural causal models have been key in unraveling the interplay between input features and model predictions, which is vital for diagnosing failures and fortifying model robustness (Goyal et al., 2019; Ribeiro et al., 2016). However, integrating causal reasoning with LLMs is still an evolving and challenging field due to the models' non-linear, high-dimensional, and opaque nature (Moraffah et al., 2020; Kim et al., 2017). Our research contributes to this field by proposing a novel causal attribution model that aligns with experimental design, aiming to bridge the gap in current methodologies and facilitate the creation of interpretable and ethical LLMs.

The literature on **causal discovery** can be broadly categorized into three classes. The first class of methods focuses on using local conditional independence tests to identify the causal skeleton and determine the direction of edges, such as the PC algorithm (Spirtes et al., 2000a; Kalisch & Bühlmann, 2007). The second class of methods uses functional causal models with additional assumptions about the data distribution, including the ICA-LiNGAM (Shimizu et al., 2006) and the causal additive model (Bühlmann et al., 2014).

Table B.1: Comparison studies among different causal discovery methods (PC, FCI, and DirectLiNGAM) under nine benchmark datasets. Methods are evaluated by FDR, TDR, SHD, and F1.

| Method | Dataset | Sachs | Galton | Alcohol | EcoSystem | MPG | DWD | Cement | Stock | Arrhythmia |
|---|---|---|---|---|---|---|---|---|---|---|
| **PC** | TDR | 0.28 | 1 | 0.20 | 0 | 0 | 0.17 | 0.25 | 0 | 0 |
| | FDR | 0.55 | 0 | 0.83 | 0 | 1 | 0.80 | 0.88 | 0 | 0 |
| | SHD | 31 | 0 | 7 | 3 | 8 | 7 | 19 | 3 | 3 |
| | F1 | 0.35 | 1 | 0.18 | 0 | 0 | 0.18 | 0.16 | 0 | 0 |
| **FCI** | TDR | 0.22 | 0 | 0.20 | 0 | 0.33 | 0.50 | 0.25 | 0 | 0 |
| | FDR | 0.67 | 0 | 0.75 | 0 | 0.71 | 0.57 | 0.92 | 0 | 0 |
| | SHD | 31 | 3 | 7 | 3 | 8 | 6 | 20 | 3 | 3 |
| | F1 | 0.26 | 0 | 0.22 | 0 | 0.31 | 0.46 | 0.12 | 0 | 0 |
| **DirectLiNGAM** | TDR | 0.50 | 1 | 0.20 | 0.33 | 0.33 | 0.33 | 0 | 0.33 | 0 |
| | FDR | 0.41 | 0 | 0.80 | 0.75 | 0.75 | 0.80 | 1 | 0.89 | 1 |
| | SHD | 17 | 0 | 7 | 5 | 7 | 8 | 35 | 8 | 4 |
| | F1 | 0.54 | 1 | 0.2 | 0.28 | 0.28 | 0.25 | 0 | 0.16 | 0 |

The third class, score-based methods, includes the greedy equivalence search (Chickering, 2002; Ramsey et al., 2017) and optimization methods with acyclicity (Zheng et al., 2018). Refer to (Yu et al., 2019; Zhu et al., 2020; Lachapelle et al., 2020; Cai et al., 2020; Zheng et al., 2020; Vowels et al., 2021) for additional cutting-edge causal structural learning methods. In this work, in contrast, we utilize the LLMs directly to solve the causal discovery problem, and compare its performance with classical algorithms relying on numerical data only, as detailed in our pairwise causal discovery task.

## B  More Details of Dataset Construction for Causal Reasoning

Specifically, the Galton dataset provides historical height measurements within families, enabling studies on hereditary influence. The Sachs dataset includes data on 11 phosphorylated proteins and phospholipids from individual primary immune system cells, offering insights into cellular behavior under various experimental conditions. Data on the health impacts of alcohol, drawn from blood tests and consumption patterns, make up the Alcohol dataset. The Ecosystem dataset comprises carbon flux measurements alongside environmental light conditions, offering a complex interplay of ecological factors. The MPG dataset reflects automotive efficiency, correlating car attributes with fuel consumption and performance. The DWD dataset combines geographical and climatological variables, such as altitude and temperature, to model environmental effects. The Cement dataset relates material composition to structural strength, furnishing a Concrete example of causal reasoning in material science. The Stock dataset consists of stock return data for various pairs of companies, such as Hang Seng Bank, allowing for a comprehensive analysis of inter-company stock performance correlations. Lastly, the Arrhythmia dataset comprises variables associated with Cardiac Arrhythmia.

## C  Implementation Details

We provide the implementation details for probing the causal inference, by using GPT-4 on the Sachs dataset as an example. The following steps outline the specifics of the procedure:

● *Initialization*: We initiate GPT-4 with a clear operational mandate by setting up the system with an unambiguous instruction that defines its role: *You are a helpful assistant to suggest potential causal pairs with direction (A → B means A causes B).* This precise configuration is crucial to focus the model's capabilities on generating causal relationships with direction from the provided data.

● *Prompt Generation*: We first randomize the order of the Sachs dataset variables to mitigate any ordering bias that may influence the LLM's output. The prompt is then constructed to direct the model's attention strictly to the task at hand: *Suggest causal pairs with direction among following variables after analyzing following data: {raw data} MUST Suggest ONLY the causal pairs with direction without saying any other things.* For the pairwise causal discovery experiment, we sample the relation from the ground truth relations and generate the corresponding simulated data to construct the prompt, as illustrated by Figure D.1.

• *Prompt Tailoring*: In line with the experimental design, we tailor prompts to account for different perturbations—removing data values, obscuring knowledge with placeholder terms, or conducting reverse causal discovery and pairwise causal discovery. Each variation is designed to test a specific aspect of the LLM's causal reasoning under altered input conditions (Figures 2, 3, and D.1 illustrate the prompt adjustments for each scenario).

• *Evaluation*: We then extract the causal pairs predicted by GPT-4 in response to these prompts and proceed to the evaluation phase. Here, we compute the evaluation metrics by contrasting the LLM-generated causal pairs with the established ground truth of the dataset. During reverse causal discovery trials, we analyze two sets of outcomes: one that aligns with the predictions for the reversed causal sequence and another set that compares against the original causal graph.

## D   More Details and Results of Pairwise Causal Discovery Task

The evaluation for pairwise causal discovery considers the following three cases: (1) employing LiNGAM directly on simulated data; (2) formulating the prompt using data accompanied by variable names; (3) constructing the prompt with data along with variable names, complemented by simplified instructions outlining the procedure for executing LiNGAM. In these three cases, accuracy serves as the performance metric for evaluation over 20 replications, and the result is summarized in Table D.1.

The findings derived from the pairwise causal discovery analysis presented in Table D.1 indicate that within the context of the perfect causal relation simulation setting, LiNGAM consistently achieves accurate predictions of the relationship, whereas LLMs fall short in comparison to the numerical method, even after incorporating the simplified instructions from LiNGAM.

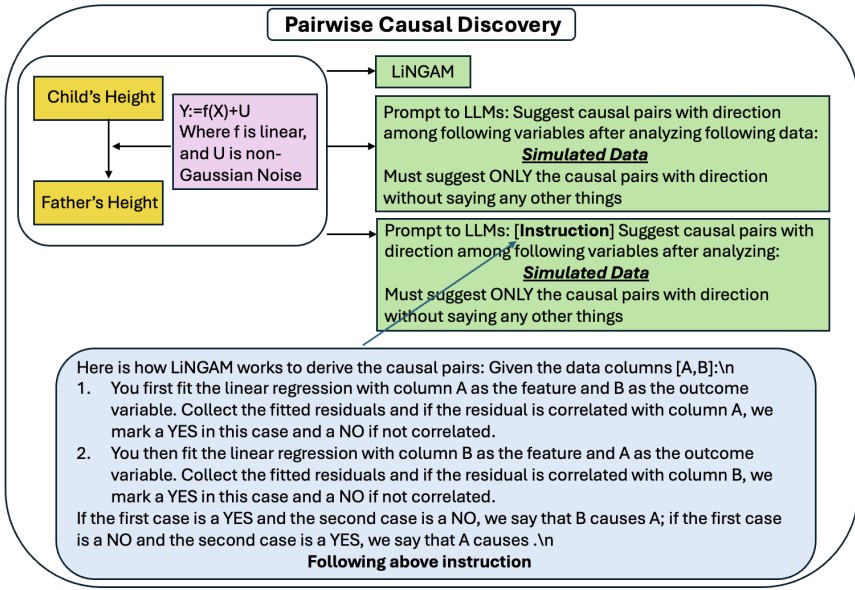

Figure D.1: Illustration of the experiment design of the pairwise causal discovery task.

## E   Additional Results of Accuracy

For the F1 score in the context of raw data (Figure E.1), GPT-4 Turbo achieves the highest or comparable F1 scores across the nine datasets. Claude 2 performs similarly to GPT-4 and slightly outperforms GPT-3.5. Yet, LlaMa2-13B exhibits a performance gap compared to GPT-3.5 across most datasets. Despite performance differences, these models exhibit consistent behavior across datasets: if one model performs poorly on a dataset, others also tend to perform poorly on the same dataset. For SHD (Figure E.2), GPT-4 Turbo

Table D.1: Accuracy of LLMs for different datasets in the pairwise causal discovery analyses.

| Method/Dataset | Sachs | Galton | Alcohol | EcoSystem | MPG | DWD | Cement | Stock | Arrhythmia |
|---|---|---|---|---|---|---|---|---|---|
| LiNGAM | 1 | 1 | 1 | 1 | 1 | 1 | 1 | 1 | 1 |
| GPT-4 turbo | 0.35 | 0.20 | 0 | 0 | 0.10 | 0.05 | 0 | 0.50 | 0.25 |
| GPT-4 turbo (educated) | 0.40 | 0.20 | 0.10 | 0.10 | 0.25 | 0.30 | 0.05 | 0.20 | 0.25 |
| GPT-4 | 0.45 | 0.30 | 0.55 | 0.40 | 0.30 | 0.60 | 0.30 | 0.45 | 0.25 |
| GPT-4 (educated) | 0.60 | 0.15 | 0.60 | 0.25 | 0.30 | 0.60 | 0.15 | 0.35 | 0.40 |
| GPT- 3.5 | 0.40 | 0.70 | 0.40 | 0.45 | 0.50 | 0.60 | 0.40 | 0.5 | 0.25 |
| GPT-3.5 (educated) | 0.40 | 0.55 | 0.05 | 0.45 | 0.55 | 0.45 | 0.05 | 0.5 | 0.45 |
| LLaMa2-13B | 0.50 | 0.15 | 0.25 | 0.40 | 0.45 | 0.40 | 0.20 | 0.75 | 0 |
| LLaMa2-13B (educated) | 0.10 | 0 | 0.10 | 0 | 0.25 | 0 | 0 | 0.50 | 0 |
| Claude 2 | 0.45 | 0 | 0 | 0 | 0.15 | 0 | 0.15 | 0.30 | 0.45 |
| Claude 2 (educated) | 0.5 | 0.05 | 0.20 | 0.20 | 0.05 | 0.30 | 0.25 | 0.60 | 0.45 |

achieves the lowest or comparable SHD across the datasets. GPT-3.5 performs slightly worse than Claude 2 but is similar to GPT-4 and LlaMa2-13B.

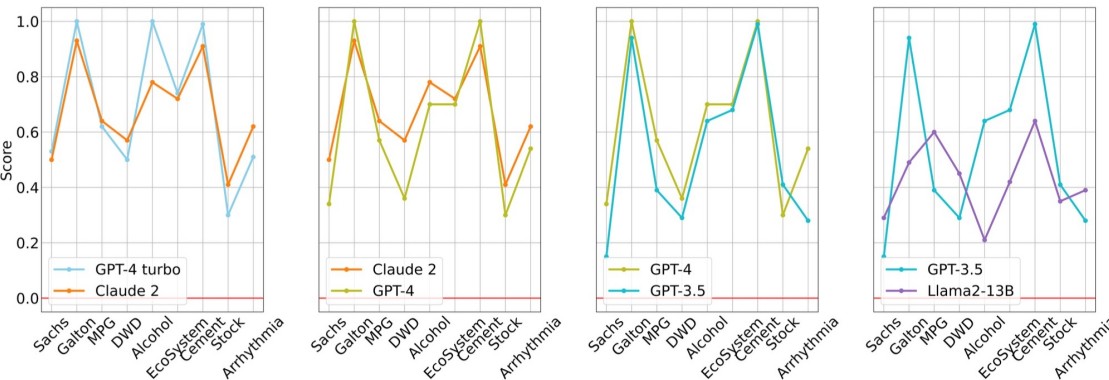

Figure E.1: The comparisons of F1 scores among LLMs, including GPT-4 turbo, GPT-4, GPT-3.5, Claude 2, and LLaMa2-13B.

## F Additional Supporting Analyses

We provide more supporting analyses to deep dive into the causal reasoning process of LLMs. First, there is the impact of variable order observed influencing the causal reasoning in LLMs. Particularly, in the case of ChatGPT, our experiments reveal that the order in which variables are presented affects the model's causal reasoning accuracy. Use the chain of thought (CoT) with the zero-shot prompt for deep-dive. In the case of omitting knowledge, GPT4/3.5, in most cases, predicts the causal pairs following column orders. For example, if the column is ordered as *bryoto, nienet, feouni, alphan*, it outputs the pairs like following *bryoto → nienet, nienet → feouni, feouni → alphan* or *bryoto → nienet, feouni → alphan*. This finding points to an inherent bias in the model's processing mechanism, likely stemming from its training phase, and highlights the importance of considering variable sequencing in the prompt design for causal inference tasks.

Moreover, **the correlation analysis result is often not causation** (see Figure F.1 for the CoT of LLMs in causal discovery). The inference that the LLM engages in correlation analysis is drawn from the observed outputs displayed in Figure F.1, where the LLMs were instructed to use the Chain of Thought (CoT) approach for causal discovery with both data and knowledge inputs provided. Specifically, the correct causal pair is *feouni → nienet*, with an increase in *feouni* causing an increase in *nienet*. Yet, the LLM, such as GPT-4, incorrectly outputs *nienet → feouni* due to the order of *nienet* preceding *feouni*. In Figure F.1, the LLMs' responses typically begin with statements that explicitly analyze or mention correlations between variables, indicating that they assess the strength and direction of relationships as an initial step. Following this analysis,

Table E.1: The results of true discovery rates of LLMs for different datasets.

| Method/Dataset | Sachs | Galton | Alcohol | EcoSystem | MPG | DWD | Cement | Stock | Arrhythmia |
|---|---|---|---|---|---|---|---|---|---|
| **GPT-4 turbo** | | | | | | | | | |
| Raw Data | 0.58±0.26 | 1±0 | 1±0 | 0.76±0.24 | 0.68±0.19 | 0.52±0.16 | 0.99±0.04 | 0.32±0.33 | 0.62±0.27 |
| Omit Data | 0.57±0.28 | 1±0 | 0.99±0.05 | 0.56±0.24 | 0.72±0.12 | 0.53±0.17 | 0.99±0.04 | 0.44±0.46 | 0.76±0.23 |
| Omit Knowledge | 0.09±0.07 | 0.42±0.44 | 0.25±0.24 | 0.43±0.39 | 0.16±0.16 | 0.35±0.20 | 0.12±0.15 | 0.25±0.27 | 0.53±0.35 |
| Reverse | 0.22±0.13 | 0.13±0.16 | 0.23±0.16 | 0.20±0.30 | 0.06±0.10 | 0.24±0.20 | 0.65±0.20 | 0.25±0.18 | 0.09±0.26 |
| Reverse-Raw | 0.49±0.28 | 1±0 | 0.95±0.09 | 0.84±0.18 | 0.58±0.13 | 0.46±0.12 | 0.97±0.12 | 0.19±0.19 | 0.62±0.27 |
| Random Guess | 0.05±0.19 | 0.40±0.49 | 0.28±0.41 | 0.40±0.49 | 0±0 | 0.06±0.12 | 0.73±0.44 | 0.20±0.40 | 0.13±0.34 |
| **GPT-4** | | | | | | | | | |
| Raw Data | 0.34±0.24 | 1±0 | 1±0 | 0.91±0.17 | 0.60±0.14 | 0.40±0.11 | 1±0 | 0.30±0 | 0.60±0.19 |
| Omit Data | 0.52±0.17 | 1±0 | 0.92±0.17 | 0.89±0.22 | 0.74±0.07 | 0.51±0.12 | 1±0 | 0.42±0.23 | 0.70±0.13 |
| Omit Knowledge | 0.33±0.20 | 0.33±0.28 | 0.42±0.17 | 0.50±0.28 | 0.32±0.15 | 0.33±0.10 | 0.21±0.14 | 0.18±0.10 | 0.44±0.13 |
| Reverse | 0.30±0.16 | 0.13±0.21 | 0.32±0.09 | 0.16±0.16 | 0.30±0.13 | 0.26±0.12 | 0.72±0.41 | 0.30±0.03 | 0.36±0.18 |
| Reverse-Raw | 0.25±0.13 | 1±0 | 1±0 | 1±0 | 0.50±0.15 | 0.31±0.11 | 1±0 | 0.30±0.01 | 0.61±0.24 |
| Random Guess | 0.07±0.25 | 0.15±0.36 | 0.13±0.34 | 0.21±0.41 | 0±0 | 0±0 | 0.19±0.39 | 0.14±0.35 | 0±0 |
| **GPT-3.5** | | | | | | | | | |
| Raw Data | 0.33±0.38 | 0.96±0.13 | 0.64±0.37 | 0.68±0.31 | 0.39±0.19 | 0.30±0.20 | 0.99±0.04 | 0.43±0.30 | 0.41±0.40 |
| Omit Data | 0.29±0.29 | 0.96±0.10 | 0.78±0.34 | 0.56±0.23 | 0.48±0.29 | 0.46±0.13 | 0.99±0.03 | 0.46±0.32 | 0.27±0.13 |
| Omit Knowledge | 0.13±0.11 | 0.55±0.29 | 0.51±0.41 | 0.43±0.43 | 0.20±0.15 | 0.21±0.15 | 0.11±0.07 | 0.36±0.32 | 0.22±0.28 |
| Reverse | 0.24±0.30 | 0.15±0.16 | 0.26±0.13 | 0.13±0.34 | 0.20±0.19 | 0.14±0.18 | 0.14±0.24 | 0.25±0.13 | 0.26±0.19 |
| Reverse-Raw | 0.40±0.35 | 1±0 | 0.65±0.39 | 0.93±0.18 | 0.41±0.25 | 0.33±0.23 | 0.95±0.13 | 0.37±0.21 | 0.41±0.31 |
| Random Guess | 0.09±0.09 | 0.15±0.36 | 0.17±0.10 | 0.42±0.48 | 0.18±0.17 | 0.16±0.25 | 0.08±0.07 | 0.29±0.30 | 0.21±0.27 |
| **LLaMa2-13B** | | | | | | | | | |
| Raw Data | 0.29±0.23 | 0.52±0.33 | 0.21±0.12 | 0.43±0.28 | 0.60±0.26 | 0.46±0.27 | 0.63±0.36 | 0.38±0.21 | 0.40±0.16 |
| Omit Data | 0.32±0.17 | 1±0 | 0.32±0.20 | 0.50±0.40 | 0.64±0.22 | 0.28±0.30 | 0.78±0.26 | 0.26±0.25 | 0.32±0.22 |
| Omit Knowledge | 0.20±0.11 | 0.42±0.14 | 0.15±0.10 | 0.29±0.12 | 0.28±0.16 | 0.24±0.26 | 0.12±0.10 | 0.29±0.19 | 0.29±0.12 |
| Reverse | 0.29±0.13 | 0.18±0.20 | 0.25±0.11 | 0.33±0.10 | 0.25±0.25 | 0.40±0.17 | 0.22±0.24 | 0.23±0.19 | 0.27±0.14 |
| Reverse-Raw | 0.27±0.15 | 0.62±0.33 | 0.20±0.10 | 0.56±0.25 | 0.48±0.25 | 0.43±0.19 | 0.68±0.39 | 0.30±0.17 | 0.39±0.16 |
| Random Guess | 0.20±0.11 | 0.18±0.22 | 0.15±0.11 | 0.19±0.18 | 0.23±0.15 | 0.11±0.11 | 0.14±0.06 | 0.16±0.17 | 0.23±0.77 |
| **Claude 2** | | | | | | | | | |
| Raw Data | 0.51±0.17 | 0.93±0.13 | 0.80±0.24 | 0.78±0.26 | 0.65±0.20 | 0.59±0.16 | 0.94±0.13 | 0.43±0.20 | 0.66±0.17 |
| Omit Data | 0.67±0.21 | 1±0 | 0.84±0.17 | 0.56±0.23 | 0.50±0.24 | 0.48±0.12 | 1±0 | 0.29±0.38 | 0.67±0.24 |
| Omit Knowledge | 0.19±0.11 | 0.38±0.26 | 0.28±0.18 | 0.37±0.26 | 0.28±0.16 | 0.15±0.12 | 0.09±0.14 | 0.23±0.19 | 0.4±0.29 |
| Reverse | 0.38±0.08 | 0.02±0.08 | 0.19±0.12 | 0.25±0.34 | 0.16±0.14 | 0.32±0.21 | 0.50±0.34 | 0.27±0.18 | 0.24±0.15 |
| Reverse-Raw | 0.57±0.12 | 0.89±0.16 | 0.87±0.16 | 0.89±0.20 | 0.64±0.26 | 0.51±0.12 | 0.95±0.12 | 0.37±0.17 | 0.63±0.17 |
| Random Guess | 0.10±0.12 | 0.26±0.28 | 0.25±0.08 | 0.31±0.27 | 0.12±0.17 | 0.27±0.22 | 0.18±0.33 | 0.27±0.30 | 0.32±0.33 |

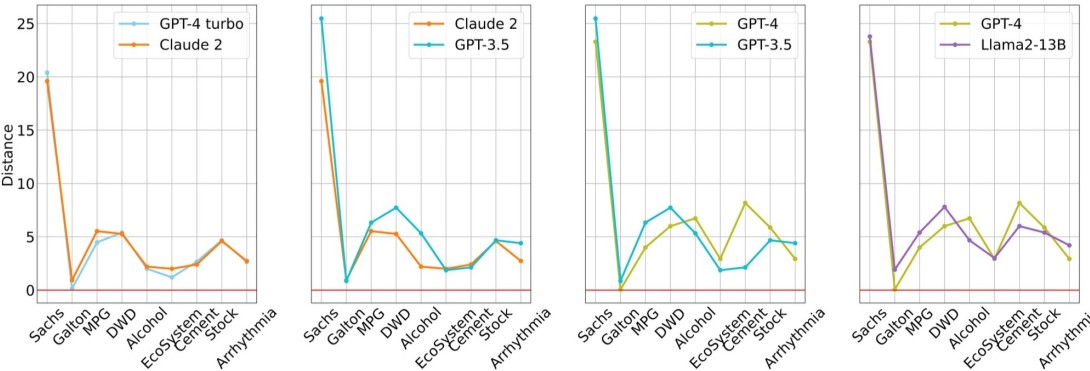

Figure E.2: The comparisons of Structural Hamming Distances among LLMs, including GPT-4 turbo, GPT-4, GPT-3.5, Claude 2, and LLaMa2-13B.

Table E.2: The results of false discovery rates of LLMs for different datasets.

| Method/Dataset | Sachs | Galton | Alcohol | EcoSystem | MPG | DWD | Cement | Stock | Arrhythmia |
|---|---|---|---|---|---|---|---|---|---|
| **GPT-4 turbo** | | | | | | | | | |
| Raw Data | 0.50±0.19 | 0±0 | 0±0 | 0.27±0.23 | 0.42±0.16 | 0.51±0.16 | 0.01±0.04 | 0.70±0.33 | 0.54±0.15 |
| Omit Data | 0.56±0.21 | 0±0 | 0.01±0.05 | 0.46±0.23 | 0.29±0.07 | 0.47±0.18 | 0.05±0.12 | 0.59±0.46 | 0.41±0.11 |
| Omit Knowledge | 0.91±0.07 | 0.80±0.19 | 0.79±0.13 | 0.72±0.24 | 0.84±0.16 | 0.65±0.20 | 0.88±0.15 | 0.84±0.16 | 0.58±0.26 |
| Reverse | 0.50±0.19 | 0.87±0.16 | 0.77±0.16 | 0.86±0.21 | 0.94±0.10 | 0.74±0.20 | 0.15±0.19 | 0.77±0.17 | 0.91±0.26 |
| Reverse-Raw | 0.52±0.29 | 0±0 | 0.08±0.16 | 0.17±0.19 | 0.50±0.14 | 0.51±0.21 | 0.03±0.12 | 0.83±0.15 | 0.49±0.20 |
| Random Guess | 0.95±0.19 | 0.60±0.49 | 0.72±0.41 | 0.60±0.49 | 1±0 | 0.94±0.12 | 0.27±0.44 | 0.80±0.40 | 0.87±0.34 |
| **GPT-4** | | | | | | | | | |
| Raw Data | 0.66±0.24 | 0±0 | 0.40±0.33 | 0.41±0.18 | 0.46±0.14 | 0.67±0.09 | 0±0 | 0.70±0 | 0.48±0.11 |
| Omit Data | 0.45±0.18 | 0±0 | 0.41±0.25 | 0.46±0.07 | 0.28±0.08 | 0.52±0.12 | 0±0 | 0.61±0.23 | 0.37±0.07 |
| Omit Knowledge | 0.67±0.20 | 0.72±0.21 | 0.72±0.08 | 0.60±0.12 | 0.68±0.15 | 0.67±0.10 | 0.79±0.14 | 0.83±0.09 | 0.56±0.13 |
| Reverse | 0.66±0.24 | 0.87±0.21 | 0.68±0.09 | 0.84±0.16 | 0.70±0.13 | 0.74±0.12 | 0.28±0.41 | 0.71±0.03 | 0.67±0.08 |
| Reverse-Raw | 0.75±0.13 | 0±0 | 0.62±0.17 | 0.48±0.22 | 0.50±0.15 | 0.72±0.08 | 0.06±0.22 | 0.70±0.01 | 0.52±0.16 |
| Random Guess | 0.93±0.25 | 0.85±0.36 | 0.87±0.34 | 0.79±0.41 | 1±0 | 1±0 | 0.81±0.39 | 0.86±0.35 | 1±0 |
| **GPT-3.5** | | | | | | | | | |
| Raw Data | 0.87±0.08 | 0.07±0.17 | 0.36±0.37 | 0.32±0.31 | 0.61±0.19 | 0.72±0.21 | 0.02±0.05 | 0.61±0.24 | 0.76±0.20 |
| Omit Data | 0.80±0.14 | 0.04±0.10 | 0.24±0.33 | 0.46±0.23 | 0.61±0.20 | 0.54±0.12 | 0.01±0.03 | 0.60±0.30 | 0.73±0.13 |
| Omit Knowledge | 0.89±0.10 | 0.67±0.12 | 0.78±0.10 | 0.69±0.30 | 0.81±0.14 | 0.80±0.13 | 0.91±0.06 | 0.67±0.28 | 0.67±0.20 |
| Reverse | 0.87±0.08 | 0.85±0.16 | 0.74±0.13 | 0.91±0.23 | 0.83±0.16 | 0.91±0.11 | 0.86±0.24 | 0.77±0.12 | 0.75±0.19 |
| Reverse-Raw | 0.81±0.13 | 0.11±0.21 | 0.42±0.38 | 0.10±0.20 | 0.62±0.21 | 0.71±0.22 | 0.07±0.19 | 0.67±0.20 | 0.68±0.15 |
| Random Guess | 0.91±0.09 | 0.70±0.35 | 0.83±0.10 | 0.59±0.48 | 0.82±0.17 | 0.83±0.26 | 0.92±0.07 | 0.77±0.22 | 0.69±0.27 |
| **LLaMa2-13B** | | | | | | | | | |
| Raw Data | 0.71±0.23 | 0.53±0.32 | 0.79±0.13 | 0.59±0.22 | 0.40±0.26 | 0.55±0.26 | 0.34±0.37 | 0.67±0.18 | 0.62±0.14 |
| Omit Data | 0.67±0.17 | 0±0 | 0.66±0.18 | 0.50±0.40 | 0.36±0.22 | 0.72±0.30 | 0.23±0.27 | 0.74±0.25 | 0.64±0.23 |
| Omit Knowledge | 0.80±0.11 | 0.58±0.14 | 0.85±0.10 | 0.71±0.12 | 0.75±0.15 | 0.82±0.16 | 0.88±0.10 | 0.71±0.18 | 0.71±0.11 |
| Reverse | 0.71±0.23 | 0.82±0.20 | 0.75±0.11 | 0.68±0.07 | 0.75±0.25 | 0.53±0.19 | 0.79±0.23 | 0.77±0.18 | 0.74±0.13 |
| Reverse-Raw | 0.72±0.14 | 0.40±0.33 | 0.80±0.10 | 0.56±0.12 | 0.53±0.25 | 0.55±0.22 | 0.31±0.38 | 0.71±0.15 | 0.61±0.16 |
| Random Guess | 0.81±0.11 | 0.82±0.22 | 0.84±0.12 | 0.81±0.18 | 0.77±0.15 | 0.89±0.11 | 0.86±0.06 | 0.84±0.16 | 0.77±0.20 |
| **Claude 2** | | | | | | | | | |
| Raw Data | 0.51±0.16 | 0.07±0.13 | 0.24±0.26 | 0.30±0.22 | 0.37±0.22 | 0.44±0.17 | 0.10±0.14 | 0.61±0.17 | 0.42±0.14 |
| Omit Data | 0.33±0.21 | 0±0 | 0.19±0.20 | 0.49±0.22 | 0.51±0.24 | 0.49±0.13 | 0±0 | 0.72±0.37 | 0.34±0.22 |
| Omit Knowledge | 0.80±0.11 | 0.66±0.20 | 0.73±0.15 | 0.66±0.21 | 0.73±0.15 | 0.85±0.12 | 0.91±0.14 | 0.78±0.17 | 0.67±0.18 |
| Reverse | 0.62±0.08 | 0.98±0.08 | 0.81±0.12 | 0.77±0.29 | 0.84±0.14 | 0.73±0.19 | 0.50±0.34 | 0.74±0.15 | 0.76±0.15 |
| Reverse-Raw | 0.45±0.11 | 0.12±0.18 | 0.19±0.21 | 0.24±0.24 | 0.38±0.24 | 0.47±0.13 | 0.08±0.12 | 0.61±0.19 | 0.41±0.15 |
| Random Guess | 0.90±0.12 | 0.79±0.21 | 0.75±0.08 | 0.87±0.19 | 0.88±0.17 | 0.73±0.22 | 0.82±0.33 | 0.73±0.30 | 0.68±0.33 |

the models then proceed to deduce causal relationships, often without mentioning or performing a conditional independence test, which would be a critical step in a more thorough causal analysis. This pattern in the responses suggests that the LLMs prioritize correlation as a basis for causal inference in this context. It's important to note, however, that this is an interpretation based on the models' generated text outputs, and

Table E.3: The results of Structural Hamming Distance (SHD) of LLMs for different datasets.

| Method/Dataset | Sachs | Galton | Alcohol | EcoSystem | MPG | DWD | Cement | Stock | Arrhythmia |
|---|---|---|---|---|---|---|---|---|---|
| **GPT-4 turbo** | | | | | | | | | |
| Raw Data | 20.40±8.82 | 0.14±0.35 | 2±0 | 1.20±1.22 | 4.47±1.75 | 5.40±0.71 | 2.67±3.53 | 4.67±1.58 | 2.64±0.97 |
| Omit Data | 17.73±2.52 | 0.16±0.35 | 1.80±0.40 | 2±1.37 | 2.80±1.05 | 5.3±0.97 | 0.80±2.99 | 3.27±1 | 2±0.65 |
| Omit Knowledge | 22.87±5.12 | 4.42±0.49 | 8.93±2.43 | 4.27±0.85 | 7.47±1.75 | 8±3.01 | 16.47±7 | 7.33±2.18 | 4±0.76 |
| Reverse | 22.53±1.75 | 4.86±0.35 | 7.93±0.25 | 4.33±0.94 | 7.93±1.18 | 8.13±0.81 | 12±3.35 | 4.73±2.02 | 4.14±0.35 |
| Reverse-Raw | 18.87±2.25 | 0.07±0.26 | 2.07±0.25 | 1.20±0.98 | 4.47±1.54 | 5.73±1.18 | 3.20±3.47 | 5±1.75 | 2.71±0.59 |
| Random Guess | 18.87±2.25 | 3.07±0.26 | 5.13±0.50 | 3±0 | 5.93±0.25 | 6±0 | 8±0 | 3.07±0.25 | 3.14±0.52 |
| **GPT-4** | | | | | | | | | |
| Raw Data | 23.29±2.58 | 0.06±0 | 6.73±3.23 | 2.93±0.93 | 4±1.51 | 6±0 | 8.17±3.69 | 5.87±0.50 | 2.93±1 |
| Omit Data | 18.86±1.68 | 0±0 | 6.13±3.95 | 2.33±0.47 | 3±1.10 | 6±0.37 | 8.5±0.5 | 4.67±1.19 | 2.80±0.54 |
| Omit Knowledge | 25.43±2.41 | 4.60±1.08 | 10.27±2.35 | 4.07±1.12 | 7.20±2.04 | 11.60±2.89 | 21.3±9.26 | 7.80±1.42 | 4.07±0.77 |
| Reverse | 25.57±6.40 | 5±0 | 13.60±1.70 | 0.16±0.16 | 4.60±1.31 | 6±0 | 12.08±7.20 | 5.67±1.14 | 5.20±0.83 |
| Reverse-Raw | 22.93±2.60 | 0±0 | 6.53±3.40 | 2.87±1.45 | 5.87±2.73 | 6±0 | 8.42±5.09 | 6±0 | 3.60±0.88 |
| Random Guess | 20±0 | 3.33±0.47 | 5.53±0.96 | 3.27±0.44 | 6.13±1.02 | 6.20±0.65 | 9.17±1.67 | 4.40±1.40 | 3.53±0.50 |
| **GPT-3.5** | | | | | | | | | |
| Raw Data | 25.47±6.23 | 0.86±1.19 | 5.33±2.12 | 1.87±1.75 | 6.33±1.70 | 7.73±2.74 | 2.13±3.54 | 4.67±1.49 | 4.40±0.71 |
| Omit Data | 20.73±2.29 | 0.50±0.82 | 3.67±1.62 | 1.84±1.77 | 5.53±1.45 | 5.27±0.44 | 2.53±3.44 | 3.44±1.34 | 3.40±0.61 |
| Omit Knowledge | 31±8.41 | 3.79±0.94 | 8.20±1.80 | 4.73±1 | 7.33±2.12 | 9.47±2.65 | 19.20±8.73 | 6±1.70 | 4.80±1.11 |
| Reverse | 27.20±8.57 | 4.79±0.67 | 9.47±2.31 | 4.13±0.81 | 6.20±1.42 | 9.60±1.62 | 13.93±2.72 | 5.11±1.45 | 4.60±0.88 |
| Reverse-Raw | 25.80±8.72 | 1.36±1.59 | 5.13±2.39 | 1.87±1.75 | 6.60±1.70 | 7.67±2.91 | 1.60±3.20 | 4.78±1.31 | 4.40±0.88 |
| Random Guess | 23.67±1.49 | 3.29±0.45 | 6.87±0.81 | 3.60±0.49 | 6.47±1.15 | 6.33±1.66 | 11.73±1.12 | 5.11±0.99 | 3.87±0.88 |
| **LLaMa2-13B** | | | | | | | | | |
| Raw Data | 23.78±3.39 | 1.92±1.27 | 4.67±0.47 | 3.0±0.89 | 5.40±1.78 | 7.80±2.04 | 6±4.89 | 5.40±1.02 | 4.20±0.83 |
| Omit Data | 22±2.11 | 1±0 | 4±0.19 | 2.13±1.93 | 5.80±0.54 | 6.40±0.71 | 7.40±4.36 | 3±0 | 3.73±0.77 |
| Omit Knowledge | 23.44±2.41 | 3.38±1.08 | 7±0 | 4.27±0.44 | 7.07±1.12 | 7.67±1.40 | 10.33±1.07 | 4.80±1.47 | 4.53±0.81 |
| Reverse | 24.56±2.54 | 4.62±0.62 | 7.33±0.47 | 3.27±0.57 | 6.53±0.50 | 8.47±1.86 | 13.80±2.56 | 4.20±1.17 | 4.13±1.15 |
| Reverse-Raw | 24.11±2.47 | 2.23±1.42 | 5±1.63 | 3±0.89 | 5.33±1.07 | 8.47±2.06 | 9.13±5.43 | 5.20±0.98 | 4.07±0.85 |
| Random Guess | 24.44±2.22 | 3.38±0.49 | 6.67±0.47 | 3.73±0.77 | 6.40±1.36 | 7.60±1.36 | 0.12.87±1.20 | 5.20±0.75 | 3.80±0.75 |
| **Claude 2** | | | | | | | | | |
| Raw Data | 19.60±1.36 | 0.93±0.93 | 2.20±1.47 | 2±0 | 5.53±1.02 | 5.27±1.24 | 2.40±2.70 | 4.60±1.89 | 2.73±1.12 |
| Omit Data | 17.07±1.81 | 1.07±0.57 | 2.07±1.69 | 2±0.96 | 5.27±0.85 | 4.93±0.93 | 3.53±3.42 | 3.13±0.34 | 1.73±1 |
| Omit Knowledge | 23.60±2.39 | 4.80±1.72 | 7.27±1.44 | 5.15±0.77 | 6.87±1.78 | 9±1.75 | 11.67±1.14 | 5.53±1.75 | 3.93±0.85 |
| Reverse | 22.07±1.06 | 4.40±0.71 | 8.13±0.96 | 3.23±0.80 | 6.20±1.17 | 6.73±1.39 | 13.33±4.66 | 4.27±1.29 | 5.13±0.62 |
| Reverse-Raw | 19.33±1.45 | 0.87±1.09 | 3.53±1.41 | 2±0 | 5.20±0.91 | 5.87±0.34 | 3.53±2.74 | 4.53±1.78 | 3.13±0.62 |
| Random Guess | 21.33±1.58 | 3.80±0.65 | 6.60±0.80 | 3.23±0.42 | 6.73±1.53 | 7.27±1.88 | 9.93±1.34 | 5±1.15 | 3.73±0.77 |

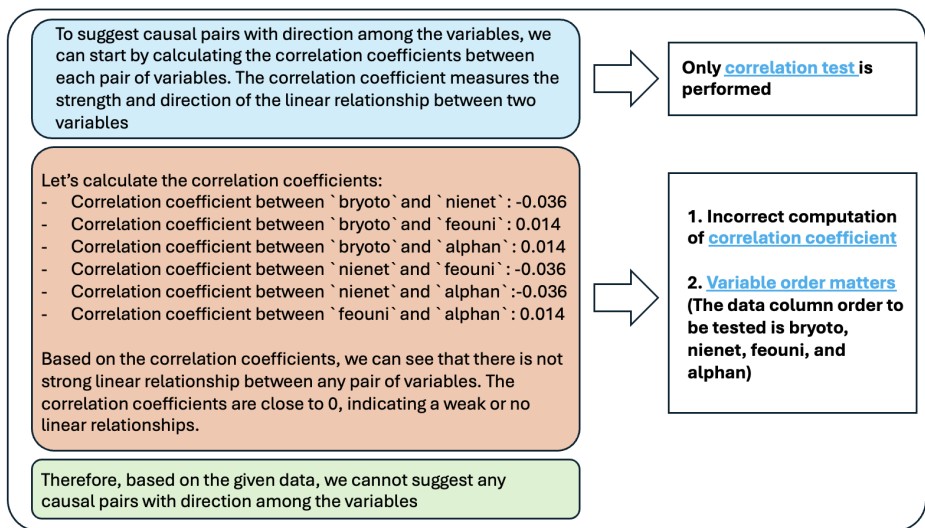

Figure F.1: Chain of Thought for causal discovery when the knowledge is omitted.

it does not imply that the LLM internally calculates correlation coefficients or conducts statistical tests as a

human would. Rather, it reflects the models' training on how to approach causal discovery tasks presented in this manner.

## G   Details of Other Tasks and Generalizability

For knowledge-based pairwise causal discovery (PCD), there are the following two dataset formats.

• PCD (One Hypo): The first format is to see if the premise and one hypothesis have a causal relationship or not. There are 4000 data points in this format. For example:

- "premise": "The woman tolerated her friend's difficult behavior."

- "hypothesis": "The woman knew her friend was going through a hard time."

• PCD (Two Hypo): The second format is to see if the premise has which cause or effect from two provided hypotheses. There are 2000 data points in this format. For example:

- "premise": "Tom always wore very few clothes when he was young."

- "ask-for": "effect",

- "hypothesis1": "His heat loss from the skin decreased."

- "hypothesis2": "He has a varicocele."

• For causality event identification (CEI), the task is to check if two words have a causal relationship. CEI has a total of 2596 data points. For example, check if "cut" and "growth" have a causal relationship in the following text:

- "The earnings growth also was fueled by the company's ability to cut net financing spending by half to around 15 million guilders ."

The detailed data structure and flowchart of tasks are provided in Figure 7 for PCD and Figure 8 for ECI, respectively.

## H   Prompt Design

The numerical data will be inserted into the **data** placeholder in the user prompt, ensuring that the dataframe layout remains unchanged.

```
// Full Graph Discovery
system = """
You are a helpful assistant to suggest potential causal pairs with direction (A -> B means A
    causes B)
"""
user = """
Suggest causal pairs with direction among following variables after analyzing following
    data::\n{data}. \n MUST Suggest ONLY the directed causal pairs without saying any other things:
"""
// Pairwise Causal Discovery (with LinGAM instruction)
"""
Here is how LinGAM works to derive the causal pairs:

    Given the data with columns [A, B]: \n
```

    1. You first fit the linear regression with column A as the feature and B as the outcome
        variable. Collect the fitted residuals
    and if the residual is correlated with column A, we mark a YES in this case and a NO if not
        correlated
    2. You then fit the linear regression with column B as the feature and A as the outcome
        variable. Collect the fitted residuals
    and if the residual is correlated with column B, we mark a YES in this case and a NO if not
        correlated

If 1 is a YES and 2 is a NO, we say that B causes A; if 1 is a NO and 2 is a YES, we say that A
    causes B. Following above instruction to suggest causal pairs with direction among following
    variables after analyzing following data:\n{data}. MUST Suggest ONLY the directed causal pairs
    without saying any other things:
"""

