# OpenReview forum: "Enhancing Causal Reasoning in Large Language Models: A Causal Attribution Model for Precision Fine-Tuning"
_TMLR — Rejected by TMLR_

### Review · Reviewer_xrmd · 2024-12-20

**Summary Of Contributions:**

The authors investigate the causal reasoning capabilities of LLMs. To this end they define measures that quantify to which degree an LLM draws conclusion from existing knowledge about the variable names and to which degree the conclusions are reached from given numerical data.  They apply these metrics to existing datasets and the results seem to indicate that existing state-of-the-art LLMs primarily rely on prior knowledge about the variable names to draw causal conclusions. Further, the authors evaluate a fine-tuning approach to improve the causal reasoning of LLMs when pre-computations are given to the LLM.

**Audience:**

Yes

**Broader Impact Concerns:**

I don’t have any concerns that go beyond the general impact that all LLM-related research has.

**Claims And Evidence:**

Yes

**Requested Changes:**

## Critical:
### Causal formalism:
- To my knowledge the do-operator is not used to express *counterfactuals* in the way Pearl defines them. Also, the quantities in definitions 2.1-2.4 are no counterfactuals. Intuitively, a counterfactual denotes the probability of an event, if we intervene on a variable but condition on the fact that this variable attained another („factual“) value. I know that this term is used more loosely in other communities. But since you explicitly use Pearl’s causal framework it is important to note this difference.
- Using the do-operator in definitions 2.1-2.4 is not well-defined without a graphical model that it refers to. E.g. if $K, D, C, U$ are the only parents of $Y$ we get $p(y | do(k, d), c, u) = p(y | k, d, c, u)$, but if there are backdoor-paths we need to adjust for them.
- This also raises the question, what exactly the motivation is, to introduce causality in the first place. Is this simply to express that sometimes the variables names or the data are omitted?
- How can the *unmeasured* confounders $u_i$ appear in the conditioning set in definitions 2.1-2.4?
- What does „$i$-th sample“ mean in the beginning of section 2.1? Figure 1 seems to suggest that whole datasets are given to the model.
- If $\hat{y_i} = y_i$ actually denotes that the „causal responses“ are correct (I read this as the output is entirely correct) then the equations in section 3.8 are not correct
- Section 2.2: „The DAG G can be estimated up to a Markov equivalence class“. At this point, you haven’t explained how the graph relates to probability distributions. I would at least mention the Markov Condition.
- The last equation in section 2.1: Why is this difference an interesting quantity and how does it „indicate robustness“?
- Section 5: the residuals of a least square regression are uncorrelated with the covariates by construction. Algorithms based on LiNGAM need to check for more general (non-linear) dependencies
- Section 6: „data may come with noise“ I’m not sure if noise is the issue with real data. In fact, causal discovery without noise poses its own challenges. Did you mean finite-sample issues or data missingness?
### Clarity:
- I am confused as to which experiment uses real and which one simulated data. I think this comes from the fact that section 3.6 appears between 3.5 and 3.7. I guess the tables in the main paper contain only real data? Section D seems to say that Table D.1 contains simulated data, but how does this then relate to the different dataset (I guess every pair is semi-synthetic with a real cause and a simulated effect)?
- In Section 3.8, does „Reverse-Raw“ only alter the ordering of the data columns? Then it is confusing to refer to „graph reversal“ without further comment
- I suppose the F1 score is calculated w.r.t. the binary decision of whether there is a directed edge between any pair $(X, Y) \in V^2$. But this should be made explicit.
- Why is there still randomness in the output if you fix the temperature to zero? Do you provide subsamples of the dataset to the LLM?
- The paper concludes that the LLMs rely on pre-existing knowledge but also that they have „limited but existent causal discovery abilities based on numerical data“. I agree that the results point towards this direction, but I am a bit concerned about the size of the standard deviations in Table 2, especially for the latter claim. E.g. for CAD of GPT-4 turbo on Sachs the standard error is magnitudes larger than the estimand. This makes it hard to asses whether the results are significant and I would be more cautious with conclusions. Ideally, one could add a statistical test like a Mann-Whitney-U-test to see whether the Random Guess FDR of GPT-4 turbo is significantly smaller than the respective, more informed FDR. (Though I think the critical part would be to emphasize this uncertainty. The statistical test would be a bonus)
- Why is Figure 4 a „hierarchy“? I don’t think all the pairs are obviously ordered and there are no further comments on this
### Fine-tuning:
I’m not entirely sure how much value the fine-tuning part adds to the paper. So at least, this needs to be motivated better. I think my main concern is that I’m not sure which question this experiment wants to answer
- Do we want to test whether LLMs are „aware“ of results like Thm 3.1? Then I don’t get why we fine tune
- Do we want to see whether the LLM has an „intuitive“ understanding of causality? Then I think the pre-computations are already too specific, as already the choice of provided statistics hints towards Thm 3.1 and given that the remaining task is basically an „if-else“ clause.
- Do we want to see whether an LLM can „be taught“ the difference between causation and correlation? Then again, I think the given statistics are too strong. It’s maybe comparable to saying „$X$ is an unshielded collider between $Y$ and $Z$“ instead of providing statements about the correlations
- Is the purpose to improve general causal reasoning abilities in LLMs? Then I would hope to see an improvement in Table 5, but the paper seems to suggest that the goal is just to not to get worse here

All this is not to say that I think new experiments are unavoidable for the paper to be publishable. But it certainly needs to clarify the goals of the experiment.

## Recommended:
- Why does Table 3 report F1? Would it report FDR one could at least interpret the Random Guess outputs of the LLMs as baseline for the scores reported in Table 2. This way it is hard to get a feeling for them. This is related to the point above about statistical significance
- Related work mentions feature importance techniques only in the appendix. At least the way definitions 2.1-2.4 are presented, it seems to me that this connection is quite important, as the general logic of the attribution is not a-priori restricted to LLMs. Along these lines, I think it could be worth to discuss the literature on causal feature attribution. E.g. Heskes et al. (Causal Shapley Value) and Janzing et al. (Feature relevance quantification in explainable AI) contain terms that are syntactically quite close to definitions 2.1-2.4.
- Figure 1 caption: describe that variable names are changed. At the point the figure is first referenced this is not clear yer. Also, why is there only one edge in the second response?
- Why is it „without loss of generality“ to focus on the causal discovery? I think this technical phrase is too strong for the given matter. The first occurrence of „w.l.o.g.“ does not make sense to me at all.
- I think it is uncommon to say the do-operator „simulates“ interventions. I would maybe say it denotes them.
- The notation in section 3.8 is really weird. It could be worth the effort to make this more readable, since this is a crucial part of the paper.
- Section 6 „where internal knowledge [of LLMs] would be more reliable“ I think this claim needs further justification, or more, it needs to clarify what the goal of causal discovery is. In my opinion a main motivation for (statistical) causal discovery is to extend or enhance human knowledge about causality beyond existing knowledge. I’m not sure if I would call mining causal knowledge from existing text „causal discovery“.  But this is not set in stone, so if you clarify what you mean, the sentence could also make sense.
- I would expect that since all datasets are available publicly, most LLMs have seen them during training. I understand that it’s virtually impossible to check this and also probably hard to get datasets for which we can rule out that they have been used during training. But I think I would be curious to hear the authors’ opinion on how this affects the conclusions of the paper (or whether they think it does)

## Minor comments and typos
- Page 1 last line „known intrinsically“ -> known to intrinsically
- Figure 1 caption „second presents“ -> second one presents
- Several places (e.g. the first bullet point of contributions and third line of page 4) where I would put „the“ before „LLM“
- Last bullet point of contributions „achieves the highest accuracy“ the highest among what?
- Section 3.4: what does „biologically rich“ mean?
- Section 3.6: is Geiger 1990 the citation you wanted here?
- Several occurrences „LinGAM“ -> „LiNGAM“ (the N stands for „non“)

**Strengths And Weaknesses:**

### Strengths:
- I think the main idea of the paper, to dissect the reasoning of LLMs into prior knowledge and numerical reasoning, is an insightful contribution to the discussion on causal reasoning with LLMs.
- The experiments cover plenty of scenarios and models

### Weaknesses:  *(See also detailed comments below)*
- The formal exposition of causal concepts is imprecise and partially wrong.
- Lacking clarity in the description and evaluation of the experimental setup
- The fine-tuning part needs to clarify its contribution

---

> ### Author Response · Authors · 2025-01-25
> **Response to Reviewer xrmd (Part 1)**
>
> Thanks for your valuable comments and suggestions! We are encouraged by your acknowledgment and positive comments on our work. Below, we summarize your questions and comments in quotes and provide our point-by-point responses. Please refer to the latest submission for the revised paper taking all your suggestions.
>
>
> ***Response to Weaknesses***
>
> 1. > The formal exposition of causal concepts is imprecise and partially wrong.
>
> To our best understanding, this comment is related to “Causal formalism”. To clarify any confusion, we would like to summarize our point-by-point responses below.
>
> First, **the used do-operator and terminologies indeed exactly follows Pearl’s works**, and we precisely provide all the related references with detailed pages and lines for your information.
>
> Second, we would like to highlight the main difference between our work and other traditional causal works is, we can **design experiments and do intervention directly to assess the proposed probabilities under different scenarios** (also known as randomized trials in the potential outcome framework). This helps to avoid using graphical modeling and identification assumptions for estimating counterfactual realities based on observational data.
>
> Third, **please note we NEVER mention our Definitions 2.1-2.4 are counterfactual definitions**. We only mention in this paper that the terms in Definitions 2.1-2.4 could help us construct counterfactual scenarios, i.e., the case with or without knowledge inputs.
>
> We hope the above highlights help. Please also check our detailed response for **Causal formalism** below.
>
>
> 2. > Lacking clarity in the description and evaluation of the experimental setup
>
> We appreciate your thorough comments in improving the clarity in the description and evaluation of the experimental setup. We’ve provided point-by-point responses in “Response to Clarity”
>
>
> 3. > The fine-tuning part needs to clarify its contribution
>
> We are grateful for your comment and would like to briefly clarify our contribution in the fine-tuning part as follows (please refer to “Response to Fine-tuning” part for detailed response). As witnessed in our experiments based on the proposed attribution score, the current LLMs' effectiveness in causal discovery heavily relies on provided context and domain-specific knowledge but can also utilize certain numerical data though not in a right way. This motivates us to consider precisely fine-tuning the way LLMs’ utilizing numerical data for causal discovery. More specifically, we would like to answer the following question: after the fine-tuning, can the LLM learn how to **correctly** utilize the numerical information provided in the input to assist causal discovery inquiries besides the knowledge they already know? To provide the proof of concept, we start the fine-tuned LLM for pairwise causal discovery, which correctly utilizes the numerical information and effectively leverages the knowledge when numerical information is unknown. As shown in our evaluation results, our fine-tuned model achieves the highest accuracy in identifying the true causal relationship by mastering both inherent knowledge and instructed logic of using numerical data.
>
> We hope the above highlights help. Please also check our detailed response for **Fine-tuning** below.

---

> > ### Author Response · Authors · 2025-01-25
> > **Response to Reviewer xrmd (Part 2)**
> >
> > ***Response to Causal formalism***
> >
> > 1. > About the do-operator and counterfactuals
> >
> > First, we would like to clarify **the used do-operator and terms actually indeed follows Pearl’s causal inference framework**. As shown in his works referenced below (see [Pearl, 2012, Sec. 1]), the post-intervention distribution of the target variable $Y=y$ with a structural equation model (SEM) $M$ resulting from the action $do(X=x)$ is given by the equation
> >
> > $$
> > P_M(y \mid do(x))=P_{M_x}(y),
> > $$
> >
> > where the resulting model is denoted by $M_x$ and $P$ is probability. In our work, we use the first expression for a clear presentation. Similar expressions are also adopted in Equations 3-5 in Section 1.2 of Pearl [2012] where the SEM $M$ is compressed, i.e.,
> >
> > $$
> > P(y \mid do(x),z),
> > $$
> >
> > where $x$ is intervened and $z$ is observed.
> >
> > Second, we admit there are different usages of counterfactual v.s. factual, but **please note we NEVER mention our Definitions 2.1-2.4 are counterfactuals** in the paper. Indeed, we only comment that the terms in Definitions 2.1-2.4 could help us construct counterfactual scenarios, i.e., the case with or without knowledge inputs.
> >
> > To link with literature, our Definitions 2.1-2.4 actually follow how Pearl used do-Calculus to define the Controlled Direct Effect (see [Pearl, 2012, Sec. 2] and [Pearl, 2001]), i.e., $E(Y|do(X = 1,M =m))−E(Y|do(X =0,M =m))$ for mediator $M$. We’ve included the above clarification in the appendix for more background.
> >
> >  - Pearl, J. (2012). The do-calculus revisited. In Proceedings of the Twenty-Eighth Conference on Uncertainty in Artificial Intelligence (pp. 3-11).
> >  - Pearl, J. (2001). Direct and indirect effects. In Proceedings of the Seventeenth Conference on Uncertainty and Artificial Intelligence, 2001 (pp. 411-420).
> >  - Pearl, J. (2009). Causality: Models, Reasoning, and Inference. Cambridge university press.
> >
> >
> > 2. > About not referring to a graphical model
> >
> > This is an excellent comment. As we commented in the previous note, in this work, we use the do-operator to express the post-intervention distribution of the target variable $Y=y$ given different interventions. Our follow-up experiment design in Section 3 allows us to directly realize these interventions to estimate the attribution scores, such as omit knowledge and then calculate the accuracy. In other words, using Pearl’s Three Layer Causal Hierarchy, we are directly collecting data on Level 2 - Intervention. As indicated in the ladder of causation, in such a scenario, we do not need a graphical model to help us infer Level 3 - Counterfactuals.
> >
> > In another note, we admit that a graphical model among the considered variables will be helpful for observational studies but it also requires a strong brief on how the LLM operates. We leave this for future research.
> >
> >  - Pearl, J. (2009). Causality: Models, Reasoning, and Inference. Cambridge university press.
> >
> >
> > 3. > About the motivation of introducing causality.
> >
> > Thank you for your question. To our best understanding of your comment, we primarily introduce the causality part regarding the causal discovery problems - the main LLMs’ task we focused on in this paper.
> >
> > To make it further clear, we only use the do-operator (proposed by Pearl at the first place) in our method to express the post-intervention distribution of the outcome given different interventions (see [Pearl, 2012, Sec. 1]).
> >
> > Finally, the main difference between our work and other traditional causal works is, we can **design experiments and do intervention directly to calculate the proposed probabilities under different scenarios**. This helps to avoid using graphical modeling and identification assumptions for estimating counterfactual realities based on observational data.
> >
> >  - Pearl, J. (2012). The do-calculus revisited. In Proceedings of the Twenty-Eighth Conference on Uncertainty in Artificial Intelligence (pp. 3-11).
> >
> >
> > 4. > About the unmeasured confounders
> >
> > Thank you for pointing this out. We correct $u$ more precisely as all other controlled confounding variables that we do not include as intervention, such as training data, language model version, etc. As we mentioned previously, we are only interested in changed accuracy under two interventions, also known as randomized trials in the potential outcome framework, the unmeasured confounders thus are less concerned. We’ve revised the paper accordingly and appreciate your comment again.

---

> ### Author Response · Authors · 2025-01-25
> **Response to Reviewer xrmd (Part 3)**
>
> ***Continued Response to Causal formalism***
>
>
> 5. > About the i-th sample and Figure 1
>
> The i-th sample corresponds to one inquiry/input the LLM receives and the output it produces, including the input context, which provides the scenario for the causal reasoning task; the embedded causal knowledge within variable names, which may carry implicit causal cues; and numerical information such as summary statistics, providing explicit causal relationships; and the causal reasoning results made by an LLM.
>
> The provided Figure 1 indicates ONE sample using the LLM to provide causal discovery results, where the blue box is the i-th context, the pink box is the i-th knowledge embedded in variable names, the orange box is the i-th  numerical information, and the red box at the last is the i-th output. A more concrete example is provided in Figure 2 using the Galton’s Family as one sample.
>
> In this paper, we consider generating such inputs/outputs based on nine causal discovery benchmark datasets as detailed in Sections 3.1 and 3.2. Specifically, we consider one benchmark data and randomizing the variable/column ordering as one sample, with 15 randomization in total, which yields 15*9 samples overall. We’ve added the above clarification in our paper.
>
>
> 6. > About $y_i$ and $\widehat{y}_i$
>
> Please note both $y_i$ and $\widehat{y}_i$ will be a set of causal pairs as ground truth and as what are provided by an LLM, respectively. Hence, $\widehat{y}_i=y_i$ does NOT mean “responses are entirely correct”. We would appreciate the reviewer's further comments on why the equations in section 3.8 are not correct.
>
>
> 7. > About Section 2.2 “The DAG G can be estimated up to a Markov equivalence class“ and how the graph relates to probability distributions
>
> We would like to emphasize the main purpose of Section 2.2 is to introduce the DAG as the outcome of interest (i.e., causal graphs/pairs) in the target task we would like to evaluate LLMs on. While we acknowledge that probability distributions and Markov Condition are helpful to explain the Markov equivalence class, it is not the main goal of this paper. We indeed included more related works for causal discovery and identification in Appendix A of the original paper.
>
>
> 8. > About the last equation in section 2.1
>
> Thank you for your insightful comments. The difference between MAK and MAD shows the discrepancy between the marginal attributions in terms of knowledge and data, i.e., how marginally including knowledge or data would increase the accuracy. This difference reflects the more important factor among knowledge and data, for LLMs’ reasoning process. Similarly, the difference between CAK and CAD helps to identify the dominating factor while conditional on the other factor’s information. The last equation in Section 2.1 indicates that no matter which difference (either MAK-MAD or CAK-CAD), the conclusion is **consistent** and thus we say it’s robust. We’ve added the above discussion in our paper.
>
>
> 9. > About Section 5 and LinGAM
>
> We agree with the reviewer that when the residuals of a least square regression are uncorrelated with the covariates, there may exist more general (non-linear) dependencies. However, we would like to highlight that **our simulation data is constructed to reflect linear dependencies**, based on the relationship between Father’s Height and Child’s Height when applying the LiNGAM algorithm.
>
>
> 10. > About Section 6: “data may come with noise”
>
> Thanks for your comments. Yes, we mean finite-sample issues or possible data missingness in the real datasets. We’ve made it clear in our revised paper.

---

> > ### Author Response · Authors · 2025-01-25
> > **Response to Reviewer xrmd (Part 4)**
> >
> > ***Response to Clarity***
> >
> >
> > 1. > About which experiment uses real and which one simulated data.
> >
> > We are grateful for your comments and would like to clarify the datasets we used as follows. In Sections 3.3-3.5 and 3.7, we conduct experiments for general causal discovery tasks with multiple variables. As we discussed earlier, we consider generating samples based on nine causal discovery benchmark datasets, detailed in Sections 3.1 and 3.2. Specifically, we consider one benchmark data and randomizing the variable/column ordering as one sample, with 15 randomization in total, which yields 15*9 samples overall.
> >
> > In Section 3.6, we consider another pairwise causal discovery task, and you are right, these causal pairs (such as Father’s Height cause Child’s Height) are selected from the benchmark datasets while the effects are simulated based on a linear relationship with non-Gaussian noise. For each semi-synthetic pair, we evaluate the accuracy by randomizing the order of the two input variables. The detailed generating procedure was provided in Section 3.6.
> >
> > We’ve added the above clarification in our paper.
> >
> >
> > 2. > About “Reverse-Raw”
> >
> > Please note that in the scenario of the reverse causal discovery task in Section 3.7, also denoted as “Reverse-Raw”, we reverse the original topological order of the causal graph, as illustrated in Figure 3. For instance, the original topological order is, “Gene” causes “Gender” causes “Height”, with the data reflecting such a topological order. After changing the order of variable names based on the transposed adjacency matrix (one to one mapping of causal graph), we have the reversed causal graph with “Height” causes “Gender” causes “Gene”. We’ve added all the above clarification and discussion in our revised paper.
> >
> >
> > 3. > About the F1 score
> >
> > The F1 score is calculated based on the binary decision of whether the LLM correctly predicts the **direction** of the directed edge under each scenario. We’ve added more details in the paper.
> >
> >
> > 4. > About randomness when the temperature is zero; do you provide subsamples?
> >
> > Yes, we provide random subsamples of certain datasets to the LLM to accommodate the prompt length restrictions. We maximize the sample size input to the LLM while staying within the length limits.
> >
> >
> > 5. > About “limited but existent causal discovery abilities based on numerical data“, the size of the standard deviations in Table 2, and adding a statistical test like a Mann-Whitney-U-test
> >
> > We would like to thank the reviewer for the insightful comments. Per your suggestion, we have performed additional statistical analysis in response to your concerns about the size of the standard deviations noted in Table 2. Specifically, we conducted the Mann-Whitney U test with an asymptotic approximation on the Sachs dataset (the most complicated benchmark dataset) for 20 iterations under the random guess and numerical data-only conditions. The resulting p-value of 0.05, though close to the threshold, indicates a statistically significant difference, which lends support to our claim of “limited but existent causal discovery abilities based on numerical data”. This analysis also addresses the concern regarding the large standard deviations and provides a more robust basis for our conclusions. We have included this discussion and the test results in the revised manuscript.
> >
> >
> > 6. > About “hierarchy” in Figure 4
> >
> > Thank you for your comment. Please note excluding the second pair, the other three pairs demonstrate that one LLM consistently outperforms the other across almost all datasets, with at most a slight disadvantage in one dataset. We’ve added more discussion in the paper.

---

> > > ### Author Response · Authors · 2025-01-25
> > > **Response to Reviewer xrmd (Part 5)**
> > >
> > > ***Response to Fine-tuning***
> > >
> > >
> > > > About which question the fine-tuning experiment wants to answer
> > >
> > > We are grateful for your comment and would like to briefly clarify our question in the fine-tuning part as follows. As witnessed in our experiments based on the proposed attribution score, the current LLMs' effectiveness in causal discovery heavily relies on provided context and domain-specific knowledge but can also utilize certain numerical data though not in a right way. This motivates us to consider precisely fine-tuning the way LLMs’ utilizing numerical data for causal discovery. More specifically, we would like to answer the following question: after the fine-tuning, can the LLM learn how to **correctly** utilize the numerical information provided in the input to assist causal discovery inquiries besides the knowledge they already know?
> > >
> > >
> > > > About the contribution of the fine-tuning part added to the paper
> > >
> > > To provide the proof of concept, we start the fine-tuned LLM for pairwise causal discovery, which correctly utilizes the numerical information and effectively leverages the knowledge when numerical information is unknown. As shown in our evaluation results, our fine-tuned model achieves the highest accuracy in identifying the true causal relationship by mastering both inherent knowledge and instructed logic of using numerical data.
> > >
> > > We’ve added the above discussion in our paper.

---

> > > > ### Author Response · Authors · 2025-01-25
> > > > **Response to Reviewer xrmd (Part 6)**
> > > >
> > > > ***Response to Recommended***
> > > >
> > > > 1. > About Table 3 reporting F1
> > > >
> > > > Thank you for your comment. We acknowledge that FDR can help to interpret the Random Guess outputs of the LLMs as baseline and we would like to clarify that we indeed already provided the results of **FDR** in Table E.2 in Appendix E of the original paper, along with the results of TDR, and SHD in Tables E.1 and E.3 in Appendix E. The reason why we presented the F1 score in the main text is because the F1 score provides a more comprehensive measure of the reasoning capability of LLMs compared to the FDR.
> > > >
> > > > As we mentioned in the earlier response, we conducted the Mann-Whitney U test with an asymptotic approximation on the Sachs dataset (the most complicated benchmark dataset) for 20 iterations under the random guess and numerical data-only conditions. The resulting p-value of 0.05, though close to the threshold, indicates a statistically significant difference.
> > > >
> > > > We have included all the related discussion in the revised manuscript.
> > > >
> > > >
> > > > 2. > About related works on causal feature attribution
> > > >
> > > > Thank you for your valuable comment and pointing to additional references. We’ve included and discussed the works by Heskes et al. (Causal Shapley Value) and Janzing et al. (Feature relevance quantification in explainable AI) in our related works and moved Appendix A “More related works - Attribution Models” to the main text.
> > > >
> > > >
> > > > 3. > About Figure 1 caption
> > > >
> > > > Thanks for your helpful comment. The first panel of Figure 1 presents one sample using the LLM to provide causal discovery results, where the blue box is the i-th context, the pink box is the i-th knowledge embedded in variable names, the orange box is the i-th numerical information, and the red box at the last is the i-th output. The second panel shows the generation of one counterfactual sample where the variable names were replaced by non-meaningful letters; and the third one describes our proposed causal attribution framework, where the knowledge is omitted corresponding to the counterfactual scenario in the second panel.
> > > >
> > > > We’ve revised the caption of Figure 1 according to the above clarification.
> > > >
> > > >
> > > > 4. >  About “without loss of generality” to focus on the causal discovery and the first occurrence of “w.l.o.g.”
> > > >
> > > > There are two occurrences of “without loss of generality” in this paper. The first one appeared in Section 2.1, where we mentioned that this paper focuses on the causal reasoning task due to its prominence and relevance in recent literature within causal inference and natural language processing, and we detailed several distinct elements in such an LLM task.
> > > >
> > > > The other appeared in Section 2.2, where we further used the causal discovery task as a representative example to detail our approach and findings, following the existing literature (see e.g., Kiciman et al., 2023; Jin et al., 2023a). While this task serves as an illustrative case for our methodologies, we acknowledge the diversity of causal tasks and the potential applicability of our framework to a broader range of scenarios in causal inference and machine learning tasks such as causal effect estimation, correlation prediction, and mathematical reasoning.
> > > >
> > > > We’ve included all the above clarifications and discussions in the revised paper.

---

> > ### Comment · Reviewer_xrmd · 2025-02-20
> >
> > 5 ) and 6)
> >
> >  I agree that the answer to 5) is coherent with Figure 1 and that given the answer to 6) the equations in sec 3.8 would be correct. My confusion comes from the following: if „i“ indicates the whole input/output pairs, $\hat y_i$ contains multiple causal pairs (as in Figure 1 $V1 \to V3$ *and* $V2\to V3$). This contradicts the answer to 6). So under the assumption that $\hat y_i$ is as suggested by Figure 1, the probabilities in Def 2.1-2.4 cannot be expressed (easily) in terms of the TDR of pairs, but should be something like the TDR w.r.t. to whole graphs.
> >
> > 9 )
> >
> > My point was not that the residuals *can* have more complex dependencies on the regressor but that they are *never* correlated with it.
> > I.e. trying to test the condition in Thm 3.1 with correlations (instead of e.g. the Hilbert-Schmidt Independence Criterion) will always return independence (in the limit of infinite data) even in the anti-causal direction. Since the experiments seem to indicate an improvement, I assume you used a library (that uses HSIC or something similar) to calculate the LiNGAM results?

---

> > > ### Author Response · Authors · 2025-03-06
> > > **Response to Official Comment by Reviewer xrmd (Part 1)**
> > >
> > > We are very grateful for your continued engagement and insightful feedback! In response to your comments, we make the following clarifications and additional revision to the paper. Below, we summarize your comments in quotes and provide our point-by-point responses. Please refer to the latest submission for the revised paper taking all your suggestions using **orange** color.
> > >
> > > 1. > Thank you for the clarifications. Now I would agree that the quantities measured in the experiment correspond to the interventional probabilities in Defs 2.1-2.4. Concerning the do operator: My point was not that the SCM (or graphical model) is missing as a subscript from the equation. But rather that this mathematical object is alway defined w.r.t. to such a model, as the quote that you cited from Pearl shows.
> > >
> > > Thank you for your positive feedback and valuable suggestions. We have added the structural graphical model into the revised paper. As we mentioned previously, we do admit that a theoreticized causal graph among the considered variables will be helpful for observational studies but it also requires a strong brief on how the LLM operates. In the revised paper, to achieve a general framework, we consider the graphical model with minimal assumptions on the causal structure as follows. The main outcome of interest, i.e., the causal reasoning accuracy made by an LLM $\hat{y}=y$ is determined by the input context $c$ that provides the scenario for the causal reasoning task, the embedded causal knowledge $k$ that carries implicit causal cues, the numerical data $d$ that gives explicit causal relationships, and all other controlled confounding variables $u$. However, we do not specify the relationship between $c$, $k$, $d$, and $u$, which is presented by the dash line in **the revised Figure 1 (Right Panel)**.
> > >
> > >
> > > 2. > I also didn’t mean to say that you must do confounder correction, but rather that whether you have to depend on the underlying SCM (or graph). Again, given the further explanations I think I agree that you don’t have to.
> > >
> > > Thank you for the clarification. We’re pleased we've reached a consensus on this matter. The graph has been provided in the latest submission and we welcome any more of your invaluable insights.
> > >
> > >
> > > 3. > I strongly agree with reviewer BieH that a visualization of the assumed causal graph would improve clarity. Probably this is what I wanted in the first place, but I was too confused to articulate it. Keep in mind, that these definitions are the very first (specific) aspect of the paper that a reader sees, before being familiarized with further details of the setup.
> > >
> > > Thanks for your comments. Please refer to our response in #1 for a visualization of the assumed causal graph and explanations.

---

> > > > ### Author Response · Authors · 2025-03-06
> > > > **Response to Official Comment by Reviewer xrmd (Part 2)**
> > > >
> > > > 4. > In your response and the new draft I am confused why you mention the Controlled Direct Effect. Do you consider either the variable names or the data as mediator? If so, why? Again, showing the causal graph could help here. Concerning counterfactuals:
> > > >
> > > > We’re grateful for your further comment. The Controlled Direct Effect mentioned in the previous response is only to help clarify the usage of do-operator in our definition. However, we DO NOT assume any mediator in our paper. We agree the causal graph is helpful and have revised the sentence to avoid any confusion.
> > > >
> > > >
> > > > 5. > I am afraid it might be a bit too subtle to differentiate between “counterfactuals” and “counterfactual scenarios” where the latter are supposed to be understood as interventional quantities instead of counterfactual ones.
> > > >
> > > > Thanks for your continued clarification. Per your suggestion, we have revised the paper throughout using “interventional scenarios” instead to describe the experiments we designed. Please refer to the latest version of our paper for details.
> > > >
> > > >
> > > > 6. > I appreciate that you agree that Defs 2.1-2.4 are not counterfactuals, but I’m confused why you also later mention that by “collecting data on Level 2 - Intervention” you “do not need a graphical model to help us infer Level 3 - Counterfactuals” (for which the graphical model couldn’t help anyway).
> > > >
> > > > We are grateful for this opportunity to make further clarification. The reason for mentioning the Pearl’s Three Layer Causal Hierarchy in our previous response, is to emphasize that our interventional scenarios (i.e., experiment design in Section 3) allows us to directly estimate the proposed attribution scores (i.e.,  $P(y_x)$), corresponding to collecting data on “Level 2 - Intervention”. This is different from causal inference using observational studies, where further causal identification assumptions are needed to infer “Level 3 - Counterfactuals” such as $P(y_x|x’)$.

---

> > > > > ### Author Response · Authors · 2025-03-06
> > > > > **Response to Official Comment by Reviewer xrmd (Part 3)**
> > > > >
> > > > > 7. > I agree that the answer to 5) is coherent with Figure 1 and that given the answer to 6) the equations in sec 3.8 would be correct. My confusion comes from the following: if “i” indicates the whole input/output pairs, $\hat{y}_i$ contains multiple causal pairs (as in Figure 1 V1->V3 and V2->V3). This contradicts the answer to 6). So under the assumption that  $\hat{y}_i$ is as suggested by Figure 1, the probabilities in Def 2.1-2.4 cannot be expressed (easily) in terms of the TDR of pairs, but should be something like the TDR w.r.t. to whole graphs.
> > > > >
> > > > > We are grateful for your continued feedback and would like to point out that in answer to 6) we mentioned both $\hat{y}_i$ and $y_i$ are a set of causal pairs, so it is aligned with answer to 5) that $\hat{y}_i$ contains multiple causal pairs (as in Figure 1 V1->V3 and V2->V3). And we agree that the TDR is w.r.t. to the whole graph. In the literature of causal discovery, TDR (True Discovery Rate) w.r.t. to the whole graph is standard metric (see Shimizu et al., 2006; Bühlmann et al., 2014; Ramsey et al., 2017; Zheng et al., 2018; Yu et al., 2019; and Zhu et al., 2020).
> > > > >
> > > > >
> > > > > 8. > My point was not that the residuals can have more complex dependencies on the regressor but that they are never correlated with it. I.e. trying to test the condition in Thm 3.1 with correlations (instead of e.g. the Hilbert-Schmidt Independence Criterion) will always return independence (in the limit of infinite data) even in the anti-causal direction. Since the experiments seem to indicate an improvement, I assume you used a library (that uses HSIC or something similar) to calculate the LiNGAM results?
> > > > >
> > > > > Thank you for your valuable feedback and detailed clarification. We agree that trying to test the condition in Thm 3.1 with correlations using limited data may encounter problems and utilizing the Hilbert-Schmidt Independence Criterion for example is more reliable. For your question, yes, the algorithmic results for LinGAM were obtained by applying the LinGAM implementation from *the causal-learn package* to the constructed datasets, which uses the independent component analysis (ICA) (Comon, 1994; Hyvärinen et al., 2009). We’ve added the above clarifications per your suggestion.
> > > > >
> > > > >  - Comon, P. (1994). Independent component analysis, a new concept? Signal processing, 36(3), 287-314.
> > > > >
> > > > >  - Hyvärinen, A., Hurri, J., Hoyer, P. O., Hyvärinen, A., Hurri, J., & Hoyer, P. O. (2009). Independent component analysis (pp. 151-175). Springer London.
> > > > >
> > > > >
> > > > > We sincerely hope that these clarifications satisfactorily respond to your comments. All these discussions and the new figure are now part of the revised paper. We welcome any further questions or insights, and thank you once again for your vital contributions to our work!

---

> ### Author Response · Authors · 2025-01-25
> **Response to Reviewer xrmd (Part 7)**
>
> ***Continued Response to Recommended***
>
> 5. >  About saying the do-operator “simulates“ interventions
>
> Please note that our use of the term "simulates" in reference to the do-operator adheres strictly to the terminology used in Judea Pearl's foundational papers on causality. See Pearl, J. (2010, 2012):
>
> “...defined through a mathematical operator called do(x), which **simulates physical interventions** by deleting certain functions from the model…”
>
>  - Pearl, J. (2010). The foundations of causal inference. Sociological Methodology, 40(1), 75-149.
>  - Pearl, J. (2012). The do-calculus revisited. In Proceedings of the Twenty-Eighth Conference on Uncertainty in Artificial Intelligence (pp. 3-11).
>
>
> 6. >  About the notation in section 3.8
>
> We appreciate your valuable suggestions. We’ve revised throughout Section 3.8 and made the notation more readable with connections to Sections 2 and 3. Please refer to Pages 6-7 in the revised paper for more information.
>
>
> 7. >  About “where internal knowledge [of LLMs] would be more reliable”
>
>
> Thank you for your insightful comments. We concur that a primary motivation for (statistical) causal discovery is indeed to expand or enhance our understanding of causality beyond what is currently known. However, existing knowledge can also play a critical role by imposing valid constraints that assist in refining the causal discovery process. For example, when choosing between two competing partially directed acyclic graphs (CPDAGs), such as A->B->C and A<-B<-C, pre-existing knowledge that B->C allows us to definitively identify the correct causal graph as A->B->C.
>
> We’ve extended the above discussion in our revised paper.
>
>
> 8. > About training data of LLMs and how it influences the conclusion
>
> This is a very interesting question. We acknowledge that it is indeed possible that LLMs may have been exposed to publicly available datasets of causal discovery during their training phases, while it is impossible to confirm whether specific datasets were used. Given the vast and diverse data sources typically involved in training LLMs, any direct influence of such datasets on our results would likely be minimal. This is because the sheer volume of data used in training these models tends to dilute the impact of any single dataset.
>
> Furthermore, hypothetically, if these LLMs had been explicitly trained on the benchmark datasets we used, one would expect their performance in causal discovery tasks to be significantly better, potentially approaching or exceeding the level of knowing the ground truth. However, this is contradicted by our observations, where LLMs achieve only around or lower than 50% accuracy in half of our benchmark datasets. This suggests that while the LLMs might be familiar with general patterns from their training data, they do not necessarily retain specific detailed knowledge from individual datasets to the extent that it would markedly improve their performance in causal discovery tasks.
>
> We’ve extended the above discussion in our revised paper.

---

> > ### Author Response · Authors · 2025-01-25
> > **Response to Reviewer xrmd (Part 8)**
> >
> > ***Response to Minor comments and typos***
> >
> > We’ve carefully fixed all typos and really appreciate the reviewer's great efforts in reviewing our paper!
> >
> >
> >
> > We sincerely hope that these clarifications and discussions satisfactorily respond to your comments. All these discussions and results are now part of the revised paper. We welcome any further questions or insights, and thank you once again for your vital contributions to our work!

---

> ### Comment · Reviewer_xrmd · 2025-02-20
>
> I want to thank the authors for the thorough response and want to add some points for clarification.
>
> 1 ) to 4)
>
> Thank you for the clarifications. Now I would agree that the quantities measured in the experiment correspond to the interventional probabilities in Defs 2.1-2.4. But my initial remark seems to have sparked some misunderstandings.
> Concerning the do operator:
>  - My point was not that the SCM (or graphical model) is missing as a subscript from the equation. But rather that this mathematical object is *alway* defined w.r.t. to such a model, as the quote that you cited from Pearl shows.
> - I also didn’t mean to say that you *must* do confounder correction, but rather that whether you have to depends on the underlying SCM (or graph). Again, given the further explanations I think I agree that you don’t have to.
> - I strongly agree with reviewer BieH that a visualization of the assumed causal graph would improve clarity. Probably this is what I wanted in the first place, but I was too confused to articulate it. Keep in mind, that these definitions are the very first (specific) aspect of the paper that a reader sees, before being familiarized with further details of the setup.
> - In your response and the new draft I am confused why you mention the Controlled Direct Effect. Do you consider either the variable names or the data as mediator? If so, why? Again, showing the causal graph could help here.
> Concerning counterfactuals:
> - I am afraid it might be a bit too subtle to differentiate between „counterfactuals“ and „counterfactual scenarios“ where the latter are supposed to be understood as interventional quantities instead of counterfactual ones.
> - I appreciate that you agree that Defs 2.1-2.4 are not counterfactuals, but I’m confused why you also later mention that by „collecting data on Level 2 - Intervention“ you  „do not need a graphical model to help us infer Level 3 - Counterfactuals“ (for which the graphical model couldn’t help anyway).

---

### Review · Reviewer_BieH · 2024-12-22

**Summary Of Contributions:**

The authors inspect current causal reasoning capabilities of LLMs in the presence of numerical data, and examine effects of prediction performance under missing data and variable names. The authors demonstrate that LLMs are mostly unable to leverage numerical data for their causal inference and that even in the presence of reversing data columns, preexisting knowledge, induced via variable names, is valued more strongly. The authors consequently introduce a method to fine-tune and prompt LLMs in the presence of statistical values, derived from the numerical data, and show that LLMs are able to partially leverage correlational cues, but still fail to perform true causal reasoning.

**Audience:**

Yes

**Claims And Evidence:**

Yes

**Requested Changes:**

Apart from the general adjustments of some statements as mentioned above, I would like to recommend the following changes to the authors:

* The authors should be more clear about in which experiments a 'pairwise discovery' versus a 'full discovery' is utilized. I would like to recommend providing the actual prompts (specifically for the first experiments analyzing the impact of data/knowledge) in the appendix.
* Could the authors present a sensitivity analysis with regard to specific placeholder names (e.g. the bias of specific name versus the bias towards specific positions in the variable order)?
* I would like to recommend adding a brief discussion on the general ability of today's LLM on processing numerical data. To which extend does the introduction of statistical quantities in the prompt enable the model to infer the correct causal edges?
* Please provide error bars in the plots.
* I would like to, again, recommend revising the figures to bring them to an equal level of quality.

**Strengths And Weaknesses:**

**Strengths**

The authors investigate the sources of LLM's reasoning capabilities by performing a systematic evaluation on different sources of information. By withholding variable names and/or actual numerical data, the authors are able to inspect whether current LLMs are 'causal/statistical parrots', recalling information purely by correlation of variable names from previous texts, or are truly able to perform causal inference from numerical data. With their evaluation the authors push forward the existing debate on the topic of causal reasoning capabilities of LLMs.



The paper provides a good overview on works concerning the broader debate of general reasoning abilities of LLMs and on the debate of causal reasoning capabilities of LLM, specifically.



The novel presented metrics are capable of discerning variations in performance from information embedded within the LLM's weights and information obtained from reasoning over external numerical data. To the best of my knowledge no previous work has considered this approach for systematically measuring the distinct impact between these different sources of knowledge.



The employed prompts seem to the viable and suited to measure the effects under investigation. The obfuscation of variable names (and removal of data) seems to be conducted in a sound manner. The 'Reverse Causal Discovery' evaluation is suited to measure the impact of preexisting knowledge over factual data in LLM's reasoning processes. Finally, the added data augmentation of LinGAM results, together with LoRA fine-tuning, induces a data-based source of information that LLMs seem to be able to leverage partially during their reasoning process.





**Weaknesses**

While the general idea and definitions in the paper seem to be sound, the paper contains a number of smaller inclarities and weaknesses regarding the general setup, experiments and evaluations. While none of the mentioned points is critical by itself, the overall number of points reduces the overall clarity of the paper.



**Causal Graph:** In Sec. 2.1 the Authors provide several definitions (e.g. Conditional Attribution of Knowledge) to measure the impact of providing knowledge via interventions. Given that the do-calculus/interventions are used for these formulas, clarity might be improved by presenting a corresponding theoreticized causal graph to better depict the relations and expected consequences of these interventions on the outcome.



**Advantage of LLM:** In Sec. 2.2 the authors refer to an advantage of LLMs having a 'much higher accuracy' in comparison to classical methods. I would like to add, that most of the common bnlearn (and other publicly available datasets) are quite likely to be part of the training data. As it's is easier to recite known knowledge and performance drops in the absence variable names, I would like to suggest placing a more differentiated statement there. Second, the presented results for causal discovery methods seem to be particularly bad in comparison, can the authors explain this? What number of samples and hyper-parameters where used for discovery?



**Problem Setup and Prompting:** Throughout the paper, the authors mention pairwise discovery, but the presented prompts (Sec. 3.2/Fig.2) seem to focus on predicting all edges of the graph at once. Generally, it is unclear to me how the exact prompting is performed. I assume that the horizontal lines (e.g. in figure 2) are put for visualization purposes. Could the authors confirm this, and possibly provide the exact prompts as part of the appendix?

In Sec. 3.2 the authors generally recommend randomizing column order to avoid biases due to variable ordering. From this general statement, it is however not clear, whether this actually done in all of their experiments? From Appendix C it seems that variable order is randomized, however, none of the plots contain error bars. Finally, no ablation on the sensitivity to specific placeholder names is presented.



**Soundness of Setup:** Generally, the paper lacks specific details regarding the number of  data points presented per prompt. Are the presented data points sufficient for a statistical analysis? Second, I highly doubt that current models are able to perform statistical testing, as it requires extensive processing of the data and the specific prompt does not seem to allow for CoT reasoning. Consider that in table 3 GPT-4 results with 'omitted data' are on par or better 7 of 9 times than the 'raw data' setup. While the proposed approach certainly is able to boost performance, I rather expect the provision of preprocessed statistical information as the main driver of performance improvements (even though one would wish for LLMs to be able to perform these operations on their own) and I would like to suggest the authors to be more upfront about this fact.



**Reverse Causal Discovery:** Figure 3 shows that the causal order is inverted using an LLM. As this task could also be performed manually, I am wondering why it is specifically important to use an LLM here and whether the resulting variable order is correct?

From the description of the reverse causal discovery setup (Sec. 3.7) and corresponding results in Figure 5, I assume that the authors evaluate the reverse task prediction accuracy with respect to the original graph structure? While it shows the strong alignment with the original predictions, the plots actually show the 'incorrect inverse' accuracy. Could the authors please confirm this? I would like to suggest adding a brief comment in section 4 to avoid confusion.



**Metrics/Missing Entries:** While the use of TDR/FDR are rather uncommon in the field (usually precision/recall or accuracy/precision are used) and seem to be asymmetrically defined, I believe that these metrics still measure relevant information. While Table B.1, misses some FDR entries (possibly due to no edge predictions being made by the LLMs?), I believe that F1 scores should always be computable. In the limit of no edge predictions being made F1 is often reported as zero.



**Quality of Figures:** Generally, the figures are of rather mixed quality. Figures 2/3/D.1/F.1 are rasterized images, partially of low resolution, containing texts with small font size. Furthermore, the small margins between text and borders box in multiple figures make them unpleasant to read. The use of Comic Sans(?) in figures 7 and 8 seems a bit odd and does not align with the rest of the paper.

Generally, many of the figures are hard to understand by themselves. E.g. in figure 6 it is not clear to me whether the blue boxes indicate general steps of some procedure, or a general sequence of prompts to the LLM. Similarly it is unclear how the orange boxes play into the prompt. That is, whether they are inserted at some point of the instructions or play some other role? Generally, I would like to suggest revising the figures to bring them to an equal level of quality in terms of presentation and content.



**Minor**

* Citations for "Models, reasoning and inference" and "Causal inference in statistics: An overview." name "Pearl et al." as author(s). I believe that Judea Pearl is commonly cited as the sole author of these works (or the other authors should be listed explicitly by name).
* The authors might want to bring their citations (authors/venues) up to date. Many of the papers cited as preprint/arxiv have been published in the mean time.

---

> ### Author Response · Authors · 2025-01-25
> **Response to Reviewer BieH (Part 1)**
>
> Thanks for your valuable comments and suggestions! We are encouraged by your acknowledgment and positive comments on our work. Below, we summarize your questions and comments in quotes and provide our point-by-point responses. Please refer to the latest submission for the revised paper taking all your suggestions.
>
>
> ***Response to Weaknesses***
>
> 1. > **Causal Graph**: Given that the do-calculus/interventions are used for these formulas, clarity might be improved by presenting a corresponding theoreticized causal graph to better depict the relations and expected consequences of these interventions on the outcome.
>
> This is an excellent comment. Please note that in this work, we use the do-operator to express the post-intervention distribution of the target variable $Y=y$ given different interventions. Our follow-up experiment design in Section 3 allows us to directly realize these interventions to estimate the attribution scores, such as omit knowledge and then calculate the accuracy. In other words, using Pearl’s Three Layer Causal Hierarchy, we are directly collecting data on Level 2 - Intervention. As indicated in the ladder of causation, in such a scenario, we do not need a graphical model to help us infer Level 3 - Counterfactuals.
>
> In another note, we do admit that a theoreticized causal graph among the considered variables will be helpful for observational studies but it also requires a strong brief on how the LLM operates. We leave this for future research.
>
> Finally, we would like to highlight the main difference between our work and other traditional causal works is, we can **design experiments and do intervention directly to assess the proposed probabilities under different scenarios** (also known as randomized trials in the potential outcome framework). This helps to avoid using graphical modeling and identification assumptions for estimating counterfactual realities based on observational data.
>
> We’ve added all the above clarification and discussion in our revised paper.
>
>
> 2. > **Advantage of LLM**: I would like to add that most of the common bnlearn (and other publicly available datasets) are quite likely to be part of the training data. I would like to suggest placing a more differentiated statement there. Second, the presented results for causal discovery methods seem to be particularly bad in comparison, can the authors explain this? What number of samples and hyper-parameters where used for discovery?
>
> Thank you for your insightful comments. First, we recognize the possibility that LLMs might have been exposed to datasets used in our benchmarks during their training. However, given the vast volume of data these models are trained on, the influence of any single dataset is likely minimal. In our study, LLMs achieved around or below 50% accuracy in half of our benchmark datasets, yet they still outperformed classical causal discovery methods. This suggests that while LLMs may recognize general patterns from their training, they do not retain detailed knowledge specific to these datasets that would significantly enhance their performance in causal discovery tasks.
>
> Second, regarding the performance of traditional causal discovery methods, we utilized all available samples with the default hyper parameters provided by the causal-learn library. The suboptimal performance of these methods could be attributed to their reliance on assumptions not met by most of the datasets used.
>
> We have revised our manuscript to include these clarifications and have adjusted our statements to more accurately reflect these nuances.

---

> ### Author Response · Authors · 2025-01-25
> **Response to Reviewer BieH (Part 2)**
>
> ***Continued Response to Weaknesses***
>
>
> 3. > **Problem Setup and Prompting**: Throughout the paper, the authors mention pairwise discovery, but the presented prompts (Sec. 3.2/Fig.2) seem to focus on predicting all edges of the graph at once. Generally, it is unclear to me how the exact prompting is performed. Could the authors confirm this, and possibly provide the exact prompts as part of the appendix?
>
> We are grateful for your comments and would like to clarify the tasks, datasets, and prompts we used as follows.
>
> First, **in Sections 3.3-3.5 and 3.7, we conduct experiments for general causal discovery tasks with multiple variables**. Here, we consider generating samples based on nine causal discovery benchmark datasets, detailed in Sections 3.1 and 3.2. Specifically, we consider one benchmark data and randomizing the variable/column ordering as one sample, with 15 randomization in total, which yields 15*9 samples overall.
>
> **In Section 3.6, we consider another pairwise causal discovery task**, and these causal pairs (such as Father’s Height cause Child’s Height) are selected from the benchmark datasets while the effects are simulated based on a linear relationship with non-Gaussian noise. For each semi-synthetic pair, we evaluate the accuracy by randomizing the order of the two input variables. The detailed generating procedure was provided in Section 3.6.
>
> Second, Figure 2 provides a concrete illustration of our input sample using the Galton’s Family example. More specifically, the i-th sample in our experiment corresponds to one inquiry/input the LLM receives (e.g., the blue box of Figure 2) and the output it produces, including the input context, which provides the scenario for the causal reasoning task, e.g., the “system” and “user” parts in Figure 2; the embedded causal knowledge within variable names, which may carry implicit causal cues, e.g., the “Father’s Height” part in Figure 2; and numerical information such as summary statistics, providing explicit causal relationships, e.g., the lines of numbers in Figure 2; and the causal reasoning results made by an LLM.
>
> The provided Figure 1 indicates one complete sample using the LLM to provide causal discovery results, where the blue box is the i-th context, the pink box is the i-th knowledge embedded in variable names, the orange box is the i-th  numerical information, and the red box at the last is the i-th output.
>
> Third, the exact prompts used for general causal discovery tasks with multiple variables were provided in Figure 2 as well as Appendix C - Implementation Details, and the exact prompts used for pairwise causal discovery were provided in Figure D.1. Per the reviewer’s suggestion, we have provided the exact prompts in the code version in Appendix H of the revised paper.
>
> Once again, we’ve added all the above clarification in our revised paper and hope these clarify the reviewer’s concern.
>
>
> 3. > In Sec. 3.2 the authors generally recommend randomizing column order… whether this actually done in all of their experiments? From Appendix C it seems that variable order is randomized, however, none of the plots contain error bars. Finally, no ablation on the sensitivity to specific placeholder names is presented.
>
> Thank you for your thoughtful comment. As we mentioned earlier, we randomize the variable/column ordering of each benchmark dataset as one sample, with 15 randomization in total, which yields 15*9 samples overall. We would like to highlight that all the provided tables for evaluation results (i.e., Tables 2-3 as well as Tables E.1-E.3) **contain both mean and standard deviation**, which indicates the variability partly owing to column order randomization.
>
> Per reviewer’s suggestion on plots, we’ve further added error bars to the figures in our revised paper. Please refer to the new Figures 4 and 5 in the revised paper.

---

> ### Author Response · Authors · 2025-01-25
> **Response to Reviewer BieH (Part 3)**
>
> ***Continued Response to Weaknesses***
>
>
> 4. > **Soundness of Setup**: Generally, the paper lacks specific details regarding the number of data points presented per prompt. Are the presented data points sufficient for a statistical analysis? Second, I highly doubt that current models are able to perform statistical testing, as it requires extensive processing of the data and the specific prompt does not seem to allow for CoT reasoning. Consider that in table 3 GPT-4 results with 'omitted data' are on par or better 7 of 9 times than the 'raw data' setup… I would like to suggest the authors to be more upfront about this fact.
>
> Thank you for your detailed feedback. Addressing your first concern, as we mentioned in the previous responses, we structured our experimental setup to include one benchmark dataset as one sample, with the ordering of variables/columns randomized. We conducted 15 such randomizations across each of the 9 benchmark datasets, resulting in a total of 135 samples. This sample size was chosen to maximize the input data presented to the LLM within the constraints of length limits.
>
> Second, regarding the ability of current models to perform statistical testing, we acknowledge that these capabilities are indeed limited by the models' inherent processing constraints and the specificities of our prompting strategy. The results in Table 3, where 'omitted data' setups often perform as well or better than 'raw data' setups, suggest that the LLMs may not be directly performing complex statistical operations. Instead, these results may indicate that providing preprocessed statistical information, rather than relying on the LLMs’ native capabilities, is a more effective method for boosting performance.
>
> We have adjusted our discussion in Section 4 to be more transparent about the experimental results and the capabilities of current LLMs.
>
>
> 5. > **Reverse Causal Discovery**: Figure 3 shows that the causal order is inverted using an LLM. As this task could also be performed manually, I am wondering why it is specifically important to use an LLM here and whether the resulting variable order is correct?
> From the description of the reverse causal discovery setup (Sec. 3.7) and corresponding results in Figure 5, I assume that the authors evaluate the reverse task prediction accuracy with respect to the original graph structure? While it shows the strong alignment with the original predictions, the plots actually show the 'incorrect inverse' accuracy. Could the authors please confirm this? I would like to suggest adding a brief comment in section 4 to avoid confusion.
>
>
> Thank you for your valuable comments. First of all, we would like to clarify that the causal order is **NOT** inverted using an LLM. In the reverse causal discovery task as discussed in Section 3.7 and depicted in Figure 3, we indeed reverse the original topological order of the causal graph automatically. For example, if  the original topological order is, “Gene” causes “Gender” causes “Height”, with the data reflecting such a topological order, we then invert this order to "Height" → "Gender" → "Gene" using a transposed adjacency matrix, which maintains a one-to-one mapping with the original causal graph structure.
>
> We’ve made this generating process more clear in the revised paper with an enhanced Figure 3.
>
> Second, yes, we evaluate the reverse task prediction accuracy with respect to the original graph structure. To clarify, the fact that altering the causal topologies of variables does not significantly impact the LLMs' causal discovery outcomes shows and further supports that, the LLMs mainly rely on their inherent knowledge to make conclusions on the causal relationships among variables. More specifically, the numerical information that indicates the true data-generating process has been largely ignored by LLMs and the variable names are the main sources they use for judgments.
>
> We have included a detailed explanation of this point in the revised paper.

---

> > ### Author Response · Authors · 2025-01-25
> > **Response to Reviewer BieH (Part 4)**
> >
> > ***Continued Response to Weaknesses***
> >
> >
> > 6. > **Metrics/Missing Entries**: While the use of TDR/FDR are rather uncommon in the field (usually precision/recall or accuracy/precision are used) and seem to be asymmetrically defined, I believe that these metrics still measure relevant information. While Table B.1, misses some FDR entries (possibly due to no edge predictions being made by the LLMs?), I believe that F1 scores should always be computable. In the limit of no edge predictions being made F1 is often reported as zero.
> >
> > Thank you for your comments regarding the metrics used in our study. We acknowledge that while TDR (True Discovery Rate) and FDR (False Discovery Rate) might appear less common in some fields, they are indeed **standard metrics in causal discovery literature**, as discussed in Section 2.2 and supported by references such as Shimizu et al., 2006; Bühlmann et al., 2014; Ramsey et al., 2017; Zheng et al., 2018; Yu et al., 2019; and Zhu et al., 2020.
> > Regarding the missing FDR entries in Table B.1, yes, these instances occur specifically where the algorithms predict no causal edges between variables. This outcome naturally results in undefined FDR values due to a lack of false discoveries. To address this and provide a more comprehensive evaluation, we agree that including F1 scores, even if zero in cases of no edge predictions, would offer a clearer picture of the models' performance. We have updated the table to include these F1 scores in the revised paper.
> >
> >
> >
> > 7. > **Quality of Figures**: Generally, the figures are of rather mixed quality. Figures 2/3/D.1/F.1 are rasterized images, partially of low resolution, containing texts with small font size…The use of Comic Sans(?) in figures 7 and 8 seems a bit odd and does not align with the rest of the paper…Generally, many of the figures are hard to understand by themselves. E.g. in figure 6…Generally, I would like to suggest revising the figures to bring them to an equal level of quality in terms of presentation and content.
> >
> >
> > Thank you for your constructive feedback on the quality and clarity of the figures in our paper.
> > Per your suggestions, we have thoroughly revised all the figures to ensure they are of uniform quality and clarity. Specifically, for Figures 2/3/D.1/F.1, we revised them with high resolution and larger font size; for Figures 7 and 8, we removed Comic Sans and revised them with the similar style as previous figures; for Figure 6, we have clarified that the blue boxes represent sequential prompts and steps, while the yellow boxes are used to highlight four possible causal relations among two variables.
> >
> > We have ensured that all figures are now vectorized to prevent resolution issues and have adjusted the font size and style to improve readability and consistency across the document.

---

> > > ### Author Response · Authors · 2025-01-25
> > > **Response to Reviewer BieH (Part 5)**
> > >
> > > ***Response to Minor***
> > >
> > >  > About citations of Judea Pearl and updating reference
> > >
> > > We’ve carefully fixed all typos and updated the reference list. We really appreciate the reviewer's great efforts in reviewing our paper!
> > >
> > >
> > > ***Response to Requested Changes***
> > >
> > > 1. > The authors should be more clear about in which experiments a 'pairwise discovery' versus a 'full discovery' is utilized. I would like to recommend providing the actual prompts (specifically for the first experiments analyzing the impact of data/knowledge) in the appendix.
> > >
> > > We are grateful for your comments. We summarize the tasks and prompts we used as follows, and please refer to our previous response to **Problem Setup and Prompting** for more details.
> > >
> > > First, **in Sections 3.3-3.5 and 3.7, we conduct experiments for general causal discovery tasks with multiple variables**. Here, we consider generating samples based on nine causal discovery benchmark datasets, detailed in Sections 3.1 and 3.2. **In Section 3.6, we consider another pairwise causal discovery task**, and these causal pairs (such as Father’s Height cause Child’s Height) are selected from the benchmark datasets while the effects are simulated based on a linear relationship with non-Gaussian noise. The detailed generating procedure was provided in Section 3.6.
> > >
> > > Second, the exact prompts used for general causal discovery tasks with multiple variables were provided in Figure 2 as well as Appendix C - Implementation Details, and the exact prompts used for pairwise causal discovery were provided in Figure D.1. Per the reviewer’s suggestion, we have provided the exact prompts in the code version in Appendix H of the revised paper.
> > >
> > >
> > > 2. > Could the authors present a sensitivity analysis with regard to specific placeholder names (e.g. the bias of specific name versus the bias towards specific positions in the variable order)?
> > >
> > > Thank you for your query. As highlighted in our previous responses, we are conscious of potential biases stemming from both specific names and their positions within the variable order. In earlier stages of our research, we did experiments with more typical variable names such as A, B, C (or C1, C2, C3, etc.). However, we observed that even these could carry implicit biases related to common usage patterns or ordering effects in datasets. Furthermore, besides specific variable names, we also observed potential biases stemming from their positions within the variable order. To mitigate these biases, we implemented a strategy of randomizing the order of variables/columns for each benchmark dataset. This process was repeated 15 times per dataset, resulting in a total of 135 samples.
> > >
> > > Additionally, to address biases related to specific names, we employ completely fake and uninterpretable terms such as "nienet" and "feouni", where the LLM has not encountered these words during its pre-training stage. This eliminates any potential biases that could arise from the model's familiarity with certain words or sequences, thus ensuring that our findings more accurately reflect the LLM's capabilities in terms of raw causal discovery. We believe this methodology effectively addresses your concerns and provides robust findings.
> > >
> > > We’ve added all the above clarification and discussion in our revised paper.

---

> > > > ### Author Response · Authors · 2025-01-25
> > > > **Response to Reviewer BieH (Part 6)**
> > > >
> > > > ***Continued Response to Requested Changes***
> > > >
> > > > 3. > I would like to recommend adding a brief discussion on the general ability of today's LLM on processing numerical data. To which extend does the introduction of statistical quantities in the prompt enable the model to infer the correct causal edges?
> > > >
> > > > Thank you for your valuable comment. In response, as mentioned in the previous responses, we have added related discussion together with our experiment result analyses in our paper that explores how LLMs handle numerical data and the extent to which the inclusion of statistical quantities in prompts enhances their ability to correctly infer causal relationships.
> > > >
> > > >
> > > > 4. > Please provide error bars in the plots.
> > > >
> > > > We appreciate the reviewer’s suggestion on plots; we've added error bars to the figures in our revised paper. Please refer to the new Figures 4 and 5 in the new manuscript.
> > > >
> > > >
> > > > 5. > I would like to, again, recommend revising the figures to bring them to an equal level of quality.
> > > >
> > > > Following your recommendations, we have undertaken a comprehensive revision of all figures to ensure uniformity in quality and clarity across the document. This revision process included enhancing resolution, standardizing font sizes and styles, and ensuring that all graphical content is presented with clear, concise labels and annotations for better understanding.
> > > >
> > > >
> > > > We sincerely hope that these clarifications and discussions satisfactorily respond to your comments. All these discussions and results are now part of the revised paper. We welcome any further questions or insights, and thank you once again for your vital contributions to our work!

---

### Review · Reviewer_Ni2E · 2024-12-26

**Summary Of Contributions:**

This paper explores integrating causal reasoning capabilities into large language models (LLMs) through fine-tuning. The authors focus on the causal discovery task, evaluating LLM performance using three types of inputs: variable names, contextual information, and numerical features. They quantify the impact of these inputs by estimating counterfactual accuracies. Their analysis reveals that LLMs struggle to effectively utilize numerical information. To address this limitation, they fine-tune LLMs to better leverage numerical data, ultimately demonstrating that the fine-tuned models outperform all baseline methods.

**Audience:**

Yes

**Claims And Evidence:**

No

**Requested Changes:**

At this point, I am not able to arrive at a decision except with a note that I am leaning towards rejection. I request the authors to respond to my questions, and then I will reread the paper and arrive at my final decision. Overall, I the impression that I get from the current version is the that the paper is a stitch of several different things that the authors have tried in the context of causal discovery with a sheer lack of motivation.

**Strengths And Weaknesses:**

Unfortunately, I am unable to highlight any notable strengths of this paper. The "novel" contributions claimed by the authors are difficult to discern, and the presentation of the paper is convoluted and needs significant rework. I will be able to provide a more informed decision upon reading the authors' responses to the following questions:

1. The abstract and title need to be significantly toned down. While they suggest the paper aims to integrate causal reasoning abilities into LLMs, the rest of the paper only discusses causal discovery, focusing on predicting causal graphs.

2. Regarding Fig. 1: (a) Why are the blue boxes referred to as context? Aren't they just prompts? Where is the description of the variables provided -- Is that part of $k_i$ or $c_i$? Why do you refer to the variable names as "knowledge"?

3. In the metric definitions in Definitions 2.1 to 2.4, how are you conditioning on $u_i$ if they are not observed? What is the notion of $u_i$ here? Is it the seed? When you condition on $u_i$, what is the notion of probability? Should it just be the accuracy of $y_i = \hat{y_i}$?

4. How do you feed $d_i$ to the LLM? What is the context length that the LLM can accept? For instance, did it overflow for the Stock dataset? Later in the paper, the authors discuss LinGAM where they say that they generated synthetic data. Are these samples also synthetic?

5. In Section 3.3, why is $P(\hat{y_i} = y_i | do(k_i = \phi, d_i), c_i, u_i)$ referred to as the "internal knowledge" of the LLM?

6. I do not see the motivation behind changing the column names to uninterpretable words. Did you run experiments with variable names set to A, B, C (or C1, C2, C3, etc.) and observe any shortcomings that prompted you to use unusual names?

7. In Section 3.4, the authors should consider the BFS approach (https://arxiv.org/abs/2402.01207) as a baseline accuracy. I believe this is the new state of the art.

8. Theorem 3.1 needs more explanation. Why was this theorem quoted here? Why did you restrict it to linear synthetic functions? The authors may need to consider testing with non-linear functions as well.

9. Table 2 appears to be a dump of numbers. What insights do the authors want readers to take from it? I see that MAD and MAK values are negative for certain datasets. Why do you think LLMs make better random guesses than when provided with more information?

10. In Section 5, it is mentioned that "For example, when knowledge is omitted and CoT is used, the LLM conducts a correlation analysis first and then outputs the causal relation purely based on the correlation without performing a conditional independence test." How do the authors know that the LLM performed a correlation analysis? I went through Section F but could not find the answer there.

11. When fine-tuning, I believe that the numbers would also be treated as text tokens. for example, using byte-pair encodings or the like. Did the authors use adequate column separators, etc? Please provide more details on how $d_i$ was fed to the LLM.

12. I am unable to comprehend Tables 4 and 5. Do the authors understand why causal discovery accuracy increased significantly while it remained unchanged on other causal-related tasks? Can the authors state precisely what fine-tuning data and test data were used in Table 4?

13. A more meaningful experiment could have been fine-tuning the model on, say, the Alcohol dataset and testing it on the DWD dataset to see how well fine-tuning transfers across datasets.

14. In the context of question 11, I do not expect LLMs to compute the exact correlation numbers as shown in Fig. F.1. Instead a meaningful way to do that is to ask the LLM to write a python code and then execute it to obtain the correlation numbers.

---

> ### Author Response · Authors · 2025-01-25
> **Response to Reviewer Ni2E (Part 1)**
>
> Thanks for your valuable comments and suggestions. We summarize your questions and comments in quotes and provide our point-by-point responses. Please refer to the latest submission for the revised paper taking all your suggestions.
>
> ***Response to Strengths And Weaknesses***
>
> 1. > The abstract and title need to be significantly toned down. While they suggest the paper aims to integrate causal reasoning abilities into LLMs, the rest of the paper only discusses causal discovery, focusing on predicting causal graphs.
>
> We appreciate your comment. As clarified in Section 2.2, we used the causal discovery task as a representative example to detail our approach and findings, following the existing literature (see e.g., Kiciman et al., 2023; Jin et al., 2023a). While this task serves as an illustrative case for our methodologies, we acknowledge the diversity of causal tasks and the potential applicability of our framework to a broader range of scenarios in causal inference and machine learning tasks such as causal effect estimation, correlation prediction, and mathematical reasoning.
>
> In Section 5, we further considered conducting additional experiments regarding the generalizability of our fine tuned model in other causal tasks and show its robustness to catastrophic forgetting. To be specific, we consider evaluating the performance of the fine tuned model in knowledge-based pairwise causal discovery and causality event identification tasks studied by Chen et al. (2024).
>
> We’ve included all the above clarifications and discussions in the revised paper.
>
>
> 2. > Regarding Fig. 1: (a) Why are the blue boxes referred to as context? Aren't they just prompts? Where is the description of the variables provided -- Why do you refer to the variable names as "knowledge"?
>
> First, the context is formally defined in Section 2.1 which provides the scenario for the causal reasoning task, and yes, the context is part of the prompts. For instance, in the causal discovery task, the context or the background prompt for LLMs is set as “You are a helpful assistant to suggest potential causal pairs with direction (A -> B means A causes B)...Suggest causal pairs with direction among following variables after analyzing following data…Suggest ONLY the directed causal pairs without saying any other things”, as illustrated in Figures 1 and 2.
>
> Second, the description of the variables is part of knowledge as the variable information is case by case while the context is consistent under the same causal reasoning task. As formally defined in Section 2.1, the embedded causal knowledge can take many formats and considering causal discovery tasks, we view variable names as knowledge which may carry implicit causal cues. We acknowledge the diversity of causal tasks and would like to point out that our framework, for instance, variable names as knowledge can be certainly extended depending on the specific task of interest.
>
> Per the reviewer’s suggestion, we revised Figure 1 with a more informative caption, which indicates one example using the LLM to provide causal discovery results, where the blue box is the i-th context, the pink box is the i-th knowledge embedded in variable names, the orange box is the i-th numerical information, and the red box at the last is the i-th output. A more concrete example is provided in Figure 2 using the Galton’s Family as one sample.
>
> We’ve included all the above clarifications and discussions in the revised paper.

---

> > ### Author Response · Authors · 2025-01-25
> > **Response to Reviewer Ni2E (Part 2)**
> >
> > ***Continued Response to Strengths And Weaknesses***
> >
> >
> > 3. > In the metric definitions in Definitions 2.1 to 2.4, how are you conditioning on $u_i$. Is it the seed? When you condition on $u_i$ what is the notion of probability? Should it just be the accuracy of $y_i=\widehat{y}_i$?
> >
> > Thank you for pointing this out. We’ve corrected $u_i$ more precisely as all other controlled confounding variables that we do not include as intervention, such as training data, language model version, as well as the seed as you mentioned, etc. In this paper, we use the do-operator to express the post-intervention distribution of the target variable $Y=y$ given different interventions. Here, please note both $y_i$ and $\widehat{y}_i$ are the set of causal pairs as ground truth and as what are provided by an LLM, respectively. Hence, $y_i=\widehat{y}_i$ means how many causal relations produced by LLMs are aligned with the ground truth. The notion of probability for $y_i=\widehat{y}_i$ means the accuracy of LLMs’ causal reasoning results given different intervention.
> >
> > Our follow-up experiment design in Section 3 allows us to directly realize these interventions to estimate the attribution scores, such as omit knowledge and then calculate the accuracy. We are only interested in changed accuracy under two interventions, also known as randomized trials in the potential outcome framework, the unmeasured confounders thus are less concerned. We’ve revised the paper accordingly and appreciate your comment again.
> >
> >
> > 4. > How do you feed $d_i$ to the LLM? What is the context length that the LLM can accept? For instance, did it overflow for the Stock dataset? Later in the paper, the authors discuss LinGAM where they say that they generated synthetic data. Are these samples also synthetic?
> >
> > We are grateful for your comments and would like to clarify the tasks, datasets, and prompts we used as follows. As discussed in Section 2.1, the numerical data $d_i$ represents explicit causal relationships, which may take forms as summary statistics, table columns, or other numerical information.
> >
> > **In Sections 3.3-3.5 and 3.7, we conduct experiments for general causal discovery tasks with multiple variables**. Here, we consider generating samples based on nine causal discovery benchmark datasets, detailed in Sections 3.1 and 3.2. Specifically, we consider one benchmark data and randomizing the variable/column ordering as one sample, with 15 randomization in total, which yields 15*9 samples overall. And the Stock dataset serves as one benchmark dataset.
> >
> > **In Section 3.6, we consider another pairwise causal discovery task**, and these causal pairs (such as Father’s Height cause Child’s Height) are selected from the benchmark datasets while the effects are simulated based on a linear relationship with non-Gaussian noise. For each semi-synthetic pair, we evaluate the accuracy by randomizing the order of the two input variables. The detailed generating procedure was provided in Section 3.6. All the pairs in the Stock dataset are considered to generate these synthetic data.
> >
> > Figure 2 provides a concrete illustration of our input sample using the Galton’s Family example. More specifically, the i-th sample in our experiment corresponds to one inquiry/input the LLM receives (e.g., the blue box of Figure 2) and the output it produces, including the input context, which provides the scenario for the causal reasoning task, e.g., the “system” and “user” parts in Figure 2; the embedded causal knowledge within variable names, which may carry implicit causal cues, e.g., the “Father’s Height” part in Figure 2; and numerical information such as summary statistics, providing explicit causal relationships, e.g., the lines of numbers in Figure 2; and the causal reasoning results made by an LLM.
> >
> > The exact prompts used for general causal discovery tasks with multiple variables were provided in Figure 2 as well as Appendix C - Implementation Details, and the exact prompts used for pairwise causal discovery were provided in Figure D.1. In the revised paper, we’ve further provided the exact prompts in the code version in Appendix H.
> >
> > Once again, we’ve added all the above clarification in our revised paper and hope these clarify the reviewer’s concern.

---

> > > ### Author Response · Authors · 2025-01-25
> > > **Response to Reviewer Ni2E (Part 3)**
> > >
> > > ***Continued Response to Strengths And Weaknesses***
> > >
> > >
> > > 5. > In Section 3.3, why is it referred to as the "internal knowledge" of the LLM?
> > >
> > > Thank you for your comment. As we discussed in the earlier response, the embedded causal knowledge can take many formats and considering the causal discovery task of interest, we view variable names as knowledge which may carry implicit causal cues. In Section 3.3, we replaced the variable names with random non-meaningful fake terms like "bryoto", "nienet", and "feouni" and conducted experiments by randomizing the variable order to omit any bias due to word position or specific letter ordering. Hence, we refer such an experiment to examine the internal knowledge of LLMs, referred to $\mathbb{P}\left(\widehat{y}_i=y_i \mid do \left(k_i=\emptyset, d_i=d_i\right), c_i,u_i\right)$ in Definition 2.1.
> > >
> > >
> > >
> > > 6. > I do not see the motivation behind changing the column names to uninterpretable words. Did you run experiments with variable names set to A, B, C (or C1, C2, C3, etc.) and observe any shortcomings that prompted you to use unusual names?
> > >
> > > Thank you for your question about the rationale behind using uninterpretable words as column names in our experiments. In earlier stages of our research, we did experiments with more typical variable names such as A, B, C (or C1, C2, C3, etc.). However, we observed that even these could carry implicit biases related to common usage patterns or ordering effects in datasets. By employing completely fake and uninterpretable terms, we eliminate any potential biases that could arise from the model's familiarity with certain words or sequences, thus ensuring that our findings more accurately reflect the LLM's capabilities in terms of raw causal discovery. Hence, we opted for the approach by changing the column names to uninterpretable words, such as "nienet" and "feouni," to simulate the "omit knowledge" case, where the LLM has not encountered these words during its pre-training stage.
> > >
> > >
> > > 7. > In Section 3.4, the authors should consider the BFS approach as a baseline accuracy. I believe this is the new state of the art.
> > >
> > > Thank you for pointing out this approach. While our main objective is not to establish a new state-of-the-art method for maximizing accuracy in causal discovery, we understand the importance of benchmarking against relevant and contemporary methods. We acknowledge the BFS approach by Jiralerspong et al. (2024) as a significant development in the field and have included a discussion of this method in our introduction to contextualize our work within current research trends.
> > >
> > > Our focus, however, is on investigating how LLMs handle the task of causal discovery, specifically examining the attribution of causal relationships rather than solely striving for high accuracy. This exploration aims to contribute insights into the capabilities and limitations of LLMs in understanding and processing causal information. We believe this focus complements the broader landscape of causal discovery research.
> > >
> > >
> > > 8. > Theorem 3.1 needs more explanation. Why was this theorem quoted here? Why did you restrict it to linear synthetic functions? The authors may need to consider testing with non-linear functions as well.
> > >
> > > Thank you for your feedback on Theorem 3.1 from Shimizu et al. (2006), which we referenced to lay a theoretical foundation for our pairwise causal discovery task. We chose to initially focus on linear synthetic functions because they provide a clear and manageable framework for establishing the identification of causal graphs, which is critical for validating the LLM's causal discovery capabilities. While we recognize the importance of exploring non-linear functions to better mirror the complexities of real-world causal relationships, non-linear functions pose more complex challenges for causal discovery, especially the data and validation process. In our fine-tuned model and related experiments, we utilized summary statistics as the primary form of numerical information provided to the LLMs. Whether the underlying functions are linear or nonlinear, the approach to using summary statistics remains consistent.

---

> > > > ### Author Response · Authors · 2025-01-25
> > > > **Response to Reviewer Ni2E (Part 4)**
> > > >
> > > > ***Continued Response to Strengths And Weaknesses***
> > > >
> > > >
> > > > 9. > Table 2 appears to be a dump of numbers. What insights do the authors want readers to take from it? I see that MAD and MAK values are negative for certain datasets. Why do you think LLMs make better random guesses than when provided with more information?
> > > >
> > > > Thank you for your inquiry regarding Table 2 and the insights it offers. The table is intended to highlight two key points about the performance of LLMs in causal discovery tasks across various datasets. Firstly, the high MAK scores reported in Table 2 show the pivotal role that inherent knowledge plays in enabling LLMs to identify causal relationships. This observation suggests that the pre-existing knowledge embedded within LLMs significantly influences their ability to parse and understand causal structures from the data provided. Secondly, the high CAK scores demonstrate that even when numerical data is present, the naming of variables significantly enhances the LLMs' accuracy in causal discovery. This improvement is attributed to the LLMs leveraging their extensive internal knowledge base, which helps them make more informed inferences about causal relationships.
> > > >
> > > > Regarding the negative values mentioned, upon careful review, **we did not find any MAK values in Table 2 to be negative**. We would appreciate further clarification on this point from the reviewer.
> > > >
> > > > However, the presence of negative MAD values for certain datasets under some LLM conditions suggests that including numerical information might occasionally mislead LLMs in discerning certain causal relationships. This phenomenon could be due to the model processing this information in a way that conflicts with its training or inherent biases, leading to poorer performance compared to when it relies on more generalized knowledge. This observation is critical as it indicates potential areas for improving how numerical data is presented and integrated into the causal analysis performed by LLMs, which indeed motivated our fine tuned model in Section 5.
> > > >
> > > >
> > > > 10. > In Section 5, it is mentioned that "For example, when knowledge is omitted and CoT is used, the LLM conducts a correlation analysis first and then outputs the causal relation purely based on the correlation without performing a conditional independence test." How do the authors know that the LLM performed a correlation analysis?
> > > >
> > > > Thank you for your query about how we determined that the LLM performs a correlation analysis in the scenario described in Section 5. The inference that the LLM engages in correlation analysis is drawn from the observed outputs displayed in Figure F.1, where the LLMs were instructed to use the Chain of Thought (CoT) approach for causal discovery with both data and knowledge inputs provided.
> > > >
> > > > In this figure, the LLMs' responses typically begin with statements that explicitly analyze or mention correlations between variables, indicating that they assess the strength and direction of relationships as an initial step. Following this analysis, the models then proceed to deduce causal relationships, often without mentioning or performing a conditional independence test, which would be a critical step in a more thorough causal analysis.
> > > >
> > > > This pattern in the responses suggests that the LLMs prioritize correlation as a basis for causal inference in this context. It's important to note, however, that this is an interpretation based on the models' generated text outputs, and it does not imply that the LLM internally calculates correlation coefficients or conducts statistical tests as a human would. Rather, it reflects the models' training on how to approach causal discovery tasks presented in this manner.
> > > >
> > > > We’ve revised Figure F.1 and included the related discussions in the new manuscript.

---

> > > > > ### Author Response · Authors · 2025-01-25
> > > > > **Response to Reviewer Ni2E (Part 5)**
> > > > >
> > > > > ***Continued Response to Strengths And Weaknesses***
> > > > >
> > > > > 11. > When fine-tuning, I believe that the numbers would also be treated as text tokens. for example, using byte-pair encodings or the like. Did the authors use adequate column separators, etc? Please provide more details on how was fed to the LLM.
> > > > >
> > > > > Thank you for your inquiry. Indeed, during fine-tuning, the handling of numerical data as text tokens is a crucial consideration, especially given the common use of byte-pair encodings in LLMs. To ensure effective processing by the LLM,  we provided the LLM with intermediate results in terms of summary statistics from the LiNGAM process, which is crucial for understanding the causal structure between variables. These summary statistics include estimated coefficients and independence tests, formatted in a clear, structured manner that mirrors the logical flow of data analysis.
> > > > >
> > > > > We did not require the LLM to perform the LiNGAM calculations itself; rather, the model was tasked with integrating these summary statistics to deduce the final causal relationships. This approach allows the LLM to focus on applying causal reasoning over pre-processed, algorithmically derived data points, rather than on computational tasks.
> > > > >
> > > > > Regarding your question about column separators and data formatting: we used adequate separators and structured the data inputs to ensure that they are distinct and recognizable by the LLM. This was achieved by carefully placing data in formatted tables, where each variable and its corresponding results from LiNGAM were clearly separated and labeled.
> > > > >
> > > > > We’ve provided the related discussions in the revised manuscript.
> > > > >
> > > > >
> > > > > 12. > I am unable to comprehend Tables 4 and 5. Do the authors understand why causal discovery accuracy increased significantly while it remained unchanged on other causal-related tasks? Can the authors state precisely what fine-tuning data and test data were used in Table 4?
> > > > >
> > > > > Thank you for your question. The significant improvement in causal discovery accuracy noted in Table 4 can be attributed primarily to the application of LoRA (Low-Rank Adaptation) fine-tuning techniques. LoRA fine-tuning is specifically designed to enhance the model's ability to process and integrate numerical data effectively into its decision-making process. This method adjusts the model's attention mechanisms and feed-forward networks in a way that enhances its capacity to recognize and utilize patterns in numerical inputs, which is crucial for tasks like causal discovery that rely heavily on data interpretation.
> > > > >
> > > > > For Table 5, the reason why performance on other causal-related tasks remained largely unchanged is that these tasks often depend more on abstract reasoning about causality and less on numerical data processing. Since our fine-tuning approach with LoRA specifically targeted enhancements in numerical data interpretation, tasks that required reasoning beyond what was presented numerically did not show the same level of improvement. This reflects the orthogonality of the tasks within the realm of causal analysis—while some rely heavily on data manipulation, others are more dependent on conceptual and theoretical understanding of causality.
> > > > >
> > > > > Figure 6 illustrates the data generation process in detail for fine tuning. Specifically, the instructions for the fine-tuned datasets are composed of three parts: (1) the instruction, (2) the definition of the variable names, and (3) one of the four possible outcomes generated by running LiNGAM. The outputs of the fine-tuned datasets depend on how the LLM integrates the numerical results with its own knowledge to determine the final causal pair. The fine-tuning data and test data are generated using the same data generation pipeline, but they involve distinct groups of causal pairs.
> > > > >
> > > > > We have clarified Figure 6 with a new caption that the blue boxes represent sequential prompts and steps, while the yellow boxes are used to highlight four possible causal relations among two variables.

---

> > > > > > ### Author Response · Authors · 2025-01-25
> > > > > > **Response to Reviewer Ni2E (Part 6)**
> > > > > >
> > > > > > ***Continued Response to Strengths And Weaknesses***
> > > > > >
> > > > > > 13. > A more meaningful experiment could have been fine-tuning the model on, say, the Alcohol dataset and testing it on the DWD dataset to see how well fine-tuning transfers across datasets.
> > > > > >
> > > > > > We appreciate the opportunity to clarify our methodology as it indeed aligns with your suggestion. In our current experimental setup, we have carefully separated the datasets into two distinct groups for training and validation to prevent any overlap. This segregation ensures that the fine-tuned model is evaluated on entirely unseen data, thereby maintaining the integrity of our validation process.
> > > > > >
> > > > > >
> > > > > >
> > > > > > 14. > In the context of question 11, I do not expect LLMs to compute the exact correlation numbers as shown in Fig. F.1. Instead a meaningful way to do that is to ask the LLM to write a python code and then execute it to obtain the correlation numbers.
> > > > > >
> > > > > >
> > > > > > We would like to clarify that in our observations, as depicted in Figure F.1, LLMs DO generate correlation numbers during the Chain of Thought process. As we mentioned in the previous response to Point #1, the numbers produced by LLMs in such contexts should be interpreted with caution as they may not necessarily reflect actual computations but rather the model's ability to generate plausible responses based on its training. This phenomenon, sometimes referred to as "hallucination", is an important aspect of LLM behavior and is indeed a direction worth exploring further. However, in the current scope of our paper, our primary focus is on examining how LLMs derive and understand causal relationships rather than on their computational accuracy.
> > > > > >
> > > > > >
> > > > > > We sincerely hope that these clarifications and discussions satisfactorily respond to your comments. All these discussions and results are now part of the revised paper. We welcome any further questions or insights, and thank you once again for your vital contributions to our work!

---

> > > > > > > ### Comment · Reviewer_Ni2E · 2025-03-07
> > > > > > > **Rebuttal Feedback**
> > > > > > >
> > > > > > > Dear Authors,
> > > > > > >
> > > > > > > Thank you for your detailed feedback on the questions. Please find my pointwise responses below:
> > > > > > >
> > > > > > > 1. I still disagree with the claims and the title of the paper. Causal inference encompasses much more than causal discovery, so I believe the title should be toned down.
> > > > > > >
> > > > > > > 3. Definitions 2.1 and 2.2 should be more concrete. Conditioning on a latent variable makes the quantities unidentifiable. Perhaps probabilities in Definitions 2.1–2.4 should be defined by taking an expectation over $u_i$, averaging over multiple calls to the LLM. In experiments, one may estimate this with a single sample, but I wonder on what is the theoretically correct approach.
> > > > > > >
> > > > > > > 8. I still do not see the relevance of quoting the theorem in your paper. It would be more appropriate to simply reference the original paper.  How is this fact helping the LLM unless you are explicitly forcing it to perform the regression and estimate the residues?
> > > > > > >
> > > > > > > 9. Apologies for the confusion. I was referring to certain MAD values being negative.  I can understand that numerical data is not used by the LLMs. But why are they detrimental?
> > > > > > >
> > > > > > > 10. This response is not very convincing. Why should LLMs be expected to output correlation coefficients directly? A more practical approach would be to have the LLM generate Python code that, when executed, computes the correlation coefficient. Even setting this aside, I do not see how this experiment naturally leads to the conclusion that LLMs use correlation.

---

### Review · Reviewer_6i8P · 2025-01-13

**Summary Of Contributions:**

This study investigates the integration of causal reasoning into large language models (LLMs). The authors reveal that LLMs generally struggle to utilize numerical data effectively for causal inference. Moreover, they find that even when data columns are reversed, the models place greater emphasis on prior knowledge inferred from variable names. To overcome this limitation, the authors fine-tune LLMs to enhance their ability to leverage numerical data. Their results show that the fine-tuned models significantly outperform all baseline approaches.

**Audience:**

Yes

**Claims And Evidence:**

No

**Requested Changes:**

- Table 2 indicates that MAK exceeds CAK in the majority of cases. I find it counterintuitive. Could you help interpret this??
- The role of Reverse Causal Discovery to understand the reasoning ability of LLMs is quite unclear to me. Could you please clarify
- In section 3.2: "Our goal is to generate the best prompt to improve the accuracy of LLMs' causal discovery"—how do you determine that your prompt is truly the "best"?
- What criteria were used to select the LLMs for the experiments?
- Table 4 caption: "Accuracy of LLMs for different datasets in the pairwise causal discovery analyses"—however, the table does not mention the datasets anywhere.

**Strengths And Weaknesses:**

The paper's presentation is so unclear that it becomes challenging to evaluate its contributions. A significant rewrite is required.
**Strengths**
1. The author proposes few metrics to assess the causal reasoning capabilities of LLMs.
2. The results are promising and could inspire the development of new algorithms for causal discovery using LLMs.

**Weaknesses**
1. Figure 1 does not clearly convey the message the author intends to communicate.
2. I couldn't grasp the point the author intended to convey through Figure 2.
3. In Section 2.1, the author should provide a clear explanation of $c_{i}, k_{i}$, and  $d_{i}$ with appropriate examples.
4. How are you conditioning on $u$ potential unmeasured confounders?
5. Section 2.1: Causal Attribution Model -  What do you mean by "model" in this context? You propose several metrics here to assess the reasoning ability of LLMs.
6. "We observed that LLMs may infer causal relationships based on the order of data columns, possibly influenced by biases from the pre-training phase." While this is highly plausible, where is the supporting evidence? "Specifically, LLMs often assume a causal relationship where the first variable causes the second, and the second causes the third." What evidence supports this assertion?
7. What is the interpretation of a negative attribution score (Table 2)? I am somewhat skeptical about using the term "attribution" in the paper, as it holds a different meaning in the XAI literature, and I don’t see a clear connection.
8. Section 3.7:
"We achieve this by manipulating the structure of the datasets while keeping their data content intact."
"If changing causal topologies has minimal impact on LLMs’ discovery outcomes, this implies that LLMs rely primarily on their inherent knowledge to infer causal relationships among variables."
Why would this be the case, given that the data content remains unchanged?
9. The data generation pipeline described in Section 5 is not clearly explained.

---

> ### Author Response · Authors · 2025-01-25
> **Response to Reviewer 6i8P (Part 1)**
>
> Thanks for your valuable comments and suggestions. We summarize your questions and comments in quotes and provide our point-by-point responses. Please refer to the latest submission for the revised paper taking all your suggestions.
>
>
> ***Response to Weaknesses***
>
> 1. > Figure 1 does not clearly convey the message the author intends to communicate.
>
> Thanks for your helpful comment. The first panel of Figure 1 presents one sample using the LLM to provide causal discovery results, where the blue box is the i-th context, the pink box is the i-th knowledge embedded in variable names, the orange box is the i-th numerical information, and the red box at the last is the i-th output. The second panel shows the generation of one counterfactual sample where the variable names were replaced by non-meaningful letters; and the third one describes our proposed causal attribution framework, where the knowledge is omitted corresponding to the counterfactual scenario in the second panel.
>
> We’ve revised the caption of Figure 1 according to the above clarification.
>
>
> 2. > I couldn't grasp the point the author intended to convey through Figure 2.
>
> Figure 2 provides a concrete illustration of our input sample using the Galton’s Family example. More specifically, the i-th sample in our experiment corresponds to one inquiry/input the LLM receives (e.g., the blue box of Figure 2) and the output it produces, including the input context, which provides the scenario for the causal reasoning task, e.g., the “system” and “user” parts in Figure 2; the embedded causal knowledge within variable names, which may carry implicit causal cues, e.g., the “Father’s Height” part in Figure 2; and numerical information such as summary statistics, providing explicit causal relationships, e.g., the lines of numbers in Figure 2; and the causal reasoning results made by an LLM.
>
> We’ve added all the above clarification in our revised paper and hope these clarify the reviewer’s concern.
>
>
> 3. > In Section 2.1, the author should provide a clear explanation with appropriate examples.
>
> Thank you for your valuable suggestion. For the $i$-th sample in our dataset, we decompose the input into several distinct elements as follows: the input context $c_i$, which provides the scenario for the causal reasoning task, e.g., the “system” and “user” parts in Figure 2; the embedded causal knowledge $k_i$ can take many formats and considering causal discovery tasks, we view variable names as knowledge which may carry implicit causal cues, e.g., the “Father’s Height” part in Figure 2; and the numerical data $d_i$, representing explicit causal relationships which may take forms as summary statistics or other numerical information, e.g., the lines of numbers in Figure 2. All other controlled confounding variables that we do not include as interventions, such as training data, language model version, etc., are denoted as $u_i$. The goal is to compare the causal reasoning made by an LLM, represented as $\widehat{y}_i$, against the true causal responses $y_i$.
>
> The provided Figure 1 indicates such an complete sample using the LLM to provide causal discovery results, where the blue box is the i-th context, the pink box is the i-th knowledge embedded in variable names, the orange box is the i-th  numerical information, and the red box at the last is the i-th output.
>
> We’ve added all the above clarification in our revised paper.
>
>
> 4. > How are you conditioning on potential unmeasured confounders?
>
> Thank you for pointing this out. We’ve corrected $u_i$ more precisely as all other controlled confounding variables that we do not include as intervention, such as training data, language model version, etc. In this paper, we use the do-operator to express the post-intervention distribution of the target variable $Y=y$ given different interventions. Our follow-up experiment design in Section 3 allows us to directly realize these interventions to estimate the attribution scores, such as omit knowledge and then calculate the accuracy. We are only interested in changed accuracy under two interventions, also known as randomized trials in the potential outcome framework, the unmeasured confounders thus are less concerned.
>
> We’ve revised the paper accordingly and appreciate your comment again.

---

> ### Author Response · Authors · 2025-01-25
> **Response to Reviewer 6i8P (Part 2)**
>
> ***Continued Response to Weaknesses***
>
> 5. > Section 2.1: Causal Attribution Model - What do you mean by "model" in this context? You propose several metrics here to assess the reasoning ability of LLMs.
>
> Thank you for your question about the use of the term "model" in Section 2.1 regarding the Causal Attribution Model. In this context, we refer to "model" not in the sense of a statistical or machine learning model, but rather as a framework. This framework comprises a series of definitions and metrics designed to assess and quantify the contributions of different components within LLMs to their overall reasoning ability.
>
>
> 6. > "We observed that LLMs may infer causal relationships based on the order of data columns, possibly influenced by biases from the pre-training phase." While this is highly plausible, where is the supporting evidence? "Specifically, LLMs often assume a causal relationship where the first variable causes the second, and the second causes the third." What evidence supports this assertion?
>
> Thank you for your query. In earlier stages of our research, we did experiments with more typical variable names such as A, B, C (or C1, C2, C3, etc.). However, we observed that even these could carry implicit biases related to common usage patterns or ordering effects in datasets. Furthermore, besides specific variable names, we also observed potential biases stemming from their positions within the variable order.  To mitigate these biases, we implemented a strategy of randomizing the order of variables/columns for each benchmark dataset. This process was repeated 15 times per dataset, resulting in a total of 135 samples. Additionally, to address biases related to specific names, we employ completely fake and uninterpretable terms such as "nienet" and "feouni", where the LLM has not encountered these words during its pre-training stage. This eliminates any potential biases that could arise from the model's familiarity with certain words or sequences, thus ensuring that our findings more accurately reflect the LLM's capabilities in terms of raw causal discovery.
>
> We’ve added all the above clarification and discussion in our revised paper.
>
>
> 7. > What is the interpretation of a negative attribution score (Table 2)? I am somewhat skeptical about using the term "attribution" in the paper, as it holds a different meaning in the XAI literature, and I don’t see a clear connection.
>
> First,  the presence of negative MAD and CAD values for certain datasets under some LLM conditions suggests that including numerical information might occasionally mislead LLMs in discerning certain causal relationships. This phenomenon could be due to the model processing this information in a way that conflicts with its training or inherent biases, leading to poorer performance compared to when it relies on more generalized knowledge. This observation is critical as it indicates potential areas for improving how numerical data is presented and integrated into the causal analysis performed by LLMs, which indeed motivated our fine tuned model in Section 5.
>
> Second, thank you for your valuable comment on the connection between XAI and attribution. In our study, we use the term "attribution" to denote the assessment of how various components of LLMs, such as input features or internal mechanisms, contribute to the model's overall causal reasoning abilities. Our goal is to quantify the influence of these elements in performing specific tasks, which aligns with the broader concept of attribution in XAI but focuses more specifically on causal reasoning processes.
>
> We’ve included and discussed more related literature in Section 6 and moved Appendix A “More related works - Attribution Models” to the main text to clarify this connection.

---

> > ### Author Response · Authors · 2025-01-25
> > **Response to Reviewer 6i8P (Part 3)**
> >
> > ***Continued Response to Weaknesses***
> >
> >
> > 8. > Section 3.7: "We achieve this by manipulating the structure of the datasets while keeping their data content intact." "If changing causal topologies has minimal impact on LLMs’ discovery outcomes, this implies that LLMs rely primarily on their inherent knowledge to infer causal relationships among variables." Why would this be the case, given that the data content remains unchanged?
> >
> > Thank you for your valuable comments. First of all, we would like to clarify the reverse causal discovery task as discussed in Section 3.7 and depicted in Figure 3. Here, we reverse the original topological order of the causal graph automatically. For example, if the original topological order is, “Gene” causes “Gender” causes “Height”, with the data reflecting such a topological order, we then invert this order to "Height" → "Gender" → "Gene" using a transposed adjacency matrix, which maintains a one-to-one mapping with the original causal graph structure. Hence, **though the data content remains unchanged, the underlying causal structure is reversed**. We’ve made this generating process more clear in the revised paper with an enhanced Figure 3.
> >
> > Second, we evaluate the reverse task prediction accuracy with respect to the original graph structure. To clarify, the fact that altering the causal topologies of variables does not significantly impact the LLMs' causal discovery outcomes shows and further supports that, the LLMs mainly rely on their inherent knowledge to make conclusions on the causal relationships among variables. More specifically, the numerical information that indicates the true data-generating process has been largely ignored by LLMs and the variable names are the main sources they use for judgments. We have included a detailed explanation of this point in the revised paper.
> >
> >
> > 9. > The data generation pipeline described in Section 5 is not clearly explained.
> >
> > Figure 6 illustrates the data generation process in detail. Specifically, the instructions for the fine-tuned datasets are composed of three parts: (1) the instruction, (2) the definition of the variable names, and (3) one of the four possible outcomes generated by running LiNGAM. The outputs of the fine-tuned datasets depend on how the LLM integrates the numerical results with its own knowledge to determine the final causal pair.
> >
> > We have clarified Figure 6 with a new caption that the blue boxes represent sequential prompts and steps, while the yellow boxes are used to highlight four possible causal relations among two variables.

---

> ### Author Response · Authors · 2025-01-25
> **Response to Reviewer 6i8P (Part 4)**
>
> ***Response to Requested Changes***
>
>
> 1. > Table 2 indicates that MAK exceeds CAK in the majority of cases. I find it counterintuitive. Could you help interpret this?
>
> Thanks for your valuable comment. Upon careful review, **we did not find that MAK exceeds CAK in the majority of cases in Table 2** but these two scores are quite close considering their standard deviation. Specifically, one can observe that MAK exceeds CAK usually happening in complicated benchmark datasets with multiple variables such as Sachs, Stock, and Arrhythmia. This suggests that including numerical information might occasionally mislead LLMs in discerning certain causal relationships when handling the complex numerical information. This is consistent with our response to your Point #7 regarding the negative CAD and MAD.
>
> To summarize, Table 2 is intended to highlight two key points about the performance of LLMs in causal discovery tasks across various datasets. Firstly, the high MAK scores reported in Table 2 show the pivotal role that inherent knowledge plays in enabling LLMs to identify causal relationships. This observation suggests that the pre-existing knowledge embedded within LLMs significantly influences their ability to parse and understand causal structures from the data provided. Secondly, the high CAK scores demonstrate that even when numerical data is present, the naming of variables significantly enhances the LLMs' accuracy in causal discovery. This improvement is attributed to the LLMs leveraging their extensive internal knowledge base, which helps them make more informed inferences about causal relationships.
>
> We’ve added all the above clarification and discussion in our revised paper.
>
>
> 2. > The role of Reverse Causal Discovery to understand the reasoning ability of LLMs is quite unclear to me. Could you please clarify
>
>
> Thank you for your valuable comments. We’ve provided detailed response to your Point #8 regarding the Reverse Causal Discovery. To summarize, **though the data content remains unchanged, the underlying causal structure is reversed in the reverse causal discovery**. We evaluate the reverse task prediction accuracy with respect to the original graph structure.
> If altering the causal topologies of variables does not significantly impact the LLMs' causal discovery outcomes, this shows and further supports that, the LLMs mainly rely on their inherent knowledge to make conclusions on the causal relationships among variables. More specifically, the numerical information that indicates the true data-generating process has been largely ignored by LLMs and the variable names are the main sources they use for judgments.
>
> We have included a detailed explanation of this point in the revised paper.

---

> > ### Author Response · Authors · 2025-01-25
> > **Response to Reviewer 6i8P (Part 5)**
> >
> > ***Continued Response to Requested Changes***
> >
> > 3. > In section 3.2: "Our goal is to generate the best prompt to improve the accuracy of LLMs' causal discovery"—how do you determine that your prompt is truly the "best"?
> >
> > Thank you for pointing this out. We’ve rewrite this sentence to be more accurate as: “Our goal is to generate a comprehensive prompt to enhance the accuracy of LLMs' causal discovery, establishing the baseline accuracy component, i.e., the first term ($\mathbb{P}\left(\widehat{y}_i=y_i \mid d o\left(k_i=k_i, d_i=d_i\right), c_i,u_i\right) $) in Definitions 2.1 and 2.2.
> >
> >
> > 4. > What criteria were used to select the LLMs for the experiments?
> >
> > For the experiments, we selected LLMs from the closed-source ChatGPT and Claude 2 series, which achieved top scores in benchmarks such as MMLU and GSM8K. Additionally, we included one of the most advanced open-source LLMs available at the time, LLaMA2 13B.
> >
> >
> > 5. > Table 4 caption: "Accuracy of LLMs for different datasets in the pairwise causal discovery analyses"—however, the table does not mention the datasets anywhere.
> >
> > Thanks for pointing out this typo! We’ve deleted “different datasets” in the caption to avoid future confusion in the revised paper.
> >
> >
> >
> > We sincerely hope that these clarifications and discussions satisfactorily respond to your comments. All these discussions and results are now part of the revised paper. We welcome any further questions or insights, and thank you once again for your vital contributions to our work!

---

### Decision · Action_Editor_CJiM · 2025-03-14

**Recommendation:** Reject

**Comment:**

Thank you for submitting your paper to TMLR. The paper poses interesting questions on LLMs' use of internal knowledge versus numerical data for causal discovery. There is also an effort to finetune LMs for better use of numerical data. However, as some reviewers noted, the paper is unable to do justice to these questions due to:
1) Lack of clarity in explaining experimental setup and discussing results. For instance, even after rebuttal, it is unclear exactly what kind of data is used for finetuning and why LinGAM results are used for finetuning but the original data is provided in the earlier attribution experiments. Claim about LLM using correlation when using numerical data is not fully supported, etc. For a detailed list, see the reviews.
2) Lack of theoretical rigour that leads to unnecessary confusion (e.g., counterfactuals versus intervention).

For these reasons, I'm unable to recommend Accept. I believe that the paper has merit, so I'd encourage the authors to take the reviews into account and submit a stronger version (if the authors choose to resubmit).

**Audience:**

Yes, the paper's contributions would be of interest to TMLR audience

**Claims And Evidence:**

* Significant lack of clarity to the extent that it hinders evaluation of contributions
* Overclaiming: paper only focuses on causal discovery but title claims causal reasoning
* Lack of theoretical rigor that leads to confusion

**Resubmission Of Major Revision:**

The authors may consider submitting a major revision at a later time.